# Minimum norm interpolation by perceptra: Explicit regularization and implicit bias

**Jiyoung Park**
Department of Statistics
Texas A&M University
wldyddl5510@tamu.edu

**Ian Pelakh**
Department of Mathematics
Iowa State University
ispelakh@iastate.edu

**Stephan Wojtowytsch**
Department of Mathematics
University of Pittsburgh
s.woj@pitt.edu

## Abstract

We investigate how shallow ReLU networks interpolate between known regions. Our analysis shows that empirical risk minimizers converge to a minimum norm interpolant as the number of data points and parameters tends to infinity when a weight decay regularizer is penalized with a coefficient which vanishes at a precise rate as the network width and the number of data points grow. With and without explicit regularization, we numerically study the implicit bias of common optimization algorithms towards known minimum norm interpolants.

## 1 Introduction

Modern neural networks mostly operate in an overparametrized regime, i.e. they possess more tunable parameters than the number of data points contributing to the loss function. Safran and Shamir [2018a], E et al. [2019b], Du et al. [2018], Chizat and Bach [2018] associate overparametrization with better training properties, and Belkin et al. [2019, 2020] find it to enhance statistical generalization (see also [Loog et al., 2020] for historical context). For many architectures, overparametrization leads to the ability to fit any values $y_i$ at a given set of data points $\{x_1, \ldots, x_n\}$, and Cooper [2018] shows that generically, the set of weights for which the neural network interpolates prescribed values is a submanifold of high dimension and co-dimension in parameter space. Which solution on the manifold is dynamically chosen by an optimization algorithm, and which solutions have favorable generalization properties, is an active area of research in theoretical machine learning.

Current practice is to estimate a model's generalization to previously unseen data by assessing its performance on a hold-out set (a posteriori error estimate) or by using uniform estimates on the generalization of all elements of a function class (a priori estimate if the function class is encoded ahead of time by explicit regularization, a posteriori if membership is determined after optimization). Neither approach yields information on what a neural network does outside the support of the data distribution, a topic of great interest for the study of distributional shift and adversarial stability.

The main contribution of this work is split between two complementary lines of investigation:

1. We prove that neural network minimizers of regularized empirical risk functionals converge to minimum norm interpolants of a given target function in an infinite parameter limit (Section 2). Our result improves previous works to more general settings (See Remark 2.2).

2. Training a neural network generally corresponds to solving a non-convex minimization problem. While we provide convergence guarantees for empirical risk minimizers, in general there is no guarantee that a training algorithm finds a global minimizer of an empirical risk functional. Even if convergence holds, it is unclear *which* minimizer is selected (in the overparametrized regime, where the set of minimizers is a high-dimensional manifold). In settings where the minimum norm interpolant is known (Section 3), we compare numerical solutions to theoretical predictions to better understand (1) the predictive

power of theoretically studying empirical risk minimizers and (2) the implicit bias of different optimization algorithms (Sections 4 and 5). We believe this to be a useful benchmark problem for a better understanding of explicit regularization and implicit bias in optimization.

Minimum norm interpolants are the regression analogue to maximum margin classifiers. They are associated with favorable generalization properties and relative stability even against adversarial perturbations.

## 1.1 Previous work

**Minimum norm interpolation.**    In classes of linear functions, minimum norm interpolation has a long history as ridge regression (minimal $\ell^2$-norm) or as the least absolute shrinkage selection and operator (LASSO, minimal $\ell^1$-norm). To the best of our knowledge, minimum norm interpolation by neural networks has only been studied for shallow neural networks in one dimension by Hanin [2021], Debarre et al. [2022], Boursier and Flammarion [2023] and in odd dimension for certain radially symmetric data by Wojtowytsch [2022]. For classification problems, minimum norm/maximum margin classifiers were considered by E and Wojtowytsch [2022]. For finite datasets, the set of minimum norm interpolants was characterized by Parhi and Nowak [2021]. A parametrization of the same function class by neural networks with multiple linear layers and a single ReLU layer induces different concepts of minimum norm interpolation as studied by Ongie and Willett [2022].

**Implicit bias.**    The implicit bias of parameter optimization algorithms has been studied for gradient flows with infinitely wide, but shallow ReLU networks by Chizat et al. [2019], Jin and Montúfar [2020] and for stochastic gradient descent and diagonal linear networks by Pesme et al. [2021]. Chizat and Bach [2020] prove convergence to a maximum margin classifier for infinitely wide ReLU networks with one hidden layer if the parameters follow the gradient flow of an (unregularized) logistic loss risk functional. Many authors, including Damian et al. [2021], Li et al. [2021], Wu et al. [2022] and Wojtowytsch [2020], study the bias of SGD towards solutions at which the loss landscape is 'flat' in the parameter space. Hochreiter and Schmidhuber [1997] conjectured such minimizers to have favorable generalization properties. In many cases, minimizers tend to be flatter if the parameters associated to them are not excessively large. Yang et al. [2021] describe several phase-transitions in parameter space for the relation between flatness in parameter space and generalization. Zhou et al. [2020] compare the implicit bias of SGD and ADAM. Smith et al. [2021], Barrett and Dherin [2020] argue that gradient descent resembles a gradient flow in a modified (regularized) loss landscape more closely than the gradient flow of the original loss function. Hajjar and Chizat [2022] investigate the role of symmetries in optimization.

**Barron space.**    The Barron class is adapted to ReLU networks with a single hidden layer and weights of bounded average magnitude. Slightly different versions of the same function space have been studied by Bach [2017], E et al. [2019a,c], Ongie et al. [2019], E and Wojtowytsch [2020, 2021], Caragea et al. [2020], Parhi and Nowak [2021], Wojtowytsch [2022], Siegel and Xu [2022, 2023] under various names such as $\mathcal{F}_1$, Radon-BV or the variation space of the ReLU dictionary.

## 1.2 Preliminaries

**Conventions.**    $\mu$ always denotes a $\sigma^2$-sub-Gaussian probability measure on the data domain $\mathbb{R}^d$, i.e.

$$\exists\, C, \sigma > 0 \qquad \text{s.t.} \quad \mathbb{E}_{X \sim \mu}\big[\exp\left(\lambda\{\|X\| - \mathbb{E}\|X\|\}\right)\big] \leq C \exp\left(\frac{\lambda^2 \sigma^2}{2}\right) \qquad \forall\, \lambda > 0.$$

All norms are Frobenius norms ($\ell^2$-norm for vectors). For $n \in \mathbb{N}$, $S_n = \{x_{n,1}, \ldots, x_{n,n}\}$ is a set of $n$ iid samples from the distribution $\mu$, independent of $S_{n'}$ for $n' \neq n$. When unambiguous, we denote $x_{n,i} = x_i$. We take $\ell(f, y) = |f - y|^2$ as the mean squared error/$\ell^2$-loss function, but we remark that the theoretical analysis remains valid if $\ell_{MSE}$ is replaced by $\ell^1$-loss or a Huber or pseudo-Huber loss

$$\ell_{Hub}(f, y) = \begin{cases} |f - y|^2 & \text{if } |y - f| < 1 \\ 2|y - f| - 1 & \text{if } |y - f| \geq 1 \end{cases}, \qquad \ell_{pH}(y, h) = \sqrt{1 + |y - f|^2} - 1.$$

For $m \in \mathbb{N}$ and $(a, W, b) \in \mathbb{R}^m \times \mathbb{R}^{m \times d} \times \mathbb{R}^{m+1}$, let

$$f_{(a,W,b)} : \mathbb{R}^d \to \mathbb{R}, \quad f_{(a,W,b)}(x) = b_0 + \sum_{i=1}^{m} a_i\, \sigma(w_i \cdot x + b_i), \quad \sigma(z) = \text{ReLU}(z) = \max\{z, 0\},$$

i.e. $f_{(a,W,b)}$ is a ReLU network with a single hidden layer and weights $a, W$ and biases $b$. The vector $w_i \in \mathbb{R}^d$ is the $i$-th row of the matrix $W$. For $m, n \in \mathbb{N}$ and $\lambda \geq 0$, we denote the regularized empirical risk functional as $\widehat{\mathcal{R}}_{n,m,\lambda} : \mathbb{R}^m \times \mathbb{R}^{m \times d} \times \mathbb{R}^{m+1} \to [0, \infty)$:

$$\widehat{\mathcal{R}}_{n,m,\lambda}(a, W, b) = \frac{1}{2n} \sum_{i=1}^n \ell\left(f_{(a,W,b)}(x_i),\, y_i\right) + \frac{\lambda}{2}\left(\|a\|_2^2 + \|W\|_{Frob}^2\right).$$

**Concepts.** We introduce what we dub 'homogeneous Barron space' $\mathcal{B}$ heuristically here, and in greater detail in Appendix C. As a measure of magnitude of the function, we consider the weight decay (or Tikhonov) regularizer $\frac{1}{2}\left(\|a\|_2^2 + \|W\|_{Frob}^2\right)$ of the parametrized function $f_{(a,W,b)}$ which does not control the magnitude of the bias vector.

Note that the function class $f_{(a,W,b)}$ and its complexity do not change when we consider representations of the form $f_m(x) = \frac{1}{m}\sum_{i=1}^m a_i\,\sigma(w_i \cdot x + b_i)$ with a regularizer $\frac{1}{2m}\left(\|a\|_2^2 + \|W\|_{Frob}^2\right) = \frac{1}{2m}\sum_{i=1}^m \left(a_i^2 + \|w_i\|_2^2\right)$. A continuum analogue to these functions represented as an 'empirical average' over individual neurons is a general expectation representation $f_\pi(x) = \mathbb{E}_{(a,w,b)\sim\pi}\left[a\,\sigma(w^T x + b)\right]$ for some probability distribution $\pi$ on parameter space and a regularizer $\frac{1}{2}\mathbb{E}_{(a,w,b)\sim\pi}[a^2 + \|w\|_2^2]$. As the parametrization of a function by a neural network is generally non-unique, we define the Barron semi-norm $[f]_\mathcal{B} = \inf_{\{\pi : f_\pi \equiv f\}} \mathbb{E}_{(a,w,b)\sim\pi}[a^2 + \|w\|_2^2]$ as the lowest value attained by the regularizer over all possible parametrizations. For a more comprehensive understanding, interested readers may refer to Appendix C.

## 2 General convergence result

We first state a general convergence result to a minimum norm interpolant of given data generated by functions in the homogeneous Barron class $\mathcal{B}$.

**Theorem 2.1.** *Take $\mu, S_n, \widehat{\mathcal{R}}_{n,m,\lambda}$ as in Section 1.2, $f^* \in \mathcal{B}$, and let $y_i = f^*(x_i)$ for $i = 1, \ldots, n$. Assume that $m, \lambda$ are parameters which scale with $n$ as $m_n, \lambda_n$ such that*

$$\lim_{n\to\infty}\left(\lambda_n + \frac{1}{m_n}\right) = 0, \qquad \lim_{n\to\infty}\left(\frac{1}{\lambda_n m_n} + \frac{\log n}{\lambda_n \sqrt{n}}\right) = 0. \tag{1}$$

*Then almost surely over the random selection of data points in $S_n$, the following holds: If $(a, W, b)_n \in \operatorname{argmin}\widehat{\mathcal{R}}_{n,m_n,\lambda_n}$ for all $n \in \mathbb{N}$, then every subsequence of $f_n := f_{(a,W,b)_n}$ has a further subsequence which converges to some limit $\hat{f}^* \in \mathcal{B}$ with $\hat{f}^* = f^*$ $\mu$-almost everywhere and $[\hat{f}^*]_\mathcal{B} \leq [f^*]_\mathcal{B}$. Convergence holds in $L^p(\mu)$ for all $p < \infty$ and uniformly on compact subsets of $\mathbb{R}^d$. If $\mathbb{E}_\mu\|x\| + \sigma^2 \geq 1$, then for all $n \geq 2$, the following explicit bound holds up to higher order terms in $n$, $m = m_n$, $\lambda = \lambda_n$ with probability at least $1 - 1/n^2$:*

$$\|f_{(a,W,b)_n} - f^*\|_{L^2(\mu)}^2 \leq C\left(\frac{[f^*]_\mathcal{B}^2}{m}\,\mathbb{E}_\mu\left[\|x\|^2\right] + [f^*]_\mathcal{B}^2\left(\mathbb{E}_\mu\|x\| + \sigma^2\right)\frac{\log n}{\sqrt{n}} + \lambda\,[f^*]_\mathcal{B}\right). \tag{2}$$

**Remark 2.2.** *Theorem 2.1 extends previous results of E et al. [2019a] in several ways:*

1. *We allow for general sub-Gaussian rather than compactly supported data distributions.*

2. *We do not control the magnitude of the bias variables.*

3. *Our results apply to $\ell^2$-loss, which is neither globally Lipschitz-continuous nor bounded.*

4. *In a limiting regime, we characterize how the empirical risk minimizers interpolate in the region where no data is given by proving uniform convergence to a minimum norm interpolant.*

*In sum, the first three points necessitate a more careful technical analysis than in the settings of prior work, and the unboundedness of data and loss introduces additional logarithmic terms not present for E et al. [2019a]. To the best of our knowledge, the last point is novel and our work is the first to use the notion of $\Gamma$-convergence in this context.*

If the data-distribuion $\mu$ is supported on the entire space $\mathbb{R}^d$ (e.g. a non-degenerate Gaussian), then $\hat{f}^* \equiv f^*$. In many cases, however, $\mu$ is supported on a small, potentially compact and generally low-dimensional subset $M$ of the data space. In this case, the function $f^*$ is only known on the closed set $M \subsetneq \mathbb{R}^d$. As a result, there are many $f \in \mathcal{B}$ such that $f \equiv f^*$ on $M$ while $f \not\equiv f^*$ on $\mathbb{R}^d$ in general. The subsequential limit is one of these functions which has a minimal semi-norm $[f]_\mathcal{B}$.

Thus, beyond knowing that $f_n$ asymptotically fits the function $f^*$ perfectly at known data, Theorem 2.1 provides information about how it may interpolate at points where $\mu$ has no information. Such knowledge is of interest when a population may naturally evolve in time (distributional shift) of if $f_n$ is applied to a new problem with similar features but distinct geometry (transfer learning).

The proof of Theorem 2.1 is given in Appendix E. In the proof, we combine a direct approximation theorem to construct risk competitors with Rademacher complexity-based generalization bounds. Concentration inequalities are used to bound tail quantities. The scaling conditions that $\frac{\log n}{\sqrt{n}} \ll \lambda$ and $\frac{1}{m} \ll \lambda$ are used to ensure that the risk functional has sufficiently strong regularization. They can be weakened if the data measure $\mu$ is supported on a finite set of points or the function $f^*$ can be represented by a neural network with a finite number of neurons respectively. For instance, an analogous statement holds with a simpler proof for a fixed data-set $S$ of $n$ data points if $\lambda$ is coupled to $m$ such that $\frac{1}{m\lambda_m} \to 0$. In this case, we take the empirical distribution $\mu = \frac{1}{n}\sum_{x \in S} \delta_x$ as the population and do not need to bound the generalization gap. This model can be considered appropriate when $n \ll m$ (heavy overparametrization). A precise statement is given in Appendix F.

The proof remains valid if we only assume that

$$\limsup_{n\to\infty} \frac{1}{\lambda_n} \widehat{\mathcal{R}}_{n,m_n,\lambda_n}(a_n, W_n, b_n) = \inf \left\{ [f]_\mathcal{B} : f \equiv f^*, \quad \mu - \text{almost everywhere} \right\},$$

i.e. if $(a, W, b)_n$ parametrizes a function of low excess risk. In the proof, we obtain a more precise version of (2). Using an a priori Lipschitz-bound, it is possible to obtain a rate of convergence in $L^p(\mu)$ for $p < \infty$ by interpolation. For uniform convergence on compact sets outside the support of $\mu$, we do not obtain a rate in this work.

The proof of uniform convergence on compact sets utilizes the notion of $\Gamma$-convergence from the calculus of variations, a very stable notion of convergence which implies that minimizers of approximating functionals converge to the minimizer of a limiting functional. This notion has recently made inroads into machine learning applications and was used e.g. by Neumayer et al. [2023].

All results can easily be generalized to any more general function class which admits the three key ingredients: A bound on its Rademacher complexity, a compact embedding theorem, and a direct approximation theorem.

## 3 Minimum norm interpolants

### 3.1 One-dimensional example

In one dimension, [Wojtowytsch, 2022, Proposition 2.5] shows that $[f]_\mathcal{B} = \int_{-\infty}^{\infty} |f''(x)| \, dx$ for any smooth function $f : \mathbb{R} \to \mathbb{R}$ which satisfies $f'(x) = 0$ at some point $x \in \mathbb{R}$ – see also previous work by Li et al. [2020], E and Wojtowytsch [2020]. This one-dimensional case is, in fact, the simplest case of a general characterization of Barron functions in any dimension by Ongie et al. [2019].

Consider the task of minimizing $[f]_\mathcal{B}$ under the condition that $f(x) = |x|$ if $|x| \geq 1$. Then the minimum is attained for any smooth convex function $f$ which satisfies the constraint since

$$2 = f'(1) - f'(-1) = \int_{-1}^{1} f''(x) \, dx \leq \int_{-1}^{1} |f''(x)| \, dx = [f]_\mathcal{B}$$

with equality if and only if $f'' \geq 0$. The same estimate holds for non-smooth Barron functions if the second derivative is interpreted as a Radon measure. For piecewise linear functions, this corresponds to summing $|f'(x_i^+) - f'(x_i^-)|$ over the non-smooth points $x_i$. More generally, the set of minimum norm interpolants of one-dimensional convex data was characterized by Savarese et al. [2019].

**Proposition 3.1.** *Let $x_0 < \cdots < x_n$ and $y_i = f^*(x_i)$ for a convex function $f^*$ and $i = 0, \ldots, n$. If $y_1 < y_0$ and $y_n > y_{n-1}$, then $f$ is a minimum Barron norm interpolant of the dataset $\{(x_i, y_i)\}_{i=0}^n$*

*if and only if $f$ is convex, $f(x_i) = y_i$ for all $i = 0, \dots, n$ and*

$$f'(x) = \frac{y_1 - y_0}{x_1 - x_0} \quad \text{for } x < x_1 \qquad \text{and} \qquad f'(x) = \frac{y_n - y_{n-1}}{x_n - x_{n-1}} \quad \text{for } x > x_{n-1}.$$

The two given slopes are the largest values that are required for derivatives at any point. We give a proof of Proposition 3.1 in Appendix G. In full generality, minimum norm solutions have been characterized by Hanin [2021] using matching convexities to achieve minimal total curvature.

In a recent article, Boursier and Flammarion [2023] show that if the full Barron norm is controlled, i.e. if the magnitude of biases is included in the regularizer, a specific convex function is selected. This corresponds to minimizing a functional

$$\int_{-1}^{1} |f''(x)| \sqrt{1 + x^2} \, \mathrm{d}x \qquad \text{rather than} \qquad \int_{-1}^{1} |f''(x)| \, \mathrm{d}x.$$

The first functional prefers $f''$ to be large close to the origin, if it has to be large anywhere. All break points occur as close to the origin as possible, in particular: Either at the origin or at data points. Unlike the Barron semi-norm penalty studied in this work, which does not select a specific minimum norm interpolant, the Barron norm penalty thus induces a *sparse* neural network interpolant.

### 3.2 Radially symmetric bump function

Another setting where we have explicit minimum norm interpolant is when we fit a bump function for radially symmetric data. Recall a result of Wojtowytsch [2022] on minimum norm fitting of certain radially symmetric data.

**Proposition 3.2.** *[Wojtowytsch, 2022, Theorem 3.1] Let $d \geq 3$ be an odd integer and*

$$\mathcal{F} = \left\{ f \in C_c(\mathbb{R}^d) : f(0) = 1 \text{ and } f(x) = 0 \text{ if } \|x\| \geq 1 \right\}.$$

*Then there exists a unique radially symmetric function $f_d^* : \mathbb{R}^d \to \mathbb{R}$ such that $f_d^* \in \operatorname{argmin}_{f \in \mathcal{F}}[f]_\mathcal{B}$. The norm of minimizers grows as $\lim_{d \to \infty} [f_d^*]_\mathcal{B}/d \approx 3.7$.*

We note that the existence of minimum norm interpolants which are not radially symmetric is not excluded, but if $\hat{f} \in \operatorname{argmin}_{f \in \mathcal{F}}[f]_\mathcal{B}$ is any other minimum norm interpolant, then its radial average $\operatorname{Av} \hat{f}$ coincides with $f_d^*$: $\operatorname{Av} \hat{f} \equiv f_d^*$. Here

$$\operatorname{Av} f(x) := \int_{SO(d)} f(Ox) \, \mathrm{d}H_O = \int_{S^{d-1}} f\left(|x| \cdot \nu\right) \, \mathrm{d}\pi_\nu^0 \tag{3}$$

where $H$ is the uniform distribution (Haar measure) on the group of rotations and $\pi^0$ is the uniform distribution on the $d-1$-dimensional sphere in $\mathbb{R}^d$. In [Wojtowytsch, 2022, Section 6], an algorithm is given to find the minimum norm interpolant $f_d^*$ by numerically solving a one-dimensional polynomial approximation problem and a linear system of moment conditions.

The uniqueness statement allows us to strengthen the result of Theorem 2.1 in this case. A natural setting is to use a sub-Gaussian data distribution $\mu$ which gives positive mass to the origin, but has no mass elsewhere in the unit ball. It should have mass everywhere outside the unit ball. Under this natural setting, we have a stronger result of Theorem 2.1.

**Corollary 3.3.** *Take $\mu, S_n, \widehat{\mathcal{R}}_{n,m,\lambda}$ as in Section 1.2, and assume in addition that*

1. *$\mu(\{0\}) > 0$ (positive mass at the origin).*

2. *$\mu(B_1(0) \setminus \{0\}) = 0$ (no mass elsewhere in the unit ball).*

3. *$\mu(U) > 0$ for any open set $U \subseteq \mathbb{R}^d \setminus \overline{B_1(0)}$ (mass everywhere outside the unit ball).*

*Assume that $m, \lambda$ scale with $n$ as in (1). Almost surely over the random selection of data points in $S_n$, the following holds: If $(a, W, b)_n \in \operatorname{argmin} \widehat{\mathcal{R}}_{n,m_n,\lambda_n}$ for all $n \in \mathbb{N}$, then sequence of radial averages $\operatorname{Av} f_n$ of $f_n := f_{(a,W,b)_n}$ converges to $f_d^*$ as in Proposition 3.2. Convergence holds in $L^2(\mu)$ for MSE loss (with an explicit rate) and uniformly on compact subsets of $\mathbb{R}^d$ (without a rate in this work).*

In Corollary 3.3, we guarantee convergence to the unique radial minimum norm interpolant $f_d^*$ (at least for the radial average), while in Theorem 2.1 we may have different subsequences that converge to different minimum norm interpolants. The proof is given in Appendix E.

## 4    Relating Interpolation, Optimization and Generalization

For a given bounded set $K \subseteq \mathbb{R}^d$, Theorem 2.1 states that for a large number of neurons $m \in \mathbb{N}$, a large number of data points $n \in \mathbb{N}$, and a small penalty $\lambda > 0$, minimizers of the empirical risk functional $\widehat{\mathcal{R}}_{n,m,\lambda}$ resemble a minimum norm interpolant everywhere in $K$. As the Barron semi-norm controls the generalization gap (see Appendix D) and a minimum norm interpolant has minimal Barron norm by definition, this suggests that minimum norm interpolants are optimal in terms of generalization, at least when arguing from this upper bound.

Many authors, including Safran and Shamir [2018b], Venturi et al. [2018], demonstrate that neural network training is a non-convex optimization problem. As such, it is not guaranteed that numerical optimizers (1) converge to interpolants at all, and (2) select minimum norm interpolants out of the large set of different neural networks which interpolate given data, even when a regularizer is included in the training loss functional.

On the other hand, there are settings where an optimization algorithm selects a minimum norm solution even without explicit regularization. This is easily proved for gradient descent on the overparametrized least squares regression problem $\mathcal{R}_n(a) = \frac{1}{n} \sum_{i=1}^{n} |a^T x_i - y_i|^2$ with initial condition $a = 0$ and $n < m = d$. Using entirely different methods, Chizat and Bach [2020] prove a similar result for binary classification by shallow neural networks with logistic loss. For regression problems using neural networks, analogous results are not available to the best of our knowledge. This in part motivates the following numerical investigation. Namely, we are interested in exploring the effects of explicit regularization and the implicit bias of optimization algorithms toward minimum norm interpolants. Knowing the analytically optimal solution in between given data provides us the opportunity to compare optimizers on a deeper level than merely testing their performance on unseen data generated from the same distribution.

As seen in Figure 1, the radial profile $r \mapsto f_d^*(re_1)$ of Wojtowytsch [2022]'s minimum norm interpolant $f_d^*$ is so close to 0 on $[r_d, \infty)$ as to be virtually indistinguishable from zero numerically for some $r_d < 1$ which decreases in $d$. Indeed, the first $(d-1)/2$ derivatives of vanish at $r = 1$ due to [Wojtowytsch, 2022, Lemma 4.1] and $0 \leq f_d^*(x) \leq Cd^{3/2}((1 - \|x\|^2)/\|x\|)^{(d-3)/2)}$ due to [Wojtowytsch, 2022, Appendix D.1] for a universal constant $C > 0$. In particular, if $d$ is large, $\|f_d^*\|_{L^\infty(\mathbb{R}^d \setminus B_{r_d}(0))}$ is negligible compared to the approximation error $d^2/m$ for any reasonable dimension $d$ and network width $m$. Consequently, the rescaled function $h_d^*(x) := f_d^*(r_d x)$ meets the constraint $h_d^* \equiv 0$ outside $B_1(0)$ almost exactly and has the smaller Barron semi-norm $[h_d^*]_{\mathcal{B}} = r_d [f_d^*]_{\mathcal{B}}$. For this reason, we compare numerical solutions to the interpolation problem in Corollary 3.3 to rescaled versions of $f_d^*$ rather than $f_d^*$ itself, at least in high dimension. For $r$ below the threshold value $r_d$, there is no noticable trade-off between rescaling $f_d^*(rx)$ and data-fitting. For larger values of $r$, the Barron semi-norm is reduced more significantly, but the data fit becomes appreciably worse.

## 5    Numerical Experiments

Our main goal in this section is to gain a more precise understanding of different optimization algorithms by comparing numerical solutions to a known minimum norm interpolant. We consider the two settings in which minimum norm interpolation by Barron functions is best understood: One-dimensional and radially symmetric functions. As a benefit, we can easily visualize the numerical results in both settings. We focus on three questions of interest.

1. **Explicit regularization.** If $\lambda > 0$ is moderately small and $m, n$ are large, then a global minimizer of $\widehat{\mathcal{R}}_{n,m,\lambda}$ resembles a minimum norm interpolant between known data points due to Theorem 2.1. Is the minimizer which we find numerically close to to a minimum norm interpolant, or does it merely fit the function at known data points?

2. **Implicit bias.** If $m, n$ are large, does a training algorithm select a minimum norm interpolant out of potentially many possible solutions without explicit regularization (i.e. for $\lambda = 0$)?

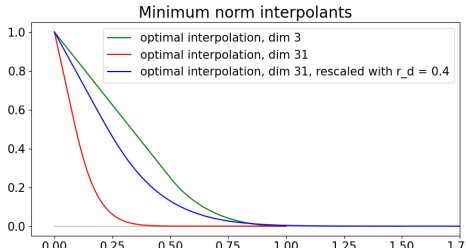
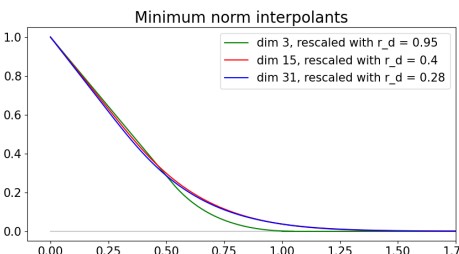

Figure 1: **Left:** In Dimension $d = 31$, the minimum Barron norm solution $f_d^*$ satisfies $f_d^* \equiv 0$ on $\mathbb{R}^d \setminus B_{r_d}(0)$ for $r_d = 0.4$ to high precision, albeit not exactly. The rescaled function $f_d^*(r_d x)$ is a suitable candidate for a minimum norm almost-interpolant to high accuracy. **Right:** For later use, we consider more aggressively rescaled functions $f_d^*(r_d x)$ for $d = 15, d = 31$ with lower semi-norm, but worse data fitting properties. We note that the rescalings of these functions which have essentially the same slope as $f_3^*$ at $r = 0$ appear to coincide. We conjecture that this statement allows for a more rigorous formulation.

3. **Learning symmetries:** The optimal minimum norm interpolant $f_d^*$ described in Proposition 3.2 is radially symmetric and satisfies $0 \leq f_d^* \leq 1$. Proposition 3.2 does not rule out the existence of other minimum norm interpolants which are not radially symmetric. Does an optimization algorithm generally find solutions which are (approximately) radially symmetric and confined to the interval $[0, 1]$? A similar consideration applies in a one-dimensional investigation with reflection symmetry.

The third question is of particular interest for algorithms like ADAM, which operate coordinate-by-coordinate and do not respect Euclidean isometries. By comparison, we expect that SGD, initialized at a radially symmetric configuration, preserves Euclidean isometries. More experiments in similar settings can be found in Appendix A.

## 5.1 One-dimensional experiments

We consider the classical interpolation problem of numerical analysis: Fit values $f^*(x_i) \in \mathbb{R}$ at points $x_i \in \mathbb{R}$ for $i \in \{1, \dots, n\}$. In contrast to classical numerical analysis, we consider overparametrized ReLU-networks with a single hidden layer as our model class. As in Section 3, we select $f^*(x) = |x|$ for simplicity.

In Figure 2, a ReLU network with a single hidden layer of width $m = 200$ was trained to fit $f^*(x) = |x|$ at a symmetric set containing 15 equi-spaced points in $(1, 2)$. Optimizers included SGD (with learning rate $\eta = 5 \cdot 10^{-5}$ and momentum $\mu = 0.99$), SGD ($\eta = 10^{-2}$, $\mu = 0$), ADAM ($\eta = 5 \cdot 10^{-5}$ and default parameters) and the quasi-Newton L-BFGS method. Deterministic gradients based on the $n = 30$ sample points were used. The final training loss was below $10^{-4}$ on average. The network weights were initialized by a scaled uniform Xavier initialization, i.e. uniformly in a symmetric interval of length $2\alpha \cdot \sqrt{6/(n_{in} + n_{out})}$ where $n_{in}$ and $n_{out}$ denote the number of input- and output-units to a layer respectively. The 'gain' factor was selected as $\alpha \in \{0.5, 1, 5\}$. Without weight decay and for small gain, the optimizers find a solution close to the smallest possible minimum norm interpolant $f(x) = |x|$. The larger the parameters for initialization gain and weight decay penalty, the closer numerical solutions are to the largest possible minimum norm interpolant $f(x) = \max\{|x|, 1\}$.

We observe that a higher gain factor $\alpha$ corresponds to faster initial training, but a high gain like $\alpha = 5$ produces interpolants which are non-convex without regularization, while a lower gain factor produces convex interpolants in longer time. This observation agrees with the findings of Chizat et al. [2019], who dub the large $\alpha$ setting the 'lazy training' regime and associate it with worse generalization performance. As Pesme et al. [2021] eloquently put it: "there is a tension between generalisation and optimisation: a longer training time might improve generalisation but comes at the cost of... a longer training time."

If $m$ is large and $\alpha$ is not too big, the variation of solutions produced by a training algorithm vary less over different stochastic realizations – see Appendix A for experiments for $m = 1,000$. The

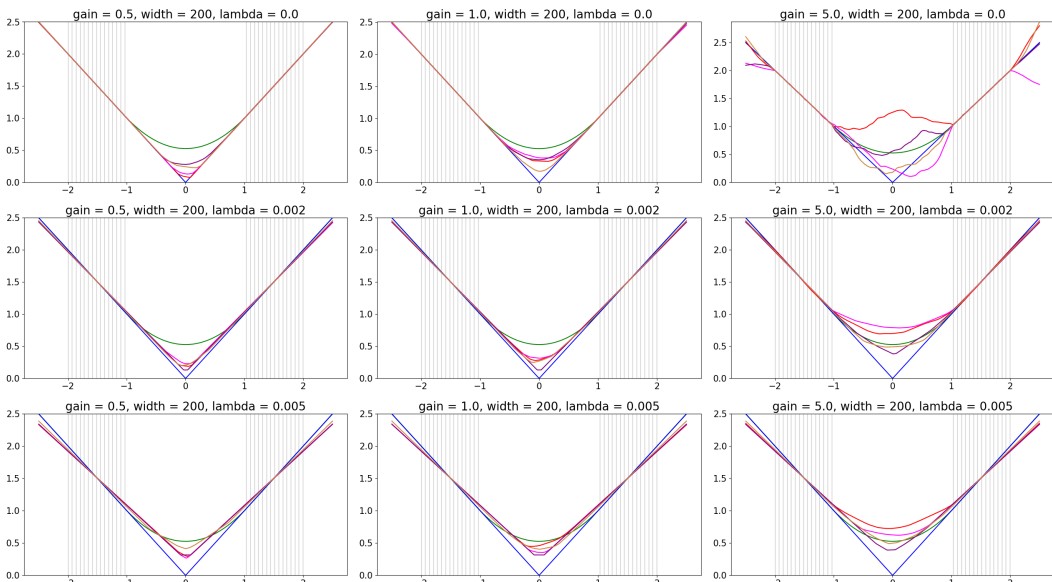

Figure 2: We compare numerical approximations of a target function for Momentum-GD (red), GD (magenta), ADAM (purple) and L-BFGS (brown). The target function is drawn in blue and the natural cubic spline in green. For each algorithm, we plot one representative solution to study symmetry selection properties. Vertical grey lines indicate known training data points. The initialization has gain $\alpha \in \{0.5, 1.0, 5.0\}$ in the left, middle and right column. The weight decay penalty is $\lambda \in \{0, 0.002, 0.005\}$ (top, middle, bottom row). For all optimization algorithms, the final loss is approximately $0, 2 \cdot 10^{-4}$ and $10^{-3}$ respectively.

For small gain, the optimizer selects a minimum norm interpolant/convex function. For large gain, the interpolant is often non-convex for any one of the optimization algorithms, unless the weight decay is given a positive value. A specific type of minimum norm interpolant in the large set of possible solutions seems to be selected by specifying optimization algorithm, weight decay and initialization. Evidently, initialization and weight decay have far greater influence than the choice of the optimizer. Small gain seems to preference the *sparsest* solution $f(x) = |x|$ has predicted with bias penalty by Boursier and Flammarion [2023]. With larger weight decay, the solutions become more convex and more symmetric, but the accuracy of data fitting decreases. Visually, the second order L-BFGS method appears to be the least affected by different choices in initialization.

dynamics are close to those of a limiting 'mean field' model studied by Chizat and Bach [2018], Rotskoff and Vanden-Eijnden [2018], Mei et al. [2018], Sirignano and Spiliopoulos [2020a] and Wojtowytsch [2020]. In these works, the limiting model is typically derived with a factor $1/m$ outside the function definition, which is implicit in the initialization here since $n_{in} + n_{out} \approx m$ for both layers and the ReLU activation is positively one-homogeneous. Global convergence to a minimizer (but not necessarily a minimum norm solution) is guaranteed (up to certain technical assumptions) by Chizat and Bach [2018] and Wojtowytsch [2020].

For comparison, we also present the natural cubic spline interpolant, i.e. the function $f$ which minimizes the stronger curvature energy $\int_{-2}^{2} |f''(x)|^2 \, dx$ under the condition that $f(x_i) = |x_i|$ for all $i = 1, \dots, n$. Unlike the minimum Barron norm interpolants, the natural cubic spline may not be convex (and in fact, it is not if $f^*$ is replaced by $h^*(x) = |x - 0.5|$).

## 5.2 Radially symmetric data

We explore the performance of numerical optimization algorithms in the setting of Corollary 3.3 with and without explicit regularization $\lambda \in \{0, 10^{-5}\}$ in dimensions $d = 3$, $d = 15$ and $d = 31$. The numerical solution is then compared to (a rescaled version of) the analytic minimum norm interpolant $f_d^*$ described in Proposition 3.2, which we compute by the algorithm described in [Wojtowytsch, 2022, Section 6]. The rescaling factor $r_d$ is chosen heuristically for an accurate match.

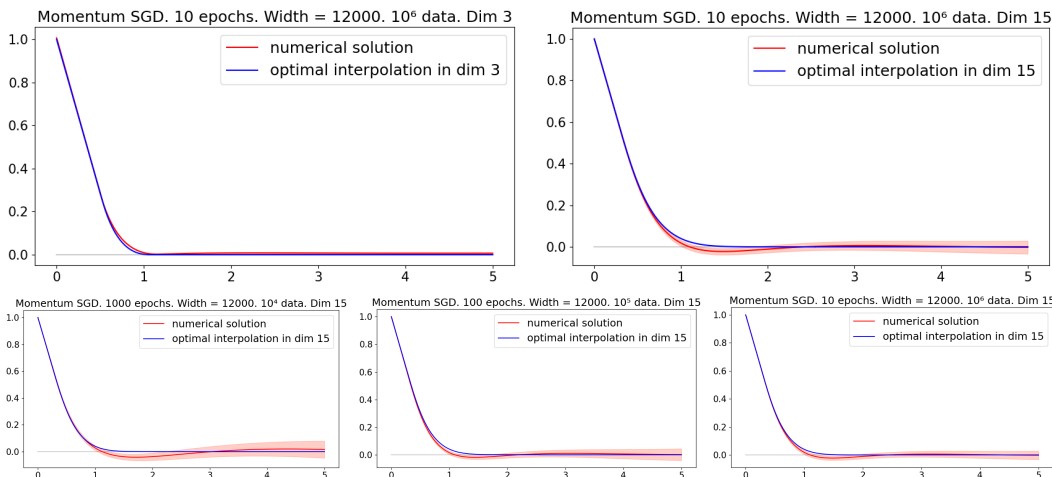

Figure 3: A neural network with a single hidden layer of width $m = 12,000$ was trained by gradient descent with learning rate $\eta = 10^{-3}$ and momentum $\mu = 0.99$ in the setting of Section 5.2. The radial average is sketched by a solid red line. One radial standard deviation around the average, computed over 500 random directions, is shaded.z **Top row:** Experiment in dimension 3 (left) and dimension 15 (right). The numerical solutions are compared to $f_d^*(r_d x)$ with $r_3 = 1/1.05$ and $r_{15} = 1/2.55$. In both cases, the 'minimum norm interpolant' shape is attained to high accuracy. Both solutions are approximately symmetric, more so in low dimension. **Bottom row:** Numerical approximations to $f_{15}^*(r_{15} \cdot)$ for neural networks of constant width $m = 12,000$, trained on data sets of different size (but for an identical number of $200,000$ training steps with stochastic estimates computed over a batch of 50 data points). The shape of the radial average is comparable across different dataset sizes, but the fit of the radial average with data is improved and the radial variance reduced for larger datasets. Note that the first two simulations are set in the overparametrized regime, whereas the last experiment on the largest dataset is underparametrized.

Data is generated from a distribution $\mu = \mu_1 + \mu_2 + \mu_3$ where $\mu_1$ is a point mass of magnitude $m_1$ at the origin, $\mu_2$ is a uniform measure on the unit sphere $S^{d-1}$ with mass $m_2$ and $\mu_3$ is the radially symmetric measure of mass $1 - m_1 - m_2$ such that $\|x\|_2$ is distributed uniformly in $[1,7]$. We numerically explored various values for $m_1 \in [0.1, 0.4]$ and $m_2 \in [0.0, 0.4]$ and found simulations to be relatively stable under a number of choices.

Results are presented in Figures 3, 4 and Appendix A. We find that all algorithms find a solution with radial average similar to $f_d^*(r_d x)$, albeit for rescaling factors $r_d$ which depend on dimension $d$ and (to a lesser extent) the optimizer. In high dimension, solutions are not perfectly radially symmetric, but the larger amount of variation over a sphere of fixed radius is observed in the domain where $f_d^* \approx 0$ rather than in the transition area $(0, r_d)$. Larger datasets improve the compliance with the optimal interface and reduce the radial standard deviation. Solutions do not remain non-negative and drop below zero before leveling off as the radius increases. The drop becomes more noticeable as the dimension increases and less pronounced for wider networks.

The results are essentially identical for normal Xavier initialization and (not radially symmetric) uniform Xavier initialization. In accordance with our expectations, the radial standard deviation is higher for Adam compared to optimizers based in Euclidean geometry. While the neural network function found by Adam resembles a minimum norm interpolant, the weight decay regularizer takes significantly higher values compared to other optimization algorithms.

# 6 Conclusion

Shallow ReLU networks converge to minimum norm interpolants of given data: Provably if explicit regularization is included and empirically if it is not. We conclude with a summary of our empirical insight into the implicit bias of neural network optimizers.

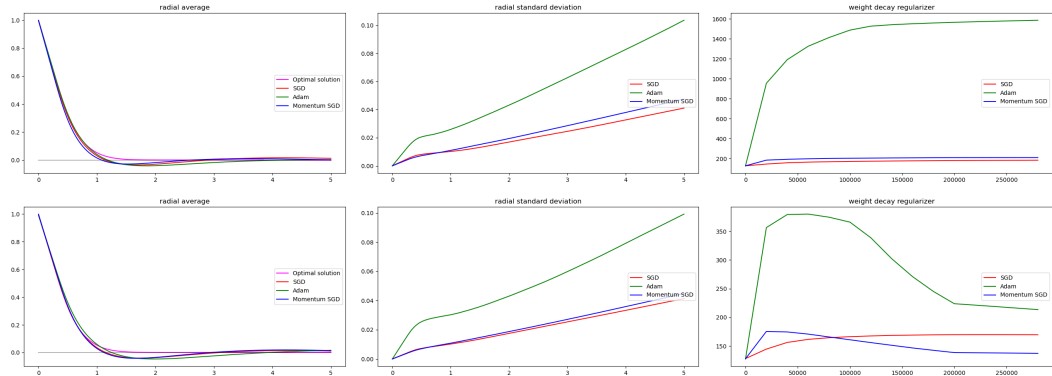

Figure 4: **Top row:** We compare different optimizers (SGD = red, Adam = green and Momentum-SGD = blue) in the setting of Section 5.2 in dimension 31 with $n = 10^4$ data points. All minimizers attain the minimum norm interpolant shape (left column, rescaled optimal solution in pink) – curiously for Adam, the correct shape is attained despite the fact that at $\approx 1,600$ the weight decay regularizer is an order of magnitude higher than for the other optimizers (right column). The high regularizer value goes hand in hand with a higher radial standard deviation (middle column). The initialization is uniform Xavier (in particular, not radially symmetric). Essentially identical results are observed for the radially symmetric normal Xavier initialization with the same degree of radial symmetry in Figure 8. **Bottom row:** If we include an explicit weight-decay regularizer with weight $\lambda = 10^{-5}$, solutions resemble an optimal interpolant for a smaller rescaling factor $r_{31,\lambda} = 1/3.8$ compared to $r_{31} = 1/3.5$ without regularization. This is expected since the norm is weighted more heavily compared to data compliance. Notably, the radial standard deviation does not decrease, even for the (heavily affected) ADAM optimizer. It remains noticeable at $0.1$ for $r = 5$ and Adam.

1. With reasonable (not too large) initialization, all algorithms studied here are biased towards minimum norm interpolant profiles.

2. At least in the case of Adam, this bias is visible on the function level, but not on the parameter level, as the weight decay regularizer increases rapidly to large magnitude. Despite this, ADAM solutions often appear 'flatter' in high dimension with a lower rescaling factor $r_d$.

3. Explicit regularization stabilizes towards a minimum norm interpolant shape, but at the cost of a decreased fit with the target values. Its impact is most significant for poorly chosen initial conditions.

4. When the minimum norm interpolant is non-unique, different types of minimum norm interpolants are found depending on the choice of initialization scheme and optimization algorithm. The impact of initialization scale appears more significant.

5. Optimization algorithms which are rooted in Euclidean geometry (such as SGD and momentum-SGD) more successfully preserve Euclidean symmetries compared to the 'coordinate-wise' Adam algorithm.

The last observation is not surprising for radially symmetric initialization laws as radially symmetric parameter distributions induce radially symmetric functions. It is, however, observed also for a uniform initialization scheme which only obeys coordinate symmetries.

We believe minimum norm interpolation to be a useful testbed to study the implicit bias of optimizers and the impact of initialization and regularization. While minimum norm interpolation by deeper networks has not been characterized yet, we anticipate no obstructions to implementing a similar program there in the future.

## Acknowledgements

The research of Jiyoung Park is supported by NSF DMS-2210689.

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

# Appendix

## A Numerical Experiments

### A.1 Hyperparameter settings and computation effort

In all experiments in Dimensions 3 and 15, the following hyperparameter settings were used unless otherwise indicated:

1. Normal Xavier initialization with gain $\alpha = \sqrt{2}$
2. SGD: Learning rate = $10^{-2}$ (Dimension 15), $10^{-3}$ (Dimension 3).
3. Momentum-SGD: Learning rate = $10^{-3}$ and momentum $\mu = 0.99$
4. ADAM: Learning rate = $10^{-3}$ and PyTorch default hyperparameters for $\beta_1 = 0.9, \beta_2 = 0.999, \varepsilon = 10^{-8}$.

For experiments in Dimension 31, we drop the learning rate for ADAM after 50 of 150 epochs by a factor of 10 and for Momentum-SGD by a factor of 10 after 100 epochs.

In Dimension 3, a learning rate of $10^{-2}$ was found numerically unstable for SGD without momentum. To compensate for the smaller learning rate and provide a fair comparison, the number of time steps was increased.

All experiments were performed on a free version of google colab or the experimenters' personal computers. One run of the model takes below fifteen minutes on a single graphics processing unit.

### A.2 Summary and interpretation of additional simulations

In this Section, we present additional numerical experiments in various situations complementary to those presented in the main body of the text. These include: Wider neural networks (Appendix A.3), experiments with different optimizers (Appendix A.4), experiments with different initialization to explore effects of scale and symmetry and the role of explicit regularization (Appendices A.8 and A.5), experiments with $\ell^1$-loss instead of $\ell^2$-loss (Appendix A.7) and repeated experiments to visualize the stochastic variation between runs (Appendix A.6).

Additionally, we present and investigation into related settings where our theoretical understanding does not apply: In Appendix A.10, we consider linearized (random feature) dynamics to explore how close we are to a (truly non-linear) neural network model. In Appendix A.11, we consider neural networks with a single hidden layer and leaky ReLU activation instead of ReLU activation. In Appendix A.12, we consider ReLU networks with multiple hidden layers. For a detailed list, see the table of contents below.

Our goal is not to explore questions of loss function, initialization, optimization algorithm and the impact of hyperparameters in a systematic fashion, but rather to establish problems in which a minimum norm interpolant can be found in an explicit fashion as instructive benchmarks to numerically study such questions. As a proof of concept, we provide a partial exploration of the parameter space. For the moment, we find ourselves confined to ReLU networks with a single hidden layer, as this is the only case in which explicit minimum norm interpolants are available. Minimum norm interpolation describes the shape of functions between known data clusters and is thus more expressive than a study of generalization error which is naturally confined to data clusters.

The additional experiments corroborate our findings in the main text. Before the detailed presentation, let us briefly summarize the conclusions.

1. Across a variety of different optimizers, Xavier type (= Glorot type) initialization schemes and loss functions, a minimum Barron norm interpolant-like shape was attained, to varying degrees of accuracy and with different rescaling factors.

2. Solutions are fairly radially symmetric with standard deviation in radial direction at most 0.1 (SGD) and 0.2 (Adam).

3. A geometrically distinct shape is observed for random feature models in the same regime.

4. Explicit regularization has little effect, even for poorly chosen initializations of Xavier type with high gain. We conjecture that the uniqueness of the radially symmetric minimum norm interpolant induces a higher degree of rigidity and bias, compared to the one-dimensional case where the set of minimum norm solutions was diverse (and infinite).

5. Functions display larger variation in the radial direction for He initialization. In this regime, explicit regularization has more apparent and beneficial effects on both solution shape and radial symmetry. The solution does not reduce to the random feature model in this case either.

## A.3   Wide neural networks in one dimension

In Figure 5, we present the same experiment as in Figure 2 for wider neural networks with $m = 1,000$ neurons in the hidden layer.

## A.4   SGD and ADAM: Dimensions 3 and 15

We repeat the experiment on Figure 3 for Stochastic Gradient Descent (SGD) optimizer without momentum and for the Adam optimizer of Kingma and Ba [2014]. The results are displayed in Figure 6 and 7 respectively. The results strongly resemble those obtained for the SGD optimizer with momentum in Figure 3.

## A.5   Radial symmetry in Dimension 31

As indicated in Figure 4, we present computational results with the radially symmetric normal Xavier initialization in Figure 8.

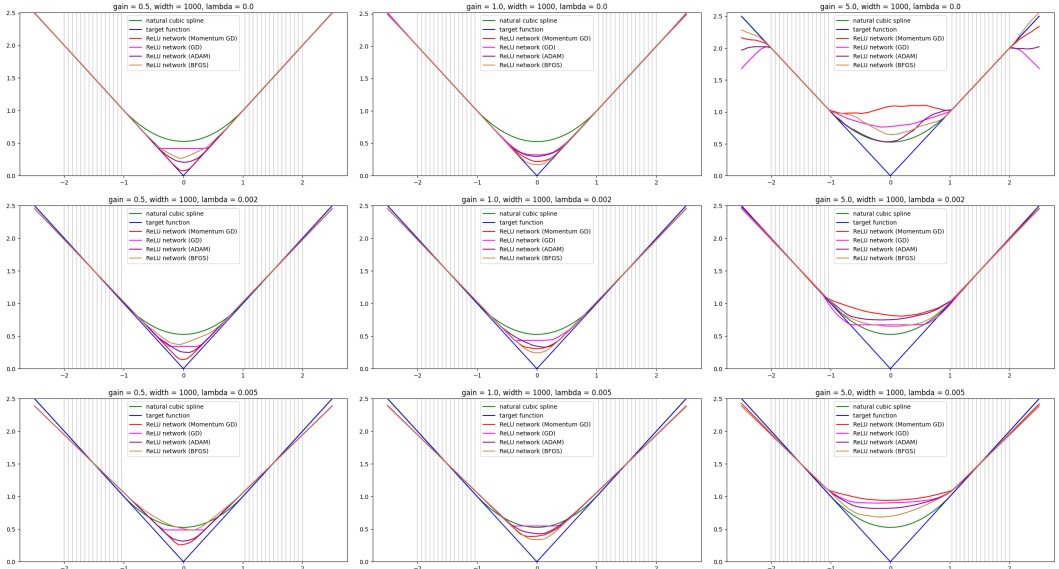

Figure 5: For wider neural networks, we observe less stochastic variation between runs, as the empirical distribution of neurons is closer to a continuum limit. Solutions are generally more convex and symmetric than their narrow counterparts. The gradient descent optimizer without momentum stands out for its tendency to select solutions with highly localized second derivatives and a preference for piecewise linear functions with few linear regions, while other training algorithms select 'smoother' solutions with curvatures which are dispersed more evenly throughout the domain.

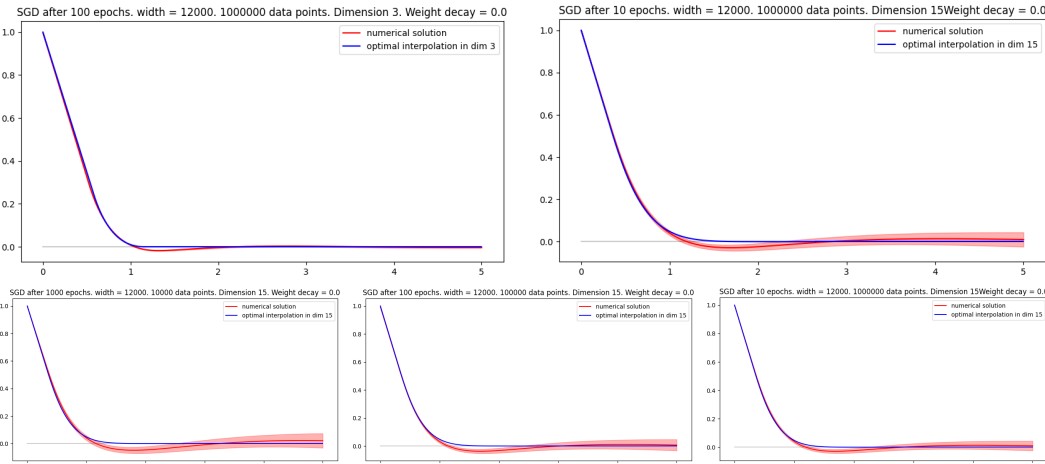

Figure 6: We perform the same experiments as for Figure 3, but with the SGD without momentum optimizer. Learning rate was adjusted to $10^{-2}$ for dimension 15, but for dimension 3 we just used $10^{-3}$ learning rate and ran 10 times more epochs, due to stability of neural network training in dimension 3 case. The results are comparable, but the rescaling factors were chosen as $1/1.15$ in dimension 3 and $1/2.65$ in dimension 15.

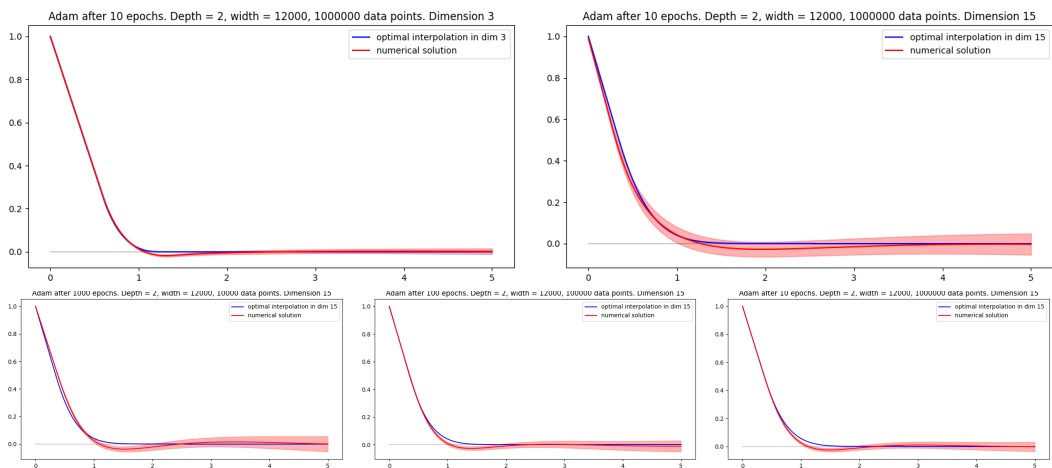

Figure 7: We repeat the experiments of Figure 3, but with Adam. The numerical solutions resemble those found by Momentum-SGD, but better rescaling factors for numerical solutions are $1/1.2$ in dimension 3 and $1/2.75$ rather than $1/1.05$ and $1/2.55$, i.e. the functions are 'flatter'.

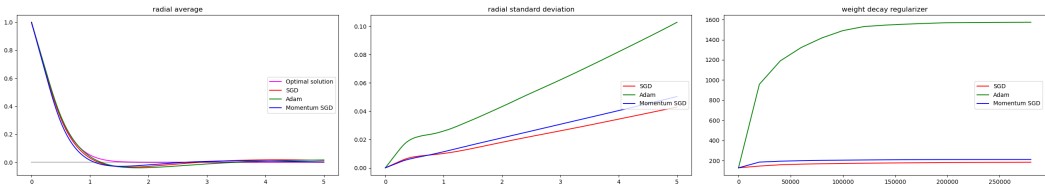

Figure 8: Results between normal and uniform Xavier initialization are essentially identical in this experiment – compare Figure 4. (Approximate) radial symmetry is attained even when parameters are initialized in a fashion which is not radially symmetric.

## A.6 Gradient descent with Momentum

In Figure 9, we present additional runs in the setting of Figure 3. Despite quantitative variation, the geometric shapes of solutions are stable over multiple runs and resemble the minimum norm interpolant $f_{15}^*$ in all cases.

## A.7 $\ell^1$-loss and Huber loss

We repeat the experiment of Figure 3 with the $\ell^1$-loss function in the place of $\ell^2$-loss. To compensate for the lack of smoothness in the loss function, we reduce the learning rate by a factor of $10$ to $10^{-4}$ and increase the number of epochs by 50% to compensate. Results are reported in Figure 10.

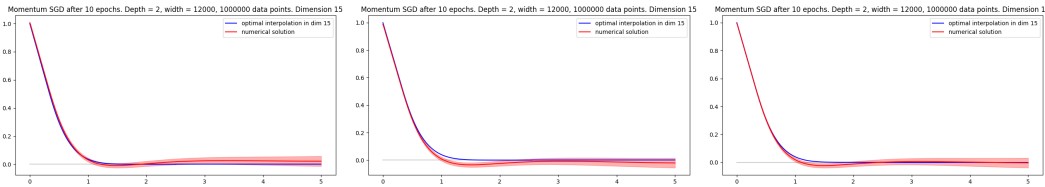

Figure 9: Three realizations of numerical interpolations in the setting of Section 5.2, computed by SGD with learning rate $\eta = 10^{-3}$ and momentum $\mu = 0.99$. In all cases, the minimum norm interpolant shape is attained approximately, but the radial averages briefly dip below zero and exhibit a local minimum which is not found in $f_d^*$. The variation between runs is notable, but not large.

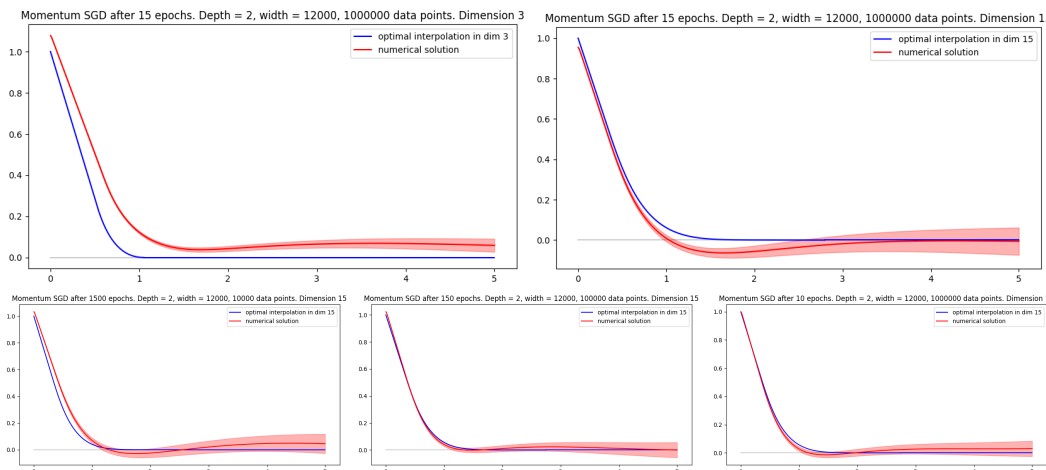

Figure 10: $\ell^1$-loss leads to similar numerical results with a smaller rescaling factor of $1/2.8$ rather than $1/2.5$ for $\ell^2$-loss. Curiously, $f(0) > 1$ for these algorithms, while $f(0) \le 1$ when optimizing $\ell^2$-loss. In Dimension $d = 3$, the non-smoothness leads to a minimization problem that is not well resolved by the numerical optimization algorithm.

Similarly, we repeated experiments with the Huber loss function. Since $|f(x_i) - f^*(x_i)| < 1$ over the data set during the final stages of training, the loss function coincides with $\ell^2$-loss in the long run and all experiments are identical to $\ell^2$-loss. We therefore do not present additional plots.

In the initial stages of training, Huber loss is more stable numerically than $\ell^2$-loss, especially for large gain or He initialization (compare Section A.9).

## A.8 Initialization scaling and explicit regularization: high-dimensional radial data

As noted in Section 5.1 and previously by Chizat et al. [2019], the choice of initialization affects the optimization process of neural networks. Motivated by our observations in the one-dimensional case, we consider the effects of initialization and explicit regularization in the radially symmetric setting (Section 5.2). Our results support the earlier claim that the effects of regularization are advantageous for poorly chosen initialization with high gain.

The experiments were performed in higher dimension 31 and with *uniform* Xavier initialization for the scenario in which it is most challenging to obtain radially symmetric solutions. As in Appendix A.7, we used $\ell^1$-loss rather than $\ell^2$-loss. To compensate for the non-smoothness of the loss function, we drop the learning rate by a factor of 10 twice during the training process. Similar results were observed for $\ell^2$-loss, but the effects of initialization and regularization were less pronounced compared to the $\ell^1$-case.

Plots for a single representative run are displayed in Figure 11. The explicit regularizer has the clearest effect in the high gain regime, where explicit regularization helps to achieve a better fit with the optimal transition curve and reduces the radial standard variation. Results were less sensitive to poor initialization than the corresponding experiments in one dimension. We conjecture that a higher degree of rigidity is introduced in this setting by the fact that there exists a *unique* minimum norm interpolant.

## A.9 He initialization

All experiments so far were performed with the initialization scaling of Glorot and Bengio [2010]. Especially for deeper neural networks, the initialization scheme of He et al. [2015] is very popular. Hanin and Rolnick [2018] proves in particular that He et al. [2015]'s normalization avoids the vanishing and exploding gradients phenomenon at initialization in expectation. While this consideration does not apply to our shallow networks, we find it informative to compare the two schemes.

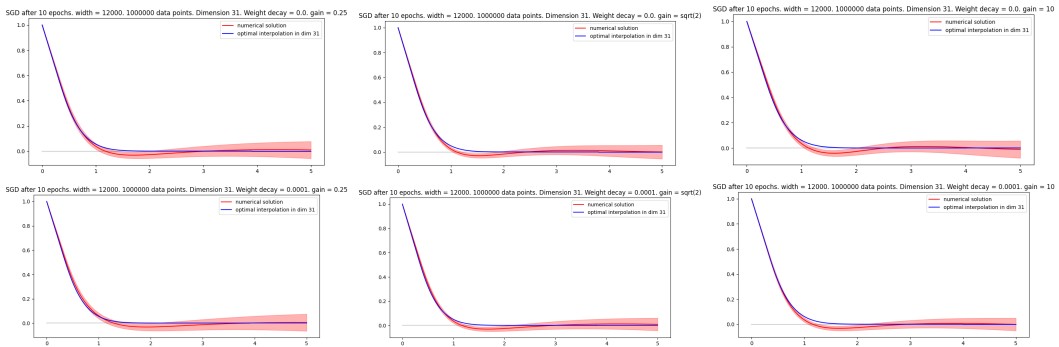

Figure 11: We vary initialization scale ($\alpha \in \{0.25, \sqrt{2}, 10\}$ from left to right) and consider training without weight decay (top row) and with weight decay $\lambda = 10^{-4}$ (bottom row). The rescaling factors are chosen to be $1/3.9$ for $\alpha = 0.25$, $1/3.8$ for $\alpha = \sqrt{2}$, and $1/4$ for $\alpha = 10$.

For shallow neural networks

$$f_m : \mathbb{R}^d \to \mathbb{R}, f_m(x) = \sum_{i=1}^{m} a_i \, \sigma(w_i \cdot x + b_i)$$

the effect of initialization is as follows:

1. According to Glorot and Bengio [2010], the parameters $a_i$ and $w_{ij}$, $1 \leq i \leq m$ and $1 \leq j \leq d$ are chosen as random variables with mean 0 and standard deviation $\sqrt{2/(m+1)}$ for $a_i$ and $\sqrt{2/(m+d)}$ for $w_{i,j}$. In particular, if $m$ is much larger than $d$, then the $|a_i| \, \|w_i\| = O(1/m)$. As we add $m$ terms of magnitude $\sim 1/m$, we consider this the 'law of large numbers' scaling.

2. According to He et al. [2015], the parameters $a_i$ and $w_{ij}$, $1 \leq i \leq m$ and $1 \leq j \leq d$ are chosen as random variables with mean 0 and standard deviation $\sqrt{2/m}$ for $a_i$ and $\sqrt{2/d}$ for $w_{i,j}$. In particular, if $m$ is much larger than $d$, then the $|a_i| \, \|w_i\| = O(1/\sqrt{m})$. As we add $m$ terms of mean zero and magnitude $\sim 1/\sqrt{m}$, we consider this the 'central limit theorem' scaling.

Many authors, such as Sirignano and Spiliopoulos [2020a,b, 2019], present the factor $1/m$ or $1/\sqrt{m}$ explicitly outside the neural network. As observed above, the effect of initialization can be significant. Unsurprisingly, results in the central limit regime, where all neurons contribute similarly at initialization, are more consistent and predictable. Experimental results are presented in Figure 12 in the one-dimensional setting and in Figures 13 and 14 in the case of radial symmetry. Notably, in high dimension, explicit regularization not only reduced radial variation, but also increased data compliance by reducing the rescaling factor $r_d$. In this setting, we observe the benefits of explicit regularization over relying on implicit bias only.

## A.10 Linearized dynamics

Parameter optimization in neural networks depends heavily on the choice of initialization. While the dynamics are truly non-linear in the regime studied by Chizat and Bach [2018], Rotskoff and Vanden-Eijnden [2018], Mei et al. [2018], Sirignano and Spiliopoulos [2020a] and Wojtowytsch [2020], there are scalings for which the directions $w_i$ remain close to their initialization for all time – see e.g. [E et al., 2019b] for a derivation. In this case, the solution produced by a neural network is similar to that of a random feature model. In this section, we numerically find the minimum norm interpolant of a random feature model by 'freezing' the inner layer coefficients at their random initialization. We find that the random feature solution differs geometrically from the Barron space solution, e.g. in that it is smooth at the origin, where the Barron space solution has a cone-like singularity of the form $f(x) = 1 - c_d\|x\|_2$ for small $x$. In particular, we find that our experiments were set appropriately in the non-linear training regime. See Figure 15.

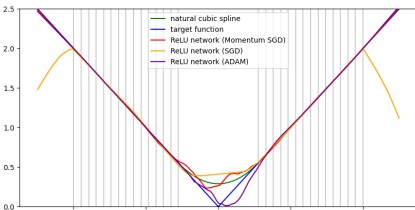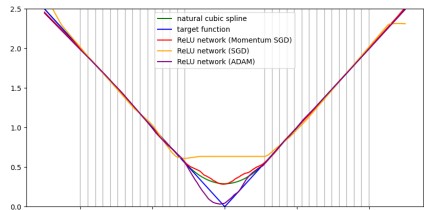

Figure 12: Experiments for He initialization in one Dimension in the same setting as Figure 2 with Xavier initialization. Left: No explicit regularization, right: Weight decay regularization $\lambda = 0.002$. Even for Glorot initialization with large gain, this regularizer was sufficient to induce convexity. For He initialization, it has a notable regularizing effect, but it is insufficient to impose convexity. We observe greater deviation from a linear function in the small intervals between known data points on either side of the big 'gap'.

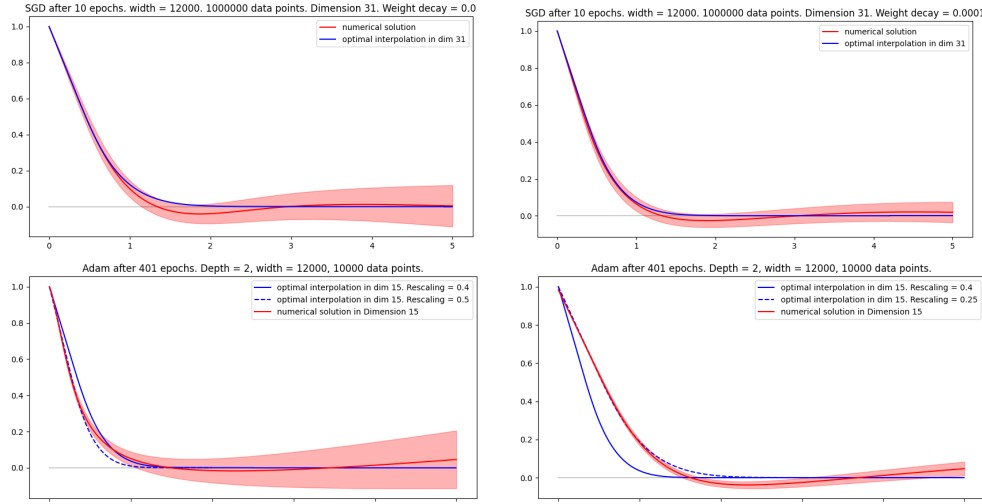

Figure 13: **Top row:** Experiments for He initialization in dimension 31 in the same setting as Appendix A.8. **Left**: No explicit regularization with a rescaling factor $r_d = 1/4.9$, **Right**: Weight decay regularization with $\lambda = 10^{-4}$ and rescaling factor $r_d = 1/4.2$. We observe that under He initialization the effects of the explicit regularizer is even more pronounced. Unlike in other experiments, the presence of regularization *increases* the rescaling factor and thus improves the fit to training data (for the radial profile).

**Bottom row:** We repeat the same as above with MSE loss rather than $\ell^1$-loss, using the Adam optimizer with learning rate $10^{-5}$ instead of SGD with momentum, and using an overparametrized rather than underparametrized neural network. Without explicit regularization (left), the radial standard deviation is substantial, while explicit regularization leads to a more radially symmetric function, albeit at the price of a higher rescaling factor. For easy comparison to Figure 3, we present the optimal profile as rescaled in the main document as well. Both functions achieve training loss $< 10^{-3}$, but clearly generalization is poor without regularization: While the radial average is close to the target function $f^* \equiv 0$ for inputs with $\|x\| \geq 1$, the radial variation is high.

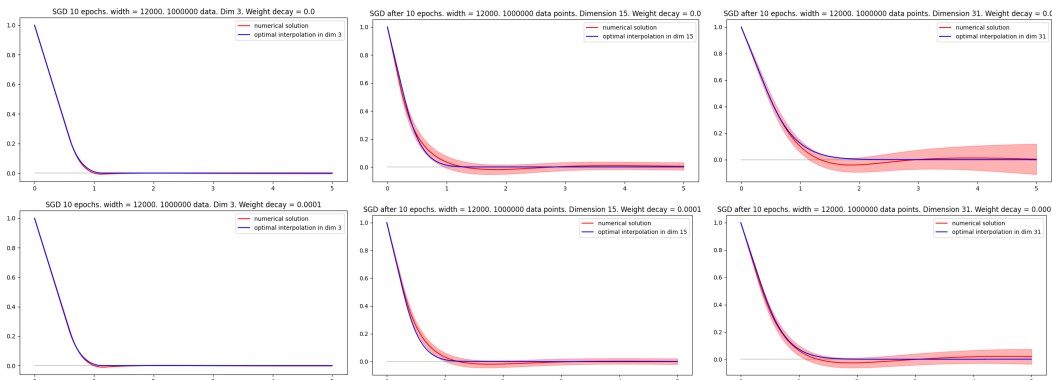

Figure 14: We examine the effects of the explicit regularizer (weight decay penalty $\lambda = 0$ for the top row and $\lambda = 10^{-4}$ for the bottom row) while varying dimensions ($d = 3, 15, 31$ from left to right) in the He initialization scheme. Optimizer settings were identical to the top row in Figure 13. The rescaling factors were $r_3 = 1/1.15$, $r_{15} = 1/2.1$, and $r_{31} = 1/4.2$. As dimension grows implicit bias may be insufficient to find a minimum norm interpolant shape with reasonable scaling factor and may not enforce radial symmetry. In this case explicit regularizer may have an advantage.

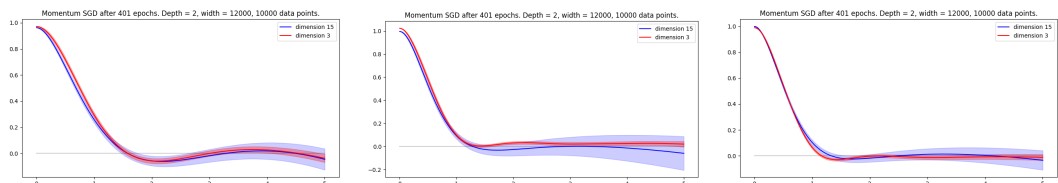

Figure 15: A random feature model trained on the same dataset as the neural networks. The solutions produced this way are geometrically distinct from neural network solutions as they are 'flat' at the origin. The left two figures correspond to different initializations: Gain $\alpha = \sqrt{2}$ (left) and gain $\alpha = 5$ (right). Notably, the variation is higher in radial direction in high dimension and higher than for the comparable neural network model. Perhaps surprisingly, higher gain appears to induce a better implicit bias in this case. No explicit regularization was used. Here we initialized the random feature by the same law as the neural network rather than initializing the outer layer at zero, since our goal is to study neural network dynamics, not find the optimal random feature solution. For the right plot, the initialization was random normal with gain 5 in the inner layer and zero in the outer layer with unsurprisingly better results.

### A.11 Leaky ReLU activation

As noted by Wojtowytsch [2022], minimum norm interpolation is not stable when passing to an equivalent norm. A Barron space theory can be developed in perfect analogy for networks with the leaky ReLU activation function, and it is easy to see that the 'Barron' spaces for both activation functions coincide with equivalent norms, depending on the negative slope of the leaky ReLU function. However, the minimum norm interpolant $f_d^*$ with respect to the ReLU-based Barron-norm is not guaranteed to coincide with the minimum interpolant for the leaky-ReLU-based Barron norm. Experimentally, however, we observe strong agreement between the geometry of numerical solutions here.

### A.12 Deeper neural networks

We train neural networks of depth $L > 2$ to fit the same radially symmetric data as in Section 5.2. We see in Figure 17 that for depth $L \geq 3$, weight decay-regularized networks strongly resemble the interpolant $f_{Lip}(x) = \max\{1 - \|x\|_2, 0\}$ with minimal (Euclidean) Lipschitz constant. This function can be written as a composition of two Barron functions $f = f_{bump} \circ f_{norm}$

$$f_{bump}(z) = \max\{1 - z, 0\} = \sigma(1 - z), \qquad f_{norm}(x) = \|x\|_2 = c_d\, \mathbb{E}_{\nu \sim \pi^0}\big[\sigma(\nu \cdot x)\big]$$

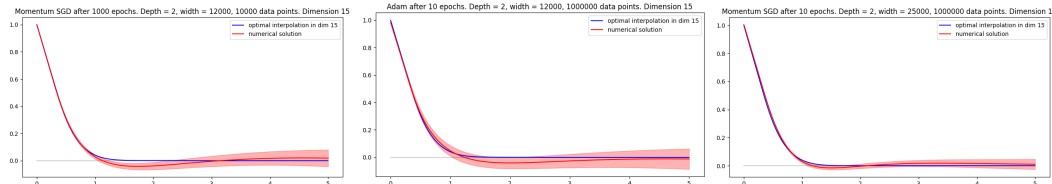

Figure 16: Neural networks with a single hidden layer and leaky ReLU activation $\sigma(z) = \max\{z, 0\} + 0.1 \min\{z, 0\}$ trained in the setting of Section 5.2. Without theoretical foundation, we observe that the shape of $f_d^*$ is attained to high accuracy also in this setting with the same rescaling factors as in the ReLU setting. **Left:** Momentum-SGD, **Middle:** Adam, **Right:** Momentum-SGD for a wider network with $m = 25,000$.

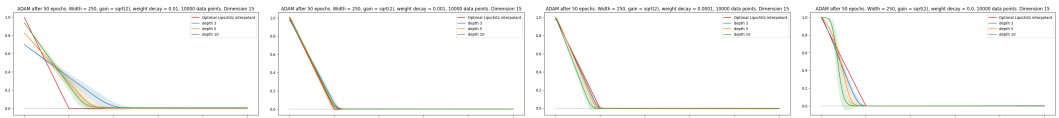

Figure 17: Neural networks of width 250 and varying depth were trained to fit data generated as in Section 5.2 with weight decay regularizers $10^{-2}, 10^{-3}, 10^{-4}$ and 0 (left to right). The initialization gain variable was chosen as $\alpha = \sqrt{2}$ as required to avoid exploding and vanishing gradients in deeper networks. Evidently, a small amount of weight decay regularization provides useful geometric prior without diminishing the quality of data fit.

Unlike networks with one hidden layer, deeper networks have positive (or nearly positive) outputs everywhere. While two-layer networks follow the minimum norm interpolation shape closely by the origin and have radial variances which increase outside the unit ball, deeper networks have positive radial variance inside the unit ball, but are essentially radially symmetric outside – compare e.g. Figure 9.

where $\pi^0$ denotes the uniform distribution on the unit sphere and $c_d \sim \sqrt{d}$ is a dimension-dependent constant. We can thus approximate $f_{Lip}$ efficiently by neural networks of depth $L \geq 3$ as long as the first layer is sufficiently wide.

Unlike their shallow counterparts, neural networks with multiple hidden layers have no strong geometric prior without weight decay regularization. With weight decay, the observed behavior was relatively stable over a range of dimensions, initializations scalings and optimization algorithms. We are led to conjecture that $f_{Lip}$ is a minimum norm interpolant in this setting. The statement remains imprecise at this point as no function space theory for deeper networks with weight decay regularizer has been developed to the same extent as Barron space theory.

## B   Γ-convergence

In this appendix, we recall the definition and a few properties of Γ-convergence, a popular notion of the convergence of functionals introduced by De Giorgi and Franzoni [1975] in the calculus of variations to study the convergence of minimization problems. Braides [2002], Dal Maso [2012] provide introductions to the theory and its applications. As the notion is likely not familiar to readers from the machine learning community, we provide some full proofs as well.

**Definition B.1.** *Let $(X, d)$ be a metric space and $F_n, F : X \to \mathbb{R} \cup \{-\infty, \infty\}$ be functions. We say that $F_n$ converges to $F$ in the sense of Γ-convergence if two conditions are met:*

1. *($\liminf$-inquality) If $x_n$ is a sequence in $X$ and $x_n \to x$, then $\liminf_{n\to\infty} F_n(x_n) \geq F(x)$.*

2. *($\limsup$-inequality) For every $x \in X$, there exists a sequence $x_n^* \in X$ such that $x_n^* \to x$ and $\limsup_{n\to\infty} F_n(x_n^*) \leq F(x)$.*

Intuitively, the first condition means that $F(x)$ is (almost) a lower bound for $F_n(x_n)$ if $n$ is 'large' and $x_n$ is 'close' to $x$, while the second condition means that there is no larger lower bound that we could choose. The sequence $x_n^*$ is often referred to as a 'recovery sequence'. Of course,

combining the liminf- and limsup-inequalities, we find that in fact $F_n(x_n^*) \to F(x)$. We employ $\Gamma$-convergence when dealing with minimization problems where uniform convergence fails, but we hope for convergence of minimizers to minimizers.

Often, $\Gamma$-convergence is considered as a continuous parameter $\varepsilon$ approaches $0^+$ rather than as the discrete parameter $n$ approaches infinity. The definitions remain largely identical (with obvious substitutions).

To get a feeling for $\Gamma$-convergence, we consider a particularly simple situation by looking at two constant sequences of functions. Note that the sequence is constant, not the functions.

**Example B.2.** *Let $X = \mathbb{R}$ and consider the constant sequences*

$$F_n(x) = f(x) = \begin{cases} 1 & x \neq 0 \\ 0 & x = 0 \end{cases}, \qquad G_n(x) = g(x) = \begin{cases} 0 & x \neq 0 \\ 1 & x = 0 \end{cases}.$$

*We claim that*

$$\left(\Gamma - \lim_{n\to\infty} F_n\right)(x) = f(x), \quad \left(\Gamma - \lim_{n\to\infty} G_n\right)(x) = 0 \qquad \forall\, x \in \mathbb{R}.$$

*If $x_n \to x$ and $x \neq 0$, then $F_n(x_n) = 1$ and $G_n(x_n) = 0$ for all but finitely many $n \in \mathbb{N}$, meaning that $F_n(x_n) \to 1 = f(x)$ and $G_n(x_n) \to 0$. It remains to consider the case $x_n \to 0$.*

*We see immediately that $F_n(x_n) \geq 0 = f(0)$ for all $n \in \mathbb{N}$. Conversely, if we take $x_n^* = 0$ for all $n$, then $x_n^* \to 0$ and $F_n(x_n^*) = f(0) \to f(0)$. In total, we conclude that $\Gamma - \lim F_n = f$.*

*For $G_n$, we find that $G_n(x_n) \geq 0$ for all $n \in \mathbb{N}$. Additionally, we can choose the sequence $x_n = 1/n$ such that $G_n(x_n) = 0$ for all $n \in \mathbb{N}$. Altogether, we find that $\Gamma - \lim G_n = 0$.*

More generally, if $F_n = F$ for all $n \in \mathbb{N}$ and some $F : X \to \mathbb{R}$, then $\Gamma - \lim_{n\to\infty} F_n = \overline{F}$ is the lower semi-continuous envelope of $F$. In particular, $\Gamma - \lim_{n\to\infty} F = F$ if and only if $F$ is lower semi-continuous. The main useful properties of $\Gamma$-convergence are summarized in the following lemma.

**Lemma B.3.** *Assume that $F_n \to F$ in the sense of $\Gamma$-convergence, $\varepsilon_n \to 0^+$ and $x_n \in X$ is a sequence such that*

$$F_n(x_n) \leq \inf_{x \in X} F_n(x) + \varepsilon_n.$$

*Assume that $x_n \to x^*$. Then $F(x^*) = \inf_{x \in X} F(x)$. In particular, if $x_n$ is a minimizer of $F_n$ and the sequence $x_n$ converges, then the limit point is a minimizer of $F$.*

Clearly, this is most useful if we can guarantee that the sequence $x_n$ converges. For many useful sequences of functionals, the existence of a convergent subsequence can be established by compactness. This is easily sufficient, as we can also pass to a subsequence in $F_n$.

*Proof.* Due to the liminf-inequality, we have

$$F(x^*) \leq \liminf_{n\to\infty} F_n(x_n) = \liminf_{n\to\infty} \inf_{x \in X} F_n(x).$$

On the other hand, let $x \in X$ be any point. Then, due to the limsup-inequality, there exists some sequence $x_n'$ such that

$$x_n' \to x, \qquad F(x) = \lim_{n\to\infty} F_n(x_n') \geq \liminf_{n\to\infty} \inf_{x \in X} F_n(x).$$

In particular $\inf_{x \in X} F(x) \geq \liminf_{n\to\infty} \inf_{x \in X} F_n(x)$. Combining the two estimates, we find that $F(x^*) \leq \inf_{x \in X} F(x)$, which means that $x^*$ is a minimizer of $F$. $\qquad\square$

For completeness, a few observations are in order.

1. The notion of $\Gamma$-convergence relies on the notion of convergence on the underlying space $X$, and $\Gamma$-limits can change when keeping the set $X$ fixed, but passing to a different topology (e.g. a weak topology in infinite-dimensional spaces).

2. $\Gamma$-convergence is made for minimization problems, and it does not behave well under multiplication by negative real numbers: In general $\Gamma - \lim(-F_n) \neq -(\Gamma - \lim F_n)$, even if both limits exist. The reason is the asymmetry between the $\liminf$- and the $\limsup$-condition. To see this, consider for instance $F_n$ and $G_n - 1$ in Example B.2.

3. If $F_n \to F$ and $G_n \to G$, it is not necessarily true that $F_n + G_n \to F + G$. While it remains true that $\liminf_{n\to\infty}(F_n + G_n)(x_n) \geq (F + G)(x)$ if $x_n \to x$, it may no longer be possible to find a recovery sequence $x_n$ for $F_n + G_n$. For example, if $F_n = 1_{\mathbb{Q}}$ and $G_n = 1_{\mathbb{R}\setminus\mathbb{Q}}$ for all $n \in \mathbb{N}$, then $F_n \xrightarrow{\Gamma} 0$ and $G_n \xrightarrow{\Gamma} 0$, but $F_n + G_n = 1$ for all $n$ and $F_n + G_n \xrightarrow{\Gamma} 1$.

   However, if $F_n \xrightarrow{\Gamma} F$ and $G_n$ converges to a *continuous* limit $G$ *uniformly*, then $(F_n + G_n) \xrightarrow{\Gamma} F + G$. In particular, uniform convergence implies $\Gamma$-convergence. Namely, if $G_n \to G$ uniformly, $G$ is continuous and $x_n \to x$, then for given $\varepsilon > 0$, we can choose $N \in \mathbb{N}$ so large that

   (a) $|G(x_n) - G(x)| < \varepsilon/2$ for all $n \geq N$ since $G$ is continuous at $x$ and
   (b) $|G_n(x_n) - G(x_n)| < \varepsilon/2$ for all $n \geq N$ due to uniform convergence.

   Then
   $$\big|G_n(x_n) - G(x)\big| \leq \big|G_n(x_n) - G(x_n)\big| + \big|G(x_n) - G(x)\big| < \varepsilon.$$

   In particular $G_n(x_n) \to G(x)$. Recall that $G$ is guaranteed to be continuous if $G_n$ is continuous for all $n \in \mathbb{N}$.

4. $\Gamma$-convergence is unrelated to pointwise convergence of functions: Neither does it imply pointwise convergence, nor is it implied by it. Namely, the sequence $G_n$ in Example B.2 has the function $g$ as a pointwise limit and the constant function 0 as a $\Gamma$-limit.

5. $\Gamma$-convergence is not a notion of convergence derived from a topology. Indeed, even if $F_n$ is a constant sequence, i.e. if $F_n = G$ for all $n$, it may happen that $\Gamma - \lim_{n\to\infty} F_n \neq G$ (see $G_n$ in Example B.2). The $\Gamma$-limit is related to $G$, though: It is the lower semi-continuous envelope of the function $G$. In fact, every $\Gamma$-limit is lower semi-continuous.

Despite its somewhat counterintuitive properties, $\Gamma$-convergence has proved invaluable in many areas of the calculus of variations. It has been applied to homogenization by Bach et al. [2021], dimension reduction for thin sheets and shells by Friesecke et al. [2002a, 2003, 2002b], Bhattacharya et al. [2016], Lewicka et al. [2010] and the study of phase boundaries by Modica and Mortola [1977], Modica [1987]. While the $\Gamma$-convergence of functionals does not imply the convergence of their gradient flows even in situations of practical significance (see e.g. the example of Dondl et al. [2019]), Serfaty [2011], Sandier and Serfaty [2004], Mugnai and Röger [2011], Ilmanen [1993], Alikakos et al. [1994] provide important examples of situations where this can be established in a suitable sense. Even more, Bronsard and Kohn [1990] use $\Gamma$-convergence to gain insight into PDE dynamics.

## C  Homogeneous Barron spaces

In this section, we introduce the abstract framework which is used to prove our main theoretical result, Corollary 3.3. A neural network with $m$ neurons in a single hidden layer can be represented as

$$f_m(x) = b_0 + \sum_{i=1}^{m} a_i\, \sigma(w_i^T x + b_i) \qquad \text{or} \quad f_m(x) = b_0 + \frac{1}{m}\sum_{i=1}^{m} a_i\, \sigma(w_i^T x + b_i). \tag{4}$$

The network weights and biases are $(a, W, b) \in \mathbb{R}^m \times \mathbb{R}^{m\times d} \times \mathbb{R}^{m+1}$. The normalization depends on personal preference, with the former being more common in practice and the latter more common in theoretic analyses. We define the weight decay regularizer by

$$R_{WD}(a, W, b) = \frac{\|a\|_{\ell^2}^2 + \|W\|_F^2}{2} = \frac{1}{2}\left(\sum_{i=1}^{m} a_i^2 + \sum_{i=1}^{m}\sum_{j=1}^{d} w_{ij}^2\right)$$

or $R_{WD}(a, W, b) = \frac{1}{2m}\sum_{i=1}^{m}\left(a_i^2 + \|w_i\|_{\ell^2}^2\right)$ respectively. Here $\|\cdot\|_2$ denotes the Euclidean $\ell^2$-norm of a vector and $\|W\|_F$ denotes the Frobenius norm of the matrix $W$ whose rows are the vectors $w_i^T$. Note that we do not control the magnitude of the biases $b_i$ in the regularizer. This is a common approach, as the bias does not influence the Lipschitz-constant of the function represented by the neural network, which is useful in studying the generalization of the neural network.

We study function classes corresponding to arbitrarily wide neural networks with a single hidden layer, where the norm corresponds to the weight decay regularizer. We dub these function spaces 'homogeneous Barron spaces' in analogy to the more classical Barron spaces studied by E et al. [2019c], Ma et al. [2020], E and Wojtowytsch [2020], which correspond to a weight decay regularizer which also controls the bias. For homogeneous Barron spaces, coordinate transformations by Euclidean motions induce an isometry of the function class, while the origin plays a special role in classical Barron spaces. This justifies our terminology, as the data space is treated as isotropic and homogeneous by this function class. Homogeneous Barron spaces have also been studied by Ongie et al. [2019], Parhi and Nowak [2021, 2022] under the name Radon-BV spaces. A closely related class of spaces has been considered as the 'variation spaces of the ReLU dictionary' by Siegel and Xu [2020, 2022, 2023].

Heuristically, (homogeneous) Barron spaces are a function class tailored to replacing the finite superposition of ReLU ridges in (4) by an arbitrary superposition while keeping the weight decay regularizer finite. Due to the lack of control over the bias term, we will see that a slightly awkward technical definition is needed. Let $\pi$ be a probability distribution on the parameter space $\mathbb{R} \times \mathbb{R}^d \times \mathbb{R}$. We would like to define

$$f_{\pi,b_0} : \mathbb{R}^d \to \mathbb{R}, \quad f_{\pi,b_0}(x) = b_0 + \mathbb{E}_{(a,w,b)\sim\pi}\big[a\,\sigma(w^T x + b)\big] = \int_{\mathbb{R}\times\mathbb{R}^d\times\mathbb{R}} a\,\sigma(w^T x + b)\,\mathrm{d}\pi_{(a,w,b)}$$

and

$$R_{WD}(\pi) = \frac{1}{2}\int_{\mathbb{R}\times\mathbb{R}^d\times\mathbb{R}} |a|^2 + \|w\|_2^2\,\mathrm{d}\pi_{(a,w,b)}.$$

$f_{\pi,b_0}$ is an analogue of neural networks with a single hidden layer, but of arbitrary and possibly uncountably infinite width. Every finite neural network can be expressed in this fashion for the empirical measure $\pi_m = \frac{1}{m}\sum_{i=1}^m \delta_{(a_i,w_i,b_i)}$, in which case $R_{WD}(\pi_m) = R_{WD}(a, W, b)$.

Unfortunately, even if $R_{WD}(\pi) < \infty$, it is not clear that the integral defining $f_\pi$ exists in a meaningful sense. We do however note that formally

$$|f_\pi(x) - f_\pi(y)| = \mathbb{E}\big[a\{\sigma(w^T x + b) - \sigma(w^T y + b)\}\big] \leq \mathbb{E}\big[|a|\,\big||w^T x + b| - |w^T y + b|\big|\big]$$

$$\leq \|x - y\|\,\mathbb{E}\big[|a|\,\|w\|\big] \leq \frac{\|x - y\|}{2}\mathbb{E}\big[|a|^2 + \|w\|^2\big] = \|x - y\|\,R_{WD}(\pi).$$

Thus, the integral defining $f_\pi(x)$ exists for all $x$ if and only if it exists for, say, $x = 0$. We exploit this in the following modified definition. Let $\pi$ be a probability distribution on the parameter space $\mathbb{R}\times\mathbb{R}^d\times\mathbb{R}$ and $y \in \mathbb{R}$. We denote

$$f_{\pi,y}(x) = y + \mathbb{E}_{(a,w,b)\sim\pi}\big[a\big(\sigma(w^T x + b) - \sigma(b)\big)\big]$$

By the same argument as before, we observe that

1. $f_{\pi,y}(0) = y$ and
2. $|f_{\pi,y}(x) - f_{\pi,y}(x')| \leq R_{WD}(\pi)\,\|x - x'\|$.

The class of functions of the form $f_{\pi,y}$ forms the homogeneous Barron space. Still, every finite neural network $f_m$ can be represented in this fashion with $b_0 = y - \sum_{i=1}^m a_i\sigma(b_i)$ or $b_0 = y - \frac{1}{m}\sum_{i=1}^m a_i\sigma(b_i)$.

**Definition C.1** (Homogeneous Barron space). *Let $f : \mathbb{R}^d \to \mathbb{R}$ be a function. We define the semi-norms*

$$[f]_{\mathcal{B}} = \inf_{f \equiv f_{\pi,y}} R_{WD}(\pi), \qquad [f]_0 = |f(0)|.$$

*The homogeneous Barron space is the function class $\mathcal{B}(\mathbb{R}^d) = \{f : \mathbb{R}^d \to \mathbb{R} : [f]_{\mathcal{B}} < \infty\}$. By the definition of a function $[f]_0 < \infty$ is automatically true.*

We note a few important properties. First, we consider two function classes:

$$\mathcal{F}_Q = \overline{\mathrm{conv}\big\{a\,\big(\sigma(w \cdot x + b) - \sigma(b)\big) : a^2 + \|w\|^2 \leq 2Q\big\}}$$

$$\mathcal{F}_Q(R) = \overline{\mathrm{conv}\big\{a\,\big(\sigma(w \cdot x + b) - \sigma(b)\big) : a^2 + \|w\|^2 \leq 2Q, |b| \leq \sqrt{Q}\,R\big\}}.$$

Note that $\mathcal{F}_Q(R) \subseteq \mathcal{F}_Q \subseteq \{f \in \mathcal{B} : f(0) = 0, [f]_\mathcal{B} \leq Q\}$. The closure is taken with respect to locally uniform convergence, i.e. pointwise convergence which is uniform on all compact sets.[1] Due to the homogeneity of ReLU activation, we may prove the following.

**Lemma C.2.** *The identity $\mathcal{F}_Q = \{f \in \mathcal{B} : f(0) = 0, [f]_\mathcal{B} \leq Q\}$ holds.*

While the claim is natural, its proof is surprisingly technical and postponed until the end of the section. The class $\mathcal{F}_Q(R)$ will be used below for technical purposes. Of major importance below are the compact embedding theorem and the direct approximation theorem.

**Theorem C.3** (Compact embedding). *Let $f_n \in \mathcal{B}$ be a sequence such that $\liminf_{n \to \infty} [f_n]_0 + [f_n]_\mathcal{B} < +\infty$. Then there exists $f$ in $\mathcal{B}$ such that*

1. *$f_n \to f$ in $C^0(K)$ for all compact sets $K \subseteq \mathbb{R}^d$.*

2. *$f_n \to f$ in $L^p(\mu)$ for all measures $\mu$ with finite p-th moments, $p \in [1, \infty)$.*

3. *$[f]_\mathcal{B} \leq \liminf_{n \to \infty} [f_n]_\mathcal{B}$.*

*Proof.* It is sufficient to show that the set $\tilde{\mathcal{F}}_Q = \{a(\sigma(w \cdot x + b) - \sigma(b)) \mid a^2 + |w|^2 \leq 2Q\}$ is compact in $C^0(K)$ and $L^2(\mu)$ for all $K$ and $\mu$ as above, in which case also its closed convex hull is compact [Rudin, 1991, Theorem 3.20.]. To this end, observe that the map

$$F : \mathbb{R} \times \mathbb{R}^d \times \mathbb{R} \to C^0(K), \qquad (a, w, b) \mapsto a(\sigma(w \cdot + b) - \sigma(b))$$

is continuous for any compact set $K$. Since $K$ is compact, we have $K \subseteq B_R(0)$ for some $R > 0$ and thus $|w \cdot x| \leq \sqrt{2Q}R$ for all $x \in K$. In particular,

$$\sigma(w \cdot x + b) - \sigma(b) = \begin{cases} w \cdot x & \text{if } b > \sqrt{2Q}R \\ 0 & \text{if } b < \sqrt{2Q}R \end{cases}.$$

Hence $\tilde{\mathcal{F}}_Q = F(\{(a, w, b) : a^2 + |w|^2 \leq 2Q, b \leq \sqrt{2Q}R\})$ is the continuous image of a compact set, hence compact. We have thus proved a compact embedding into $C^0(K)$ for any compact set $K$.

Exhausting $\mathbb{R}^d$ by the sequence of compact sets $\overline{B_m(0)}$, $m \in \mathbb{N}$ and using a diagonal sequence argument, we see that under the conditions of Theorem C.3, there exists $f \in \mathcal{B}$ such that $f_n \to f$ pointwise everywhere on $\mathbb{R}^d$ and uniformly on compact subsets. Additionally, we observe that $f(0) = 0$ and there exists a uniform upper bound on the Lipschitz constants of the sequence $f_n$. We conclude that $f_n \to f$ in $L^p(\mu)$ from the Dominated Convergence Theorem using $Q\|x\|$ as a dominating function. $\square$

As a consequence, we find the following.

**Corollary C.4.**     *1. $\mathcal{B}$ is a Banach space.*

  2. *$C_c^\infty(\mathbb{R}^d) \subseteq \mathcal{B}(\mathbb{R}^d)$.*

*Proof.* The first claim follows as in [Siegel and Xu, 2021, Lemma 1], where it is proved in a more general context for dictionaries which are compact in a Hilbert space – in our case, the dictionary $x \mapsto a\{\sigma(w^T x + b) - \sigma(b)\}$. The second claim follows from [Ongie et al., 2019, Corollary 1]. $\square$

We conclude with a theorem which establishes a rate of approximation for Barron functions in a weaker topology.

**Theorem C.5** (Direct approximation). *Let $f \in \mathcal{B}$ and $\mu$ a measure on $\mathbb{R}^d$ with finite second moments. Then for any $m \in \mathbb{N}$ there exist $c \in \mathbb{R}$ and $(a_i, w_i, b_i) \in \mathbb{R} \times \mathbb{R}^d \times \mathbb{R}$ such that*

$$\sum_{i=1}^m a_i^2 + \|w_i\|^2 \leq [f]_\mathcal{B}, \quad \left\| f - c - \sum_{i=1}^m a_i \sigma(w_i^T x + b_i) \right\|_{L^2(\mu)} \leq \frac{2[f]_\mathcal{B}}{\sqrt{m}} \sup_{\|w\|=1} \sqrt{\int_{\mathbb{R}^d} |w^T x|^2 \, d\mu_x}.$$

---

[1] This notion of convergence is generated by a topology, but not a metric. Other notions of convergence can be considered and induce the same function class.

*Proof.* A proof of this result can be found in [Wojtowytsch, 2022, Appendix C] in the proof of Proposition 2.6. □

*Proof of Lemma C.2.* **Step 1.** Assume that $f \in \mathcal{B}$ such that $f(0) = 0$ and $[f]_\mathcal{B} \leq Q$. By the Direct Approximation Theorem (which is proved in [Wojtowytsch, 2022, Appendix C] without using Lemma C.2), we find that for every $m \in \mathbb{N}$ and every measure $\mu$ on $\mathbb{R}^d$ with finite second moments, there exists

$$f_m(x) = \frac{1}{m} \sum_{i=1}^{m} a_i \{\sigma(w_i \cdot x + b_i) - \sigma(b_i)\} \quad \in \mathcal{F}_Q$$

such that $\|f_m - f\|_{L^2(\mu)} \leq C_\mu \|f\|_\mathcal{B} m^{-1/2}$. In particular, $f$ is in the closed convex hull of $\{a\left(\sigma(w^T x + b) - \sigma(b)\right) : a^2 + |w|^2 \leq 2Q\}$ if the closure is taken with respect to the $L^2(\mu)$ topology. Additionally, the sequence $f_m$ has a uniformly bounded Lipschitz constant and is therefore compact in $C^0(K)$ for all compact $K$ by a corollary to the Arzela-Ascoli theorem [Dobrowolski, 2010, Satz 2.42]. In particular, $f_m \to f$ uniformly and thus $f \in \mathcal{F}_Q$.

**Step 2.** Denote $f_{(a,w,b)}(x) = a\{\sigma(w^T x + b) - \sigma(b)\}$. Since $f_{(a,w,b)}(0) = a\{\sigma(b) - \sigma(b)\} = 0$ for all $a, w, b$, we conclude that $f(0) = 0$ for all $f \in \mathcal{F}_Q$. If $f \in \mathcal{F}_Q$, then there exists a sequence

$$f_n(x) = \sum_{i=1}^{N_n} \lambda_{i,n} a_{i,n} \left\{\sigma(w_{i,n} \cdot x + b_{i,n}) - \sigma(b_{i,n})\right\}$$

such that $f_n \to f$ locally uniformly. If the biases remain uniformly bounded, the sequence of empirical distributions

$$\pi_n = \frac{1}{N_n} \sum_{i=1}^{N_n} \lambda_{i,n} \, \delta_{(a_{i,n}, w_{i,n}, b_{i,n})}$$

has a convergent subsequence by Prokhorov's Theorem [Klenke, 2006, Satz 13.29]. We denote the limiting distribution as $\pi$. The convergence of Radon measures implies the convergence of $f_n$ to $f_\pi(x) = \mathbb{E}_{(a,w,b) \sim \pi}\left[a\{\sigma(w^T x + b) - \sigma(b)\}\right]$ by definition. Since $f_n$ converges locally uniformly by assumption, $f = f_\pi \in \mathcal{B}$ and $[f]_\mathcal{B} \leq Q$.

If the biases do not remain bounded, we note that for every compact set $K \subseteq \mathbb{R}^d$ we can extract a convergent subsequence of the measures by the same argument used to prove Theorem C.3, effectively making the sequence of biases bounded. We can extend the argument to the entire space exploiting that

$$\lim_{b \to \infty} \left(\sigma(w \cdot x + b) - \sigma(b)\right) \to \sigma(b/|b|) \, w \cdot x$$

locally uniformly. □

# D   Rademacher complexity of homogeneous Barron space

Following a classical strategy implemented e.g. by E et al. [2019a] in a similar context, we estimate the Rademacher complexity of homogeneous Barron space and use it to bound the generalization gap (i.e. the discrepancy between empirical risk and population risk). In our setting, we face additional technical obstacles:

1. We deal with general sub-Gaussian data distributions $\mu$ rather than data distributions with compact support.

2. We do not control the magnitude of the bias variables.

3. We consider $\ell^2$-loss, which is neither globally Lipschitz-continuous nor bounded.

In combination, these complications require a refined technical analysis similar to Appendix C. Let us summarize several notations which will be needed below.

- $\widehat{\mathrm{Rad}}$ – the empirical Rademacher complexity of a function class over a given dataset.

- $\mathrm{Rad}_n$ – the expected Rademacher complexity of a function class over a data set composed of $n$ iid samples from the data distribution $\mu$.

- $\mathcal{R}$ – the population risk $\mathcal{R}(f) = \|f - f^*\|^2_{L^2(\mu)} = \mathbb{E}_{x \sim \mu}[|f(x) - f^*(x)|^2]$. We generally take this to operate on the level of functions, parametrized or not. By an abuse of notation, we identify $\mathcal{R}(a, W, b) := \mathcal{R}(f_{(a,W,b)})$.

- $\widehat{\mathcal{R}}_n$ – the empirical risk $\widehat{\mathcal{R}}_n(f) = \|f - f^*\|^2_{L^2(\mu_n)} = \frac{1}{n} \sum_{i=1}^n |f(x_i) - f^*(x_i)|^2$ over a data set $\{x_1, \dots, x_n\}$ where $\mu_n = \frac{1}{n} \sum_{i=1}^n \delta_{x_i}$ is the empirical measure. Equally, $\widehat{\mathcal{R}}_n$ can be considered for functions or parameters with the natural identification.

- $\widehat{\mathcal{R}}_{n,m,\lambda}$. The regularized empirical risk

$$\widehat{\mathcal{R}}_{n,m,\lambda}(a, W, b) = \widehat{\mathcal{R}}_n(a, W, b) + \frac{\lambda}{2}\big(\|a\|^2 + \|W\|^2\big).$$

  We only consider this quantity on the parameter level, where it is computable. While the weight decay regularizer is an upper bound for $[f_{(a,W,b)}]_{\mathcal{B}}$, the two are generally not the same since the parameter-to-function map of a neural network is generally not injective.

- $R_{WD}$ – the weight decay regularizer.

- $\mathcal{F}_Q$ – the set of functions for which $[f]_{\mathcal{B}} \leq Q$ and $f(0) = 0$.

- $\mathcal{F}_{A,Q}$ – the set of functions for which $[f]_{\mathcal{B}} \leq Q$ and $|f(0)| \leq A$.

As is common in the mathematics community, $C$ will generally denote a constant which does not depend on quantities (unless specified otherwise) and which may change value from line to line. Some facts about sub-Gaussian distributions, which we believe to be well-known to the experts, are collected in Appendix H.

**Definition D.1** (Rademacher Complexity). *Let $S = \{x_1, \dots, x_n\}$ be a set of points in $\mathbb{R}^d$ (a data sample) and $\mathcal{F}$ a real-valued function class. We define the empirical Rademacher complexity of $\mathcal{F}$ on the data sample as*

$$\widehat{\mathrm{Rad}}(\mathcal{F}; S) = \mathbb{E}_\varepsilon \left[ \sup_{f \in \mathcal{F}} \frac{1}{n} \sum_{i=1}^n \varepsilon_i f(x_i) \right]$$

*where $\varepsilon_i$ are iid random variables which take the values $\pm 1$ with equal probability $\frac{1}{2}$. The population Rademacher complexity is defined as*

$$\mathrm{Rad}_n(\mathcal{F}) = \mathbb{E}_{S \sim \mu^n}\big[\widehat{\mathrm{Rad}}(\mathcal{F}; S)\big],$$

*i.e. as the expected empirical Rademacher complexity over a set of $n$ iid data points.*

In this section, we will find a upper bound of Rademacher Complexity of $\mathcal{B}$. We will denote by $S_n$ the set of $n$ samples, and $\widehat{\mathrm{Rad}}(\mathcal{F}, S_n)$ the sample Rademacher Complexity of $\mathcal{F}$ given the samples $S_n$. We furthermore denote $R := \max\{\|x_1\|, \dots, \|x_n\|\}$ and consider the function classes $\mathcal{F}_Q$ and $\mathcal{F}_Q(R)$ as in Appendix C:

$$\mathcal{F}_Q = \overline{\mathrm{conv}\big\{a\left(\sigma(w \cdot x + b) - \sigma(b)\right) : a^2 + \|w\|^2 \leq 2Q\big\}}$$
$$\mathcal{F}_Q(R) = \overline{\mathrm{conv}\big\{a\left(\sigma(w \cdot x + b) - \sigma(b)\right) : a^2 + \|w\|^2 \leq 2Q, |b| \leq \sqrt{Q}\,R\big\}}.$$

**Lemma D.2.** *Let $S_n = \{x_1, \dots, x_n\}$ be a data set in $\mathbb{R}^d$. Then*

$$\widehat{\mathrm{Rad}}(\mathcal{F}_Q, S_n) \leq \frac{(1 + 3\sqrt{2})Q}{\sqrt{n}} \max_{1 \leq i \leq n} \|x_i\|.$$

*Assume $\mu$ is a $\sigma^2$ sub-Gaussian distribution in $\mathbb{R}^d$. Then*

$$\mathrm{Rad}(\mathcal{F}_Q) \leq (1 + 3\sqrt{2})Q \left( \frac{\mathbb{E}_{x \sim \mu}[\|x\|]}{\sqrt{n}} + \sigma\sqrt{2\frac{\log n}{n}} \right)$$

*for all $n \geq 2$.*

*Proof.* Initially, we fix a set $S = \{x_2, \ldots, x_n\}$ of $n$ points. We will later take the expectation over $S$, using the sub-Gaussian property of $\mu$ for an explicit norm bound. Define $R := \max_{1 \leq i \leq n} \|x_i\|$. To this end, we first prove the following claim, which enables us to focus on only single neuron functions instead of entire $\mathcal{F}_Q$:

**Claim:** Let $\varepsilon_1, \ldots, \varepsilon_n \in \mathbb{R}$. Then

$$\sup_{\mathcal{F}_Q} \sum_i \epsilon_i f(x_i) = \sup_{a^2 + \|w\|^2 \leq 2Q} \sum_i \epsilon_i a \{ \sigma(w^T x_i + b) - \sigma(b) \}$$

*Proof of Claim.* Note that $\mathcal{F}_Q$ is the closed convex hull of single neuron ridge functions, i.e. single neuron ridge functions are the extreme points of the closed convex set $\mathcal{F}_Q$.

To verify the claim, first note that $f \mapsto \sum_{i=1}^n \varepsilon_i f(x_i)$ is a continuous linear functional on $C^0(K)$ for any compact $K \subseteq \mathbb{R}^d$ containing the finite set $S$. It is well known that $C^0(K)$ is a Banach Space. Therefore, if $\{ a(\sigma(w \cdot x + b) - \sigma(b)) \mid a^2 + |w|^2 \leq 2Q \}$ is compact in $C^0(K)$, then [Rudin, 1991, Theorem 3.20.] implies that $\mathcal{F}_Q$, a closed convex hull of the compact set, is also compact. Then, from the compactness of $\mathcal{F}_Q$, we can use Bauer [1958]'s maximum principle and see that the supremum is attained at an extreme point. Compactness follows from the compact embedding Theorem, see Theorem C.3 above. $\qquad\square$

Over the next steps, we will bound $\widehat{\mathrm{Rad}}(\mathcal{F}_Q, S_n)$.

**Step 1.** In this step, we prove that

$$\widehat{\mathrm{Rad}}(\mathcal{F}_Q; S_n) = \widehat{\mathrm{Rad}}(\mathcal{F}_Q(R); S_n).$$

To show this, we first observe that if $|b| \geq \|w\| R$, then $\sigma(w \cdot x + b) - \sigma(b) = \sigma(sgn(b)) w \cdot x$ since $|w^T x_i| \leq \|w\| \|x_i\| \leq |b| R$ for all $1 \leq i \leq n$. This means for $\forall |b| \geq \|w\| R$, the precise value of $b$ does not change the value of $\sigma(w \cdot x + b) - \sigma(b)$.

Now, we compute the $\widehat{\mathrm{Rad}}(\mathcal{F}_Q, S_n)$:

$$n\,\widehat{\mathrm{Rad}}(\mathcal{F}_Q, S_n) = \mathbb{E}_\epsilon \left[ \sup_{\mathcal{F}_Q} \sum_i \epsilon_i f(x_i) \right]$$

$$= \mathbb{E}_\epsilon \left[ \sup_{a^2 + \|w\|^2 \leq 2Q} \sum_i \epsilon_i a \{ \sigma(w \cdot x_i + b) - \sigma(b) \} \right]$$

$$= \mathbb{E}_\epsilon \left[ \sup_{a^2 + \|w\|^2 \leq 2Q, |b| \leq \|w\| R} \sum_i \epsilon_i a \{ \sigma(w \cdot x_i + b) - \sigma(b) \} \right]$$

$$= n\,\widehat{\mathrm{Rad}}(\mathcal{F}_Q(R), S_n)$$

For the first line, we used the claim.

**Step 2.** Using the uniform bound on the magnitude of the bias from the previous step, in this step we bound the Rademacher complexity by

$$\widehat{\mathrm{Rad}}(\mathcal{F}_Q; S) = \widehat{\mathrm{Rad}}(\mathcal{F}_Q(R); S) \leq \mathbb{E}_\varepsilon \left[ \sup_{|w| \leq Q,\ |b| \leq QR} \left| \frac{1}{n} \sum_{i=1}^n \varepsilon_i\, \sigma(w \cdot x_i + b) \right| \right] + \frac{QR}{\sqrt{n}}$$

We verify this from the definition of Rademacher Complexity of $\mathcal{F}_Q(R)$. Note that $a\sigma(wx + b) = (\lambda a)\,\sigma((w/\lambda)x + b/\lambda)$. In particular, we may assume without loss of generality that $|a|^2 = \|w\|^2 \leq Q$ for optimal balance which makes $a^2 + \|w\|^2$ minimal without changing the neuron output.

$$n\,\mathrm{Rad}(\mathcal{F}_Q(R), S_n) = \mathbb{E}_\epsilon \left[ \sup_{\mathcal{F}_Q} \epsilon_i f(x_i) \right]$$

$$= \mathbb{E}_\epsilon \left[ \sup_{a^2 + \|w\|^2 \leq Q, \, |b| \leq \|w\|R} \sum_{i=1}^n \epsilon_i a \big( \sigma(w \cdot x_i + b) - \sigma(b) \big) \right]$$

$$\leq \mathbb{E}_\epsilon \left[ \sup_{|a| = \|w\| \leq \sqrt{Q}, \, |b| \leq \sqrt{Q}R} \left( \left| \sum_i \epsilon_i a \sigma(w \cdot x_i + b) \right| + \left| \sum_i \epsilon_i a \sigma(b) \right| \right) \right].$$

In this step, we only consider the first term.:

$$\mathbb{E}_\varepsilon \left[ \sup_{|a| = \|w\| \leq \sqrt{Q}, \, |b| \leq \sqrt{Q}R} \left| \sum_i \epsilon_i a \sigma(b) \right| \right] \leq \mathbb{E}_\epsilon \left[ \sup_{|a| \leq \sqrt{Q}, \, |b| \leq \sqrt{Q}R} \left| \sum_i \epsilon_i a \sigma(b) \right| \right]$$

$$\leq \sup_{|a| \leq \sqrt{Q}, \, |b| \leq \sqrt{Q}R} |a| \sigma(b) \, \mathbb{E}_\epsilon \left| \sum_i \epsilon_i \right| \leq QR \sqrt{n}.$$

The first line is again by applying the claim to $\mathcal{F}_Q(R)$. In the last line, we used two facts:

1. $\sigma$ is ReLU, so $|a| \sigma(b) \leq |ab| \leq \sqrt{Q} \cdot \sqrt{Q}R = QR$ and

2. the observation that

$$\mathbb{E}_\epsilon \left| \sum_i^n \epsilon_i \right| \leq \sqrt{ \mathbb{E}_\epsilon \left| \sum_i^n \epsilon_i \right|^2 } = \sqrt{ \sum_{i,j=1}^n \mathbb{E}_\varepsilon [\varepsilon_i \varepsilon_j] } = \sqrt{ \sum_{i=1}^n \mathbb{E}_\varepsilon [\varepsilon_i^2] } = \sqrt{n}$$

since $\mathbb{E}[\varepsilon_i] = 0$, $\varepsilon_i$ and $\varepsilon_j$ are independent if $i \neq j$ and $\varepsilon_i^2 \equiv 1$.

**Step 3.** In this step, we prove that

$$\frac{1}{n} \mathbb{E}_\varepsilon \left[ \sup_{|a| = \|w\| \leq \sqrt{Q}, \, |b| \leq \sqrt{Q}R} \left| \sum_{i=1}^n \varepsilon_i a \sigma(w \cdot x_i + b) \right| \right] \leq \frac{3\sqrt{2}QR}{\sqrt{n}}$$

To this end, we modify the data points as $\tilde{x}_i = (x_i, R)$ and the parameters as $\tilde{w} = (w^T, \frac{b}{R})$. Then, observe that $w \cdot x_i + b = \tilde{w} \cdot \tilde{x}_i$, $\|\tilde{x}_i\| = \sqrt{\|x_i\|^2 + R^2} \leq \sqrt{2}R$, and $a^2 + \|\tilde{w}\|^2 = a^2 + \|w\|^2 + (\frac{b}{R})^2 \leq 3Q$. Therefore, we can write the above by the following:

$$\frac{1}{n} \mathbb{E}_\varepsilon \left[ \sup_{a^2 = \|w\|^2 \leq Q, \, |b| \leq Q} \left| \sum_{i=1}^n \varepsilon_i a \sigma(w \cdot x_i + b) \right| \right] \leq \frac{1}{n} \mathbb{E}_\varepsilon \left[ \sup_{a^2 + \|\tilde{w}\|^2 \leq 3Q} \left| \sum_{i=1}^n \varepsilon_i a \sigma(\tilde{w} \cdot \tilde{x}_i) \right| \right]$$

$$= \frac{1}{n} \mathbb{E}_\varepsilon \left[ \sup_{a^2 + \|\tilde{w}\|^2 \leq 3Q} |a| \, \|\tilde{w}\| \left| \sum_{i=1}^n \varepsilon_i \sigma \left( \frac{\tilde{w}}{\|\tilde{w}\|} \cdot \tilde{x}_i \right) \right| \right]$$

$$\leq \frac{1}{n} \mathbb{E}_\varepsilon \left[ \sup_{a^2 + \|\tilde{w}\|^2 \leq 3Q, \|u\|_2 \leq 1} \frac{a^2 + \|\tilde{w}\|^2}{2} \left| \sum_{i=1}^n \varepsilon_i \sigma(u \cdot \tilde{x}_i) \right| \right]$$

$$\leq \frac{3Q}{2n} \mathbb{E}_\varepsilon \left[ \sup_{\|u\|_2 \leq 1} \left| \sum_{i=1}^n \varepsilon_i \sigma(u \cdot \tilde{x}_i) \right| \right]$$

$$\leq \frac{3Q}{n} \mathbb{E}_\varepsilon \left[ \sup_{\|u\|_2 \leq 1} \sum_{i=1}^n \varepsilon_i \sigma(u \cdot \tilde{x}_i) \right]$$

$$= 3Q \, \widehat{\mathrm{Rad}}(\sigma \circ \mathcal{H}_2, \tilde{S}_n) \leq 3Q \, \widehat{\mathrm{Rad}}(\mathcal{H}_2, \tilde{S}_n) \leq \frac{3Q}{\sqrt{n}} \max_i \|\tilde{x}_i\|_2 \leq \frac{3\sqrt{2} \, QR}{\sqrt{n}}$$

Here, $\mathcal{H}_2 := \{u \in \mathbb{R}^d \mid \|u\|_2 \leq 1\}$ and $\tilde{S}_n = \{\tilde{x}_1, \ldots, \tilde{x}_n\}$. When removing the absolute value, we used that the sum is always non-negative and that it is symmetric when replacing $\varepsilon$ by $-\varepsilon$. When removing $\sigma$, we make use of the Contraction Lemma for Rademacher complexity [Shalev-Shwartz and Ben-David, 2014, Lemma 26.9]. Finally, for $\mathrm{Rad}(\mathcal{H}_2, \tilde{S}_n)$ we used the expression for the Rademacher complexity of the class of linear functions on Hilbert space. [Shalev-Shwartz and Ben-David, 2014, Lemma 26.10]. This concludes Step 3.

**Step 4.** In this step, we finally consider sets $S$ which are sampled from the product measure $\mu^n$, i.e. sets where $x_1, \ldots, x_n$ are independent data samples with law $\mu$. From steps $1-3$, we know that

$$\text{Rad}(\mathcal{F}_Q) = \mathbb{E}_{S_n \sim \mu^n}\left[\widehat{\text{Rad}}(\mathcal{F}, S_n)\right] \leq \frac{(1+3\sqrt{2})Q}{\sqrt{n}}\mathbb{E}_{(x_1, \ldots, x_n) \sim \mu^n}\left[\max_{1 \leq i \leq n} \|x_i\|\right].$$

We bound $\mathbb{E}_{(x_1, \ldots, x_n) \sim \mu^n}\left[\max_{1 \leq i \leq n} \|x_i\|\right]$ by Lemma H.1 to obtain

$$\text{Rad}(\mathcal{F}_Q) \leq \left(1 + 3\sqrt{2}\right)Q\left(\frac{\mathbb{E}_{x \sim \mu}[\|x\|]}{\sqrt{n}} + \sigma\sqrt{2\frac{\log n}{n}}\right) \qquad\qquad \square$$

A similar result follows immediately for the more general function class

$$\mathcal{F}_{A,Q} := \{f \in \mathcal{B} : [f]_{\mathcal{B}} \leq Q, |f(0)| \leq A\}. \tag{5}$$

**Corollary D.3.** *Under the same conditions as Lemma D.2, we have*

$$\text{Rad}(\mathcal{F}_{A,Q}) \leq \left(1 + 3\sqrt{2}\right)Q\left(\frac{\mathbb{E}_{x \sim \mu}[\|x\|]}{\sqrt{n}} + \sigma\sqrt{2\frac{\log n}{n}}\right) + \frac{A}{\sqrt{n}}$$

*Proof.* We note that $f \in \mathcal{F}_{A,Q}$ if and only if $f = \tilde{f} + \alpha$ with $f \in \mathcal{F}_Q$ and $|\alpha| \leq A$. Hence, for any fixed dataset $S$, we have

$$n\widehat{\text{Rad}}_n(\mathcal{F}_{A,Q}) = \mathbb{E}_\varepsilon\left[\sup_{f \in \mathcal{F}_{A,Q}} \sum_{i=1}^n \varepsilon_i f(x_i)\right] = \mathbb{E}_\varepsilon\left[\sup_{f \in \mathcal{F}_Q, |\alpha| \leq A} \sum_{i=1}^n \varepsilon_i\left(\alpha + f(x_i)\right)\right]$$

$$\leq \mathbb{E}_\varepsilon\left[\sup_{f \in \mathcal{F}_Q} \sum_{i=1}^n \varepsilon_i f(x_i)\right] + \mathbb{E}_\varepsilon\left[\sup_{|\alpha| \leq A} \sum_{i=1}^n \varepsilon_i\alpha\right] \leq \widehat{\text{Rad}}_n(\mathcal{F}_Q) + \frac{A}{\sqrt{n}}$$

by the argument of Step 2 in the proof of Lemma D.2. $\qquad\qquad \square$

A bound on the Rademacher complexity, together with the sub-Gaussian property of the distribution $\mu$, allows us to control the 'generalization gap' in homogeneous Barron spaces.

**Corollary D.4.** *Assume that $\mu$ is a $\sigma^2$-sub-Gaussian distribution on $\mathbb{R}^d$. Let $(X_1, \ldots, X_n)$ be iid random variables with law $\mu$ and $f^*$ a $\mu$-measurable function such that*

$$|f^*(x) - f^*(0)| \leq B_1 + B_2\|x\|$$

*$\mu$-almost everywhere. Let*

$$\widehat{\mathcal{R}}_n(f) = \frac{1}{n}\sum_{i=1}^n |f(X_i) - f^*(X_i)|^2, \qquad \mathcal{R}(f) = \mathbb{E}_{x \sim \mu}\left[|f(x) - f^*(x)|^2\right].$$

*Then with probability at least $1 - 2\delta$ over the random draw of $X_1, \ldots, X_n$, the bound*

$$\sup_{f - f^*(0) \in \mathcal{F}_{A,Q}} \left(\mathcal{R}(f) - \widehat{\mathcal{R}}_n(f)\right) \leq C^*\left((Q + B_2)\left(\mathbb{E}_{x \sim \mu}\|x\| + \sigma^2 + 1\right) + A + B_1\right)^2 \frac{\log(n/\delta)}{\sqrt{n}}$$

*holds for a constant $C^* > 0$ which does not depend on $\delta, Q, d, \mu$ or $n$.*

*Proof.* **Step 1.** From Lemma H.2, with probability at least $1 - \delta$ we have

$$\max_{1 \leq i \leq n} \|X_i\| \leq \mathbb{E}_{x \sim \mu}[\|x\|] + \sigma\sqrt{2\log(n/\delta)}.$$

We denote $R_n := \mathbb{E}_{x \sim \mu}[\|x\|] + \sigma\sqrt{2\log(n/\delta)}$ for simplicity.

**Step 2.** Consider the modified loss function

$$\ell_\xi(f) = \min\left\{f^2, \xi^2\right\},$$

which is bounded by $\xi^2$ and satisfies $|\partial_f \ell_\xi| \le 2R$, i.e. $\ell_\xi$ is $2\xi$-Lipschitz continuous. We thus observe that, with probability at least $1 - \delta$ over the choice of random set $S = \{x_1, \ldots, x_n\}$, we have

$$\mathbb{E}_{x \sim \mu}\big[\ell_\xi\big(f(x) - f^*(x)\big)\big] - \frac{1}{n}\sum_{i=1}^{n}\ell_\xi\big(f(x_i) - f^*(x_i)\big) \le 4\xi\,\mathbb{E}\big[\widehat{\mathrm{Rad}}(\mathcal{F}_Q, S_n)\big] + \xi^2\sqrt{\frac{2\log(2/\delta)}{n}}$$

by [Shalev-Shwartz and Ben-David, 2014, Theorem 26.5] and the Contraction Lemma for Rademacher complexities, [Shalev-Shwartz and Ben-David, 2014, Lemma 26.9]. In particular

$$\mathbb{E}_{x \sim \mu}\big[\ell_\xi\big(f(x) - f^*(x)\big)\big] \le \frac{1}{n}\sum_{i=1}^{n}\ell_\xi\big(f(x_i) - f^*(x_i)\big) + \xi^2\sqrt{\frac{2\log(2/\delta)}{n}}$$
$$+ 4(1 + 3\sqrt{2})Q\xi\left(\frac{\mathbb{E}_{x \sim \mu}\|x\|}{\sqrt{n}} + \sigma\sqrt{\frac{2\log n}{n}}\right) + \frac{A\xi}{\sqrt{n}}.$$

**Step 3.** By the union bound, with probability at least $1 - 2\delta$, both the norm bound of Step 1 and the generalization bound of Step 2 hold. In the following, we assume that both bounds hold. Note that
$$|f(x) - f^*(x)| \le |f(x) - f^*(0)| + |f^*(x) - f^*(0)| \le (A + B_1) + (Q + B_2)\|x\|.$$
In particular, if $\xi \ge (A + B_1) + (Q + B_2)R$, then $|f(x) - f^*(x)| \le \xi$ on $B_R(0)$, so $\ell_\xi(f(x) - f^*(x)) \equiv \ell(f(x) - f^*(x))$. Applying the generalization bound with $R_n = \mathbb{E}_{x \sim \mu}\|x\| + \sigma\sqrt{2\log(n/\delta)}$ and $\xi_n = (Q + B_2)R_n + (A + B_1)$, we find that $\ell_{\xi_n}(f(x_i) - f^*(x_i)) = \ell(f(x_i) - f^*(x_i))$ for all $i$ by assumption and thus, with probability at least $1 - 2\delta$, we have

$$\mathbb{E}_{x \sim \mu}\big[\ell_\xi\big(f(x) - f^*(x)\big)\big] \le \frac{1}{n}\sum_{i=1}^{n}\big(f(x_i) - f^*(x_i)\big)^2 + \big((Q + B_2)R_n + A + B_1\big)^2\sqrt{\frac{2\log(2/\delta)}{n}}$$
$$+ 4(1 + 3\sqrt{2})Q\big((Q + B_2)R_n + A + B_1\big)\left(\frac{\mathbb{E}_{x \sim \mu}\|x\|}{\sqrt{n}} + \sigma\sqrt{\frac{2\log n}{n}}\right).$$

**Step 4.** Finally, we bound the population risk with the true loss function rather than $\ell_\xi$. Thus we find that for $B_R := B_R(0)$ we have

$$\mathbb{E}_{x \sim \mu}\big[\big(f(x) - f^*(x)\big)^2\big] = \mathbb{E}_{x \sim \mu}\left[\big(f(x) - f^*(x)\big)^2 \mathbf{1}_{B_R}\right] + \mathbb{E}_{x \sim \mu}\left[\big(f(x) - f^*(x)\big)^2 \mathbf{1}_{\mathbb{R}^d \setminus B_R}\right]$$
$$\le \mathbb{E}_{x \sim \mu}\big[\ell_\xi\big(f(x) - f^*(x)\big)\big] + \mathbb{E}_{x \sim \mu}\big[(A + B_1 + (Q + B_2)\|x\|)^2 \mathbf{1}_{\mathbb{R}^d \setminus B_R}\big]$$

for $R \ge 3$. From Lemma H.3 with $R_n = \mathbb{E}\|x\| + \sigma\sqrt{2\log(n/\delta)}$, we have

$$\mathbb{E}_{x \sim \mu}\big[\|x\|^2 \mathbf{1}_{B_{R_n}(0)^c}(x)\big] \le \sqrt{2\pi}\exp\left(-\frac{\log(n/\delta)}{2}\right)\big((\mathbb{E}\|x\|)^2 + 2\sigma^2\big) = \sqrt{2\pi}\frac{(\mathbb{E}\|x\|)^2 + 2\sigma^2}{\sqrt{n/\delta}}.$$
$\square$

**Remark D.5.** *In particular, Corollary D.4 applies if the target function $f^*$ is Lipschitz-continuous with $B_1 = 0$ and $B_2 = [f^*]_{Lip} \le [f^*]_{\mathcal{B}}$. However, continuity is not necessary, and even noisy labels would be admissible. We do not pursue this generality here.*

## E  Proofs of the convergence theorems

In this appendix, we present the proofs of Theorems 2.1 and 3.3. In these, we combine the upper bound of the Rademacher complexity of the unit ball in homogeneous Barron space in form of the generalization bound of Corollary D.4 with a $\Gamma$-convergence argument (to guarantee the convergence of minimizers to minimizers). The main ingredients in the proof of $\Gamma$-convergence are

- the compact embedding theorem for homogeneous Barron space (to guarantee that every subsequence of $f_n$ has a convergent subsequence) and
- the direct approximation theorem for homogeneous Barron space (to obtain a bound on the lowest achievable energy using a neural network with $m$ neurons).

We first present convergence proofs in $L^p(\mu)$ in Section E.1, followed by proofs of $\Gamma$-convergence in Section E.2. We combine the arguments to prove the statements from the main body of the document in Section E.3.

## E.1 Convergence in $L^p(\mu)$

We start by establishing convergence in $L^2(\mu)$ at an explicit convergence rate. Then, using this $L^2(\mu)$ convergence we will extend this result to general $L^p(\mu)$.

We introduce one of our main theorem of this section, which gives us an explicit bound of $L^2(\mu)$-loss. For convenience, we denote $\theta := (a, W, b)$ for the rest of the section.

**Theorem E.1** ($L^2$-convergence). *Let $\hat\theta \in \operatorname{argmin}_\theta \widehat{\mathcal{R}}_{n,m,\lambda}(\theta)$. If $\delta \geq e^{-n}$, and $f^* \in \mathcal{F}_{Q^*}$, then with probability at least $1 - 4\delta$ over the choice of random points $x_1, \ldots, x_n$ we have*

$$\mathcal{R}(f_{\hat\theta}) \leq C\left(\frac{(Q^*)^2}{m}\left(\mathbb{E}\big[\|x\|^2\big]\right) + \lambda Q^* + Q^*\left(\mathbb{E}\|x\| + \sigma^2 + [f^*]_{\mathcal{B}}\right)\frac{\log(n/\delta)}{\sqrt{n}}\right)$$

*up to higher order terms in the small quantities $(\lambda m)^{-1}, m^{-1}, n^{-1/2}\log n$.*

*Proof.* **Outline.** We use Theorem C.5 for $L^2(\mu_n)$ with $f = f^*$ to obtain a function for which $\widehat{\mathcal{R}}_{n,m,\lambda}$ is low. The empirical risk minimizer (ERM) has even lower risk. The weight decay penalty additionally provides a norm-bound in homogeneous Barron space for the ERM, and we use Corollary D.4 to control the generalization gap.

**Step 1.** Due to Theorem C.5, there exists a $\tilde\theta := (\tilde a, \tilde w, \tilde b) \in \mathbb{R}^m \times \mathbb{R}^{m \times d} \times \mathbb{R}^{m+1}$ such that

$$R_{WD}(\tilde\theta) \leq [f^*]_{\mathcal{B}} \tag{6}$$

and

$$\widehat{\mathcal{R}}_n(f_{\tilde\theta}) = [f_{\tilde\theta} - f^*]^2_{L^2(\mu_n)} \leq \frac{4[f^*]^2}{m}\sup_{\|w\|=1}\int_{\mathbb{R}^d}|w^T x|^2 d\mu_n \leq \frac{4[f^*]^2}{m}\left(\frac{1}{n}\sum_{i=1}^{n}\|x_i\|^2\right).$$

We will always consider $\delta$ such that $\log(1/\delta) \leq n$. In this regime, plugging-in the bound on the second moments of $\mu_n$ from Lemma H.4 gives the corresponding bound

$$\widehat{\mathcal{R}}_n(f_{\tilde\theta}) \leq \frac{4[f^*]^2}{m}\left(\mathbb{E}[\|x\|^2] + 8\sigma^2\sqrt{\frac{\log(1/\delta)}{n}}\right) \tag{7}$$

with probability $1 - \delta$. We will assume that this estimate is valid for the remainder of the proof. In particular, since $\hat\theta$ minimizes $\widehat{\mathcal{R}}_{n,m,\lambda}$, we find that

$$\widehat{\mathcal{R}}_{n,m,\lambda}(\hat\theta) \leq \widehat{\mathcal{R}}_{n,m,\lambda}(\tilde\theta) \leq \frac{4[f^*]^2}{m}\left(\mathbb{E}[\|x\|^2] + 8\sigma^2\sqrt{\frac{\log(1/\delta)}{n}}\right) + \lambda[f^*]_{\mathcal{B}}. \tag{8}$$

**Step 2.** Next, we bound $[f_{\hat\theta}]_{\mathcal{B}}$ and $|f_{\hat\theta}(0) - f^*(0)|$ ($Q$ and $A$ in Corollary D.4). We first bound the Barron semi-norm by

$$[f_{\hat\theta}]_{\mathcal{B}} \leq \frac{1}{\lambda}\widehat{\mathcal{R}}_{n,m,\lambda}(\hat\theta) \leq \frac{1}{\lambda}\widehat{\mathcal{R}}_{n,m,\lambda}(\tilde\theta).$$

Moving on to bounding $A$, we find from the empirical risk bound

$$\min_{1\leq i\leq n}|f - f^*|(x_i) = \sqrt{\min_{1\leq i\leq n}|f - f^*|^2(x_i)} \leq \sqrt{\frac{1}{n}\sum_{i=1}^{n}|f - f^*|(x_i)} \leq \sqrt{\widehat{\mathcal{R}}_n(f)}$$

and in particular

$$\min_{1\leq i\leq n}|f_{\hat\theta} - f^*|(x_i) \leq \sqrt{\widehat{\mathcal{R}}_{n,m,\lambda}(\tilde\theta)}.$$

With probability at least $1 - \delta$, we have

$$\max_{1\leq i\leq n}\|x_i\| \leq \mathbb{E}_{x\sim\mu}\big[\|x\|\big] + \sigma\sqrt{2\log(n/\delta)}.$$

by Lemma H.2. Again, we assume that the estimate holds in the following. Hence, the index $i$ for which the minimum is attained in (8) satisfies the bound

$$\|x_i\| \leq \mathbb{E}_{x\sim\mu}\big[\|x\|\big] + \sigma\sqrt{2\log(n/\delta)}.$$

Combining the bounds on $|f_{\hat{\theta}}(x_i) - f^*(x_i)|$ and the Lipschitz constants of $f_{\hat{\theta}}, f^*$, we find that

$$
\begin{aligned}
|f_{\hat{\theta}} - f^*|(0) &\le |f_{\hat{\theta}} - f^*|(x_i) + \big([f_{\hat{\theta}}]_{\mathcal{B}} + [f^*]_{\mathcal{B}}\big)\,\|x_i\| \\
&\le \sqrt{\widehat{\mathcal{R}}_{n,m,\lambda}(\tilde{\theta})} + \left([f^*]_{\mathcal{B}} + \frac{1}{\lambda}\,\widehat{\mathcal{R}}_{n,m,\lambda}(\tilde{\theta})\right)\left(\mathbb{E}_{x \sim \mu}\big[\|x\|\big] + \sigma\,\sqrt{2\,\log(n/\delta)}\right).
\end{aligned}
$$

**Step 3.** Comparing $\hat{\theta}$ to $\tilde{\theta}$, we observe that

$$
\begin{aligned}
\mathcal{R}(f_{\hat{\theta}}) &= \widehat{\mathcal{R}}_n(f_{\hat{\theta}}) + \mathcal{R}(f_{\hat{\theta}}) - \widehat{\mathcal{R}}_n(f_{\hat{\theta}}) \\
&\le \widehat{\mathcal{R}}_{n,m,\lambda}(\hat{\theta}) + \mathcal{R}(f_{\hat{\theta}}) - \widehat{\mathcal{R}}(f_{\hat{\theta}}) \\
&\le \widehat{\mathcal{R}}_{n,m,\lambda}(\tilde{\theta}) + \mathcal{R}(f_{\hat{\theta}}) - \widehat{\mathcal{R}}_n(f_{\hat{\theta}})
\end{aligned}
$$

where we used the fact that $\hat{\theta}$ is a minimizer of $\widehat{\mathcal{R}}_{n,m,\lambda}(\theta)$ and (6). In the following, we use the bound on $[f_{\hat{\theta}}]_{\mathcal{B}}$ to control the generalization gap.

**Step 4.** Recall that $[f_{\hat{\theta}}]_{\mathcal{B}} \le [f^*]_{\mathcal{B}} + \frac{1}{\lambda}\,\widehat{\mathcal{R}}_{n,m,\lambda}(f_{\tilde{\theta}}) = [f^*]_{\mathcal{B}} + O((\lambda m)^{-1})$. Thus, with probability at least $1 - 2\delta$, we obtain the bound

$$
\big(\mathcal{R} - \widehat{\mathcal{R}}_n\big)(f_{\hat{\theta}}) \le \left([f^*]_{\mathcal{B}} + \frac{1}{\lambda}\,\widehat{\mathcal{R}}_{n,m,\lambda}(\hat{\theta})\right)\left(\mathbb{E}\|x\| + \sigma^2 + [f^*]_{\mathcal{B}}\right)\frac{\log(n/\delta)}{\sqrt{n}}
$$

from Corollary D.4 for a slightly modified constant $C > 0$ (with $B_1 = 0$) and up to higher order terms in $(\lambda m)^{-1}$ and $n$.

**Step 5.** By the union bound, all probabilistic bounds hold simultaneously with probability at least $1 - 4\delta$. In this case

$$
\mathcal{R}(f_{\hat{\theta}}) \le C\left(\frac{Q^2}{m}\left(\mathbb{E}\big[\|x\|^2\big] + \sigma^2\,\sqrt{\frac{\log(1/\delta)}{n}}\right) + \lambda Q + [f^*]_{\mathcal{B}}\left(\mathbb{E}\|x\| + \sigma^2 + [f^*]_{\mathcal{B}}\right)\frac{\log(n/\delta)}{\sqrt{n}}\right)
$$

up to higher order terms in $m^{-1}, \log n/\sqrt{n}, (\lambda m)^{-1}$ etc. $\qquad\square$

Since $\mathcal{R}(\theta) = \|f_\theta - f^*\|_{L^2(\mu)}^2$, we can interpret Theorem E.1 as a convergence statement in $L^2(\mu)$ at a suitable rate. The statement generalizes to $L^p$-convergence at a rate.

**Corollary E.2** ($L^p$-convergence). *Let $p \in [1, \infty]$ and $\hat{\theta}$ as in Theorem E.1. Then there exists a constant $\tilde{C} > 0$ depending on $\mathbb{E}\|x\|, \mathbb{E}[\|x\|^2], \sigma^2$ and $p$ such that*

$$
\|f_{\hat{\theta}} - f^*\|_{L^p(\mu)} \le \tilde{C}\left(\widehat{\mathcal{R}}_{n,m,\lambda}(\hat{\theta})^{1/2} + [f^*]_{\mathcal{B}}\right)^{1-1/p}\|f_{\hat{\theta}} - f^*\|_{L^2(\mu)}^{1/p}.
$$

*Proof.* Since $\mu$ is sub-Gaussian, we note that all moments of $\mu$ are finite: $\mathbb{E}[(1 + \|x\|)^q] < \infty$ for all $q \in [1, \infty)$. In particular, if $g$ is a measurable function which satisfies $|g(x)| \le C_g(1 + \|x\|)$ for some $C)g > 0$, then

$$
\|g\|_{L^p(\mu)}^p = \mathbb{E}\big[g \cdot g^{p-1}\big] \le \mathbb{E}\big[g^2\big]^{1/2}\,\mathbb{E}\big[g^{2(p-1)}\big]^{1/2} = \|g\|_{L^2}\|g\|_{L^{2(p-1)}}^{p-1}.
$$

If $g = f_{\hat{\theta}} - f^* \in \mathcal{B}$, then by the continuous embedding $\mathcal{B} \hookrightarrow L^q(\mu)$ we find that

$$
\|f_{\hat{\theta}} - f^*\|_{L^{2(p-1)}(\mu)} \le C\left(|f_{\hat{\theta}} - f^*|(0) + [f_{\hat{\theta}} - f^*]_{\mathcal{B}}\right).
$$

Recall that $|f_{\hat{\theta}} - f^*|(0) \le \widehat{\mathcal{R}}_{n,m,\lambda}(\hat{\theta})^{1/2} + C[f^*]_{\mathcal{B}}$. $\qquad\square$

We note that Corollary E.2 is generally suboptimal. Indeed, for $p \le 2$, the stronger bound

$$
\|f_{\hat{\theta}} - f^*\|_{L^p(\mu)} \le \|f_{\hat{\theta}} - f^*\|_{L^2(\mu)} = O\left(\left(\frac{1}{m} + \lambda + \frac{\log n}{\sqrt{n}}\right)^{1/2}\right)
$$

holds as $L^2(\mu)$ embeds continuously into $L^p(\mu)$.

## E.2 Gamma-expansion of regularized risk functionals

As before, we denote $\theta_n = (a, W, b)_n \in \mathbb{R}^{m_n} \times \mathbb{R}^{m_n \times d} \times \mathbb{R}^{m_n+1}$. Since $\mathcal{R}(f_\theta) = \|f_\theta - f^*\|^2_{L^2(\mu)}$, Theorem E.1 can be taken as a statement that $f_{\hat{\theta}_n} \to f^*$ as $n \to \infty$ in $L^2(\mu)$. However, this does not tell us about the behavior of $f_{\hat{\theta}_n}$ in a $\mu$-null set, i.e. where the distribution $\mu$ provides us no information. This interpolation between known values can be deduced from our next result. We first present a simplified version, in which we assume that we have already taken the limits $m, n \to \infty$ before taking $\lambda \to 0$. We couple the limits $n, m_n, \lambda_n$ below.

We use the notion of $\Gamma$-convergence from the calculus of variations. For a brief introduction, see Appendix B. $\Gamma$-convergence depends on the underlying topology of the space, and we make the following convention: We say that $f_\lambda \xrightarrow{good} f$ if $f_\lambda \to f$ locally uniformly (uniformly on compact sets) and in $L^2(\mu)$. Other definitions are admissible and lead to the same general theory. Since Barron functions grow at most linearly at $\infty$ due to Lipschitz-continuity, we note that this is notion of convergence is generated by a metric

$$d(f, g) = \max_{x \in \mathbb{R}^d} \frac{|f(x) - g(x)|}{1 + \|x\|^2}$$

at least on bounded subsets of Barron space. This suffices for all applications below and spares us from considering $\Gamma$-convergence on more general topological spaces – which is also possible.

**Theorem E.3.** *Let*

$$\mathcal{R}_\lambda : \mathcal{B} \to [0, \infty), \quad \mathcal{R}_\lambda(f) = \|f - f^*\|^2_{L^2(\mu)} + \lambda\,[f]_{\mathcal{B}}.$$

*We denote*

$$F_\lambda : \mathcal{B} \to [0, \infty) \qquad F_\lambda(f) = \frac{\mathcal{R}_\lambda(f)}{\lambda} = \frac{\|f - f_d^*\|^2_{L^2(\mu)}}{\lambda} + [f]_{\mathcal{B}}$$

$$F : \mathcal{B} \to [0, \infty] \qquad F(f) = \begin{cases} [f]_{\mathcal{B}} & \text{if } f = f^* \text{ }\mu\text{-a.e.} \\ +\infty & \text{else} \end{cases}.$$

*Then $\Gamma - \lim_{\lambda \to 0} F_\lambda = F$ with respect to the notion of convergence $\xrightarrow{good}$ defined above.*

Notably, the $\Gamma$-limit of $\mathcal{R}_\lambda$ itself would be zero at all points of interest. Rescaling to consider $F_\lambda$ instead has fits into the framework of $\Gamma$-expansions considered by Braides and Truskinovsky [2008]. Denote

$$\mathcal{F} = \{f \in \mathcal{B} : f \equiv f^* \text{ }\mu\text{-a.e.}\} \tag{9}$$

*Proof.* **Step 1. liminf-inequality.** First consider $f \in \mathcal{F}$ and assume that $\{f_\lambda\}_{\lambda>0}$ is a family of functions such that $f_\lambda \xrightarrow{good} f$.[2] Then by the compactness theorem for Barron functions in coarser topologies (Theorem C.3) we have the following:

$$\liminf_{\lambda \to 0^+} F_\lambda(f_\lambda) \geq \liminf_{\lambda \to 0^+} [f_\lambda]_{\mathcal{B}} \geq [f]_{\mathcal{B}} = F(f).$$

Now assume that $f \notin \mathcal{F}$ and that $f_\lambda \xrightarrow{good} f$. We need to show that $F_\lambda(f_\lambda) \to +\infty$. Since $f \notin \mathcal{F}$, we see that $\mathcal{R}(f) = \|f - f_d^*\|^2_{L^2(\mu)} > 0$. Denote $\varepsilon = \sqrt{\mathcal{R}(f)}$ and observe that there exists $\Lambda > 0$ such that $\|f_\lambda - f\|_{L^2(\mu)} < \varepsilon/2$ for all $\lambda < \Lambda$ by the definition of the notion of convergence.

In particular, we find that

$$\|f_\lambda - f^*\|_{L^2(\mu)} \geq \|f - f^*\|_{L^2(\mu)} - \|f_\lambda - f\|_{L^2(\mu)} \geq \varepsilon/2$$

for all $\lambda < \Lambda$ by the inverse triangle inequality and thus

$$\liminf_{\lambda \to 0^+} F_\lambda(f_\lambda) \geq \liminf_{\lambda \to 0^+} \frac{(\varepsilon/2)^2}{\lambda} = +\infty.$$

---

[2] It is easy to generalize this to continuous limits, but if preferred, then $\lambda = \lambda_n$ can be taken to be a discrete sequence converging to zero.

**Step 2. limsup-inequality.** Again, we first consider the case $f \in \mathcal{F}$. Set $f_\lambda = f$ for all $\lambda > 0$ and observe that $f_\lambda \to f$ as $\lambda \to 0^+$ (trivially). By the same argument $F_\lambda(f_\lambda) = F(f) = [f]_\mathcal{B}$ for all $\lambda$, i.e. the constant sequence is a recovery sequence since $F_\lambda(f_\lambda) \to F(f)$.

On the other hand, if $f \notin \mathcal{F}$, then $\mathcal{R}(f) > 0$ and thus $F_\lambda(f) \to +\infty = F(f)$. Again, we can use the constant sequence as a recovery sequence, somewhat trivially. $\qquad\square$

**Corollary E.4.** *Assume that $f_\lambda \in \operatorname{argmin}_{f \in \mathcal{B}} \mathcal{R}_\lambda$, i.e. $f_\lambda$ minimizes $\mathcal{R}_\lambda$. Then there exists $\hat{f} \in \mathcal{B}$ such that $f_\lambda \xrightarrow{good} \hat{f}$.*

*Proof.* Clearly $f_\lambda$ minimizes $\mathcal{R}_\lambda$ if and only if it minimizes $F_\lambda = \lambda^{-1} \mathcal{R}_\lambda$. We note that

$$[f_\lambda]_\mathcal{B} \le \lambda^{-1} \mathcal{R}_\lambda(f_\lambda) \le \lambda^{-1} \mathcal{R}(f^*) + [f^*]_\mathcal{B} = [f^*]_\mathcal{B}.$$

In particular, by the compact embedding of Theorem C.3, there exists $\hat{f} \in \mathcal{B}$ such that $f_\lambda \xrightarrow{good} \hat{f}$ up to subsequence. By the properties of $\Gamma$-convergence, we conclude that $\hat{f}$ is a minimizer of $F$. $\quad\square$

We present a special case of Corollary E.4 in the setting of Proposition 3.2 which exploits the *uniqueness* of the minimizer. Recall the definition of the radial average in (3) and $f_d^*$ from Proposition 3.2.

**Corollary E.5.** *Assume that $f^*(0) = 1$, $f^*(x) = 0$ if $\|x\| \ge 1$ and $\mu$ satisfies the conditons of Corollary 3.3. Assume additionally that $f_\lambda \in \operatorname{argmin}_{f \in \mathcal{B}} \mathcal{R}_\lambda$, i.e. $f_\lambda$ minimizes $\mathcal{R}_\lambda$. Then $\operatorname{Av} f_\lambda \xrightarrow{good} f_d^*$.*

*Proof.* **Step 1.** Clearly $f_\lambda$ minimizes $\mathcal{R}_\lambda$ if and only if it minimizes $F_\lambda = \lambda^{-1} \mathcal{R}_\lambda$. $\operatorname{Av} f_\lambda$ is also a minimizer of $F_\lambda$ since the functional is convex and rotationally symmetric, so by averaging in radial direction, we are taking a (continuous) convex combination of minimizers, which is a minimizer again.

**Step 2.** We find that

$$[\operatorname{Av} f_\lambda]_\mathcal{B} \le F_\lambda(\operatorname{Av} f_\lambda) \le F_\lambda(f_\lambda) \le F_\lambda(f_d^*) = [f_d^*]_\mathcal{B}.$$

By the compactness theorem for Barron functions, Theorem C.3, there exists $f \in \mathcal{B}$ such that $\operatorname{Av} f_\lambda \xrightarrow{good} f$ (up to a subsequence). Since $F_\lambda \to F$ in the sense of $\Gamma$-convergence, we find that $f$ is a minimizer of $F$. Since $f$ is also radially symmetric, we find by Proposition 3.2 that $f \equiv f_d^*$.

**Step 3.** By the exact same logic, we could show that every subsequence of $\{\operatorname{Av} f_\lambda\}$ has a further subsequence which converges to $f_d^*$. By a standard argument in topology, the whole sequence converges. $\qquad\square$

A similar statement can be proved in the more complicated case where $f_\lambda$ is a neural network with finitely many neurons and $F_\lambda$ uses a finite data set rather than a continuous expectation. In this case, the parameter $\lambda$ must be coupled to the number of parameters $m$ and the number of data points $n$, such that $\lambda \to 0$, but not too quickly. The proof is a more technically challenging variant of those of Theorem E.3 and Corollaries E.4 and E.5, which utilizes the generalization bound of D.4. To this end, we first introduce a new notion of convergence. That is, we define a notion of convergence from the parameter to function. We define a notion of convergence by saying that $\theta_k := (a_k, W_k, b_k) \xrightarrow{good} f$ iff $f_{\theta_k} \xrightarrow{good} f$ as $k \to \infty$.

**Theorem E.6.** *Consider the parameter space $\Theta_m \subseteq \mathbb{R}^m \times \mathbb{R}^{m \times d} \times \mathbb{R}^{m+1}$ of neural networks with a single hidden layer of width $m$ and the associated functions*

$$f_\theta(x) := b_0 + \sum_{i=1}^{m} a_i \sigma(w_i \cdot x + b_i)$$

*Let $m_n, \lambda_n$ scale with $n$ according to (1). We denote*

$$F_n : \Theta_{m_n} \to [0, \infty) \qquad F_n(\theta) = \frac{\widehat{\mathcal{R}}_{n,m_n,\lambda_n}(\theta)}{\lambda_n} = \frac{\widehat{\mathcal{R}}_n(f_\theta)}{\lambda_n} + R_{WD}(\theta)$$

$$F : \mathcal{B} \to [0, \infty] \qquad F(f) = \begin{cases} [f]_\mathcal{B} & \text{if } f = f^* \ \mu\text{-a.e.} \\ +\infty & \text{else} \end{cases}.$$

*Then almost surely over the choice of data points, we have $\Gamma - \lim_{n \to \infty} F_n = F$ almost surely with respect to the notion of convergence $\theta_k \xrightarrow{good} f$ defined above.*

*Proof.* We use $\mathcal{F}$ as in (9) throughout. In the proof we assume that all stochastic quantities in Corollary D.4 and Theorem E.1 are satisfied with probability at least $1 - \delta_n$ for $\delta_n = n^{-2}$. The quantity $\log(\delta_n)$ therefore becomes comparable to $\log n / n \ll \lambda_n$. Since $\sum_{n=1}^\infty n^{-2} < \infty$, we find that all conditions are met for all but finitely many $n \in \mathbb{N}$ by the Borel-Cantelli Lemma. For questions of asymptotic convergence, we may therefore assume that the statements of both Theorems apply without qualifying for high probability. Note that $n^{-2} \geq e^{-n}$ for all $n \geq 2$ as needed for Theorem E.1.

**Step 1. liminf-inequality.** Again, we consider the cases $f \in \mathcal{F}$ and $f \notin \mathcal{F}$ separately. First, when $f \in \mathcal{F}$, we apply the same method we did in Theorem E.3. For any sequence of parameters $\theta_n \xrightarrow{good} f$, by Theorem C.3 the following holds:

$$\liminf_{n \to \infty} F_n(\theta_n) \geq \liminf_{n \to \infty} R_{WD}(\theta_n) \geq \liminf_{n \to \infty} [f_{\theta_n}]_\mathcal{B} \geq [f]_\mathcal{B} = F(f).$$

The second inequality comes from $[f_{\theta_n}]_\mathcal{B}$ being an infimum of weight decay regularizers with any arbitrary probability measure on parameter space.

Second, when $f \notin \mathcal{F}$, we need to show $\liminf_{n \to \infty} F_n(\theta_n) = \infty$ for any $\theta_n \xrightarrow{good} f$. We distinguish two prototypical cases:

1. $[f_{\theta_n}]_\mathcal{B} \to +\infty$ as $n \to \infty$. In this case $F(\theta_n) \geq [f_{\theta_n}]_\mathcal{B} \to +\infty$ as well by the same logic as above.

2. $\limsup_{n \to \infty} [f_{\theta_n}]_\mathcal{B} < +\infty$. In this case, we take $\varepsilon := \|f - f^*\|_{L^2(\mu)}/2 > 0$. Then there exists $N \in \mathbb{N}$ such that for all $n \geq N$ we have
$$\|f_{\theta_n} - f^*\|_{L^2(\mu)} \geq \|f - f^*\|_{L^2(\mu)} - \|f_{\theta_n} - f\|_{L^2(\mu)} \geq \varepsilon$$
for all $n \geq N$ by definition. Additionally
$$F_n(\theta_n) \geq \frac{\widehat{\mathcal{R}}_n(f_{\theta_n}) - \mathcal{R}(f_{\theta_n})}{\lambda_n} + \frac{\mathcal{R}(f_{\theta_n})}{\lambda_n} + [f_{\theta_n}]_\mathcal{B}$$
$$\geq \frac{\widehat{\mathcal{R}}_n(f_{\theta_n}) - \mathcal{R}(f_{\theta_n})}{\lambda_n} + \frac{\varepsilon^2}{\lambda_n} + [f_{\theta_n}]_\mathcal{B}.$$
Due to Corollary D.4 and the arguments of Theorem E.1 to control the discrepancy at 0, we have
$$\widehat{\mathcal{R}}_n(f_{\theta_n}) - \mathcal{R}(f_{\theta_n}) = O\left(\frac{\log n}{\sqrt{n}}\right).$$
in this case. Since $\log n / \sqrt{n} \ll \lambda_n$ by assumption, we note that
$$\lim_{n \to \infty} \frac{\widehat{\mathcal{R}}_n(f_{\hat\theta_n}) - \mathcal{R}(f_{\hat\theta_n})}{\lambda_n} = 0$$
and thus
$$\lim_{n \to \infty} F_n(\theta_n) \geq \liminf_{n \to \infty} \left(0 + \frac{\varepsilon^2}{\lambda_n} + 0\right) = +\infty.$$

The same holds in the general case by passing to subsequences.

**Step 2. limsup-inequality.** As in Theorem C.5, the case $f \notin \mathcal{F}$ follows from the $\liminf$-inequality in an essentially trivial fashion. We therefore only consider the case $f \in \mathcal{F}$. An approximating sequence in this case is constructed from Theorem C.5 as in Theorem E.1 or Theorem E.3. Namely, we find $\tilde\theta_n$ such that

$$F_n(\tilde\theta_n) \leq \frac{C}{\lambda_n m_n}\left(1 + \frac{\log n}{\sqrt{n}}\right) + [f^*]_\mathcal{B} \qquad \Rightarrow \qquad \limsup_{n \to \infty} F_n(\tilde\theta_n) \leq [f^*]_\mathcal{B}. \qquad \square$$

**Remark E.7.** *The key ingredients for the proofs of both Theorem E.1 and Theorem E.6 are Theorem C.5 and Corollary D.4, but they are combined differently. While they are paired in Theorem E.1 to obtain a precise rate, they occur separately in Theorem E.6: Theorem C.5 is used for the* $\lim\sup$*-inequality while Corollary D.4 enters in the proof of the* $\lim\inf$*-inequality. Analogously, the condition* $\lambda_n \ll \log n/\sqrt{n}$ *is used in the proof of the* $\lim\inf$*-condition while the fact that* $\frac{1}{m_n} \ll \lambda_n$ *is used in the proof of the* $\lim\sup$*-inequality.*

### E.3   Proofs of the main theorems

The statements of the Theorems in the main body of the text can easily be deduced from the statements proved in this Appendix.

*Proof of Theorem 2.1.* Convergence in $L^p$ holds by Theorem E.1 for $1 \leq p \leq 2$ and Corollary E.2 (general $p$). The proof of uniform convergence follows from Theorem E.6 in the same fashion that Corollary E.4 follows from Theorem E.3. The explicit bound is obtained from Theorem E.1 with $\delta_n = \frac{1}{4n^2}$. $\qquad\square$

*Proof of Corollary 3.3.* This follows in the same way as the proof of Theorem 2.1 with modifications as in Corollary E.5. $\qquad\square$

## F   Theorem 2.1 for finite data sets

Finally, we note that a version of Theorem 2.1 holds if the data set $S = \{x_1, \ldots, x_n\}$ is kept fixed. The proof is a combination of those of Theorems E.3 and E.6, as we deal with a finite approximating neural network, but do not require generalization bounds. The details are left to the reader.

**Theorem F.1.** *We make the following assumptions.*

1. *Let $S = \{(x_1, y_1), \ldots, (x_n, y_n)\}$ be a fixed dataset of $n$ data points in $x_i \in \mathbb{R}^d$ and labels $y_i \in \mathbb{R}$.*

2. *Let the loss function $\ell(f, y)$ be the mean squared error $\ell_{MSE}(f, y) = |f - y|^2$.*

3. *Assume that $\lambda_m$ is a sequence of parameters such that $\lambda_m \to 0$, $1/m \ll \lambda_m$ as $m \to \infty$.*

*Consider the regularized empirical risk functional $\widehat{\mathcal{R}}_m : \mathbb{R}^m \times \mathbb{R}^{m \times d} \times \mathbb{R}^m \to [0, \infty)$,*

$$\widehat{\mathcal{R}}_m(a, W, b) = \frac{1}{2n} \sum_{i=1}^n \ell\left(f_{(a,W,b)}(x_i),\ y_i\right) + \frac{\lambda_m}{2}\left(\|a\|_2^2 + \|W\|_{Frob}^2\right).$$

*Then if $(a, W, b)_m \in \arg\min \widehat{\mathcal{R}}_m$ for all $m \in \mathbb{N}$, then every subsequence of $f_m := f_{(a,W,b)_m}$ has a further subsequence which converges to some limit $\hat{f}^* \in \mathcal{B}$ uniformly on compact subset of $\mathbb{R}^d$. The limiting function satisfies*

$$\hat{f}^* \in \underset{\{f \in \mathcal{B} : f(x_i) = y_i\ \forall i\}}{\arg\min}\ [f]_{\mathcal{B}}.$$

## G   Minimum norm interpolation in one dimension

*Proof of Proposition 3.1.* Any Barron function $f$ is also Lipschitz-continuous, in particular differentiable almost everywhere and $f(b) - f(a) = \int_a^b f'(x)\,\mathrm{d}x$ for all $a, b \in \mathbb{R}$ by Rademacher's Theorem (see e.g. [Evans and Gariepy, 2015, Section 3.1]). In particular, there exist points $x^+, x^- \in (a, b)$ such that

$$f'(x^-) \leq \frac{f(b) - f(a)}{b - a} = \frac{1}{b - a}\int_a^b f'(x)\,\mathrm{d}x \leq f'(x^+).$$

If $f'$ is not constant, both inequalities are satisfied strictly. Under the assumptions, there exists $a \in (x_0, x_1)$ and $b \in (x_{n-1}, x_n)$ such that

$$f'(a) \leq \frac{f(x_1) - f(x_0)}{x_1 - x_0} \leq 0 \leq \frac{f(x_n) - f(x_{n-1})}{x_n - x_{n-1}} \leq f'(b). \tag{10}$$

Since the derivative $f'$ changes sign, we conclude by [Wojtowytsch, 2022, Proposition 2.5] that $[f]_\mathcal{B} = \int_{-\infty}^{\infty} \mathrm{d}|\mu|$ where the Radon measure $\mu$ is the distributional derivative of $f'$ and $|\mu|$ denotes the total variation measure of $\mu$. Since $f'$ is differentiable at $a, b$, neither point is an atom of $\mu$ and thus

$$\frac{f(x_n) - f(x_{n-1})}{x_n - x_{n-1}} - \frac{f(x_1) - f(x_0)}{x_1 - x_0} \leq f'(b) - f'(a) \leq \int_a^b \mathrm{d}\mu \leq \int_a^b \mathrm{d}|\mu| \leq [f]_\mathcal{B}.$$

Equality holds if and only if $|\mu| = \mu$, i.e. if $\mu \geq 0$ in the sense of signed measures and if and only if the inequality in (10) can only be satisfied with equality. The first condition means that $f$ must be convex, the second implies that the derivative of $f$ must be constant in the intervals $(x_0, x_1)$ and $(x_{n-1}, x_n)$. The same can easily be seen for the larger intervals $(-\infty, x_1)$ and $(x_{n-1}, \infty)$. $\qquad\square$

## H  Sub-Gaussian random variables

In this Appendix we quickly gather some facts about sub-Gaussian random variables. For the reader's convenience, we provide full proofs. Experts are encouraged to skip ahead to Appendix D.

The first property we observe is a bound of expected maximum value of sub-Gaussian.

**Lemma H.1.** *Let $\mu$ be $\sigma^2$-sub-Gaussian, and for $i = 1, \ldots, n$, let $x_i$ be a iid random samples from a law $\mu$. Then, the following holds:*

$$\mathbb{E}\big[\max_{1 \leq i \leq n} \|x_i\|\big] \leq \mathbb{E}_{x \sim \mu}\big[\|x\|\big] + \sqrt{2 \log n}\, \sigma.$$

*Proof.* From the sub-Gaussian assumption we have

$$\log\big(\mathbb{E}_{x \sim \mu}\big[\exp\big(\lambda(\|X\| - \mathbb{E}[\|X\|])\big)\big]\big) \leq \frac{\sigma^2 \lambda^2}{2}$$

for some fixed $\sigma > 0$ and all $\lambda > 0$. In particular, Jensen's inequality implies the sub-Gaussian maximal inequality

$$\mathbb{E}\big[\max_{1 \leq i \leq n} \|x_i\|\big] = \mathbb{E}_{x \sim \mu}\big[\|x\|\big] + \mathbb{E}\big[\max_{1 \leq i \leq n}\big(\|x_i\| - \mathbb{E}\|x_i\|\big)\big]$$

$$\leq \mathbb{E}_{x \sim \mu}\big[\|x\|\big] + \frac{1}{\lambda} \log\left(\mathbb{E}\left[\exp\left(\lambda \max_{1 \leq i \leq n}\big(\|x_i\| - \mathbb{E}\|x_i\|\big)\right)\right]\right)$$

$$\leq \mathbb{E}_{x \sim \mu}\big[\|x\|\big] + \frac{1}{\lambda} \log\left(\mathbb{E}\left[\sum_{i=1}^n \exp\big(\lambda(\|x_i\| - \mathbb{E}\|x_i\|)\big)\right]\right)$$

$$\leq \mathbb{E}_{x \sim \mu}\big[\|x\|\big] + \frac{1}{\lambda} \log\left(\sum_{i=1}^n \mathbb{E}\big[\exp\big(\lambda(\|x_i\| - \mathbb{E}\|x_i\|)\big)\big]\right)$$

$$= \mathbb{E}_{x \sim \mu}\big[\|x\|\big] + \frac{1}{\lambda} \log\left(n \exp\left(\frac{\lambda^2 \sigma^2}{2}\right)\right)$$

$$\leq \mathbb{E}_{x \sim \mu}\big[\|x\|\big] + \frac{\log n}{\lambda} + \frac{\sigma^2}{2} \lambda$$

for all $\lambda > 0$. We specifically select $\lambda = \sqrt{\frac{2 \log n}{\sigma^2}}$, making the bound

$$\mathbb{E}\big[\max_{1 \leq i \leq n} \|x_i\|\big] \leq \mathbb{E}_{x \sim \mu}\big[\|x\|\big] + \sqrt{2 \log n}\, \sigma. \qquad\square$$

Next, we observe the concentration of maximum values among samples of sub-Gaussian distribution.

**Lemma H.2.** *Let $\mu$ be $\sigma^2$-sub-Gaussian, and for $i = 1, \ldots, n$, let $x_i$ be a iid random samples from a law $\mu$. Then, with probability at least $1 - \delta$, the following holds:*

$$\max_{1 \leq i \leq n} \|x_i\| \leq \mathbb{E}_{x \sim \mu}\big[\|x\|\big] + \sigma \sqrt{2 \log(n/\delta)}.$$

*Proof.* For all $t > 0$, we observe that

$$\mu^n \left( \max_{1 \le i \le n} \|x_i\| \ge \mathbb{E}_{x \sim \mu}[\|x\|] + t \right) = \mu^n \left( \max_{1 \le i \le n} \exp\left(\lambda(\|x_i\| - \mathbb{E}\|x_i\|)\right) \ge \exp(\lambda t) \right)$$

$$\le e^{-\lambda t} \mathbb{E}\left[ \max_{1 \le i \le n} \exp\left((\lambda(\|x_i\| - \mathbb{E}\|x_i\|))\right) \right]$$

$$\le e^{-\lambda t} \sum_{i=1}^n \mathbb{E}\left[ \exp\left((\lambda(\|x_i\| - \mathbb{E}\|x_i\|))\right) \right]$$

$$\le n \exp\left( -\lambda t + \frac{\lambda^2 \sigma^2}{2} \right).$$

For fixed $t$, the bound becomes tightest for $\lambda = t/\sigma^2$ with

$$\mu^n \left( \max_{1 \le i \le n} \|x_i\| \ge \mathbb{E}_{x \sim \mu}[\|x\|] + t \right) \le n \exp\left( -\frac{t^2}{2\sigma^2} \right) \le \delta$$

if

$$-\frac{t^2}{2\sigma^2} \le \log\left(\frac{\delta}{n}\right) \qquad \Leftrightarrow \qquad t \ge \sigma \sqrt{2 \log(n/\delta)}.$$

Thus with probability at least $1 - \delta$, we have

$$\max_{1 \le i \le n} \|X_i\| \le R_n := \mathbb{E}_{x \sim \mu}[\|x\|] + \sigma \sqrt{2 \log(n/\delta)}. \qquad \square$$

In the following Lemma, we investigate expectation of squared norm of sub-Gaussian near the tail.

**Lemma H.3.** *Let $\mu$ be $\sigma^2$-sub-Gaussian distribution in $\mathbb{R}^d$. For any $R > \mathbb{E}_{x \sim \mu}\|x\|$, we have the following:*

$$\mathbb{E}_{x \sim \mu}\left[ \|x\|^2 \mathbb{1}_{B_R(0)^c}(x) \right] \le \exp\left( -\frac{(R - \mathbb{E}\|x\|)^2}{4\sigma^2} \right) \sqrt{2\pi} \left( (\mathbb{E}\|x\|)^2 + 2\sigma^2 \right).$$

*Proof.* Recall that

$$\mu \left( \{ x : \|x\| \ge (\mathbb{E}_{x' \sim \mu}\|x'\| + t) \} \right) \le \exp\left( -\frac{t^2}{2\sigma^2} \right)$$

as demonstrated in the proof of Lemma H.1 (consider $n = 1$). Thus

$$\mathbb{E}_{x \sim \mu}\left[ \|x\|^2 \mathbb{1}_{B_R(0)^c}(x) \right] \le \int_R^\infty s^2 \mu(\{\|x\| \ge s\}) \, \mathrm{d}s \le \int_R^\infty s^2 \exp\left( -\frac{(s - \mathbb{E}\|x\|)^2}{2\sigma^2} \right) \, \mathrm{d}s$$

$$\le \exp\left( -\frac{(R - \mathbb{E}\|x\|)^2}{4\sigma^2} \right) \int_R^\infty \exp\left( -\frac{(s - \mathbb{E}\|x\|)^2}{4\sigma^2} \right) \, \mathrm{d}s$$

$$\le \exp\left( -\frac{(R - \mathbb{E}\|x\|)^2}{4\sigma^2} \right) \int_R^\infty \exp\left( -\frac{(s - \mathbb{E}\|x\|)^2}{4\sigma^2} \right) \, \mathrm{d}s$$

$$= \exp\left( -\frac{(R - \mathbb{E}\|x\|)^2}{4\sigma^2} \right) \sqrt{2\pi} \left( (\mathbb{E}\|x\|)^2 + 2\sigma^2 \right).$$

$$\square$$

In next Lemma we introduce how the mean of squared norm of sub-Gaussian is concentrated.

**Lemma H.4.** *Assume $\mu$ is a $\sigma^2$-sub-Gaussian distribution in $\mathbb{R}^d$. Let $x_1, \ldots, x_n$ be iid random samples from law $\mu$. Then, with probability at least $1 - \delta$,*

$$\frac{1}{n} \sum_{i=1}^n \|x_i\|^2 \le \mathbb{E}_{x \sim \mu}[\|x\|^2] + 8\sigma^2 \max\left( \frac{\log(1/\delta)}{n}, \sqrt{\frac{\log(1/\delta)}{n}} \right).$$

*In particular, if $\delta \geq e^{-n}$ then*

$$\frac{1}{n} \sum_{i=1}^{n} \|x_i\|^2 \leq \mathbb{E}\big[\|x\|^2\big] + 8\sigma^2 \sqrt{\frac{\log(1/\delta)}{n}}$$

*with probability at least $1 - \delta$.*

*Proof.* Firstly since $\|x_i\|$ is $\sigma^2$-sub-Gaussian, from [Honorio and Jaakkola, 2014, Appendix B] we observe that $\|x_i\|^2$ is $(4\sqrt{2}\sigma^2, 4\sigma^2)$-sub-exponential. Next, by independence, for all $\lambda > 0$ and for all $|t| \leq \frac{1}{4\sigma^2}$ we have the following:

$$
\begin{aligned}
\mathbb{E}\left[\exp\left(\lambda\big(\sum_{i=1}^{n} \|x_i\|^2 - \mathbb{E}\big[\sum_{i=1}^{n} \|x_i\|^2\big]\big)\right)\right] &= \prod_{i=1}^{n} \mathbb{E}\left[\exp\left(\lambda\big(\|x_i\|^2 - \mathbb{E}\big[\|x_i\|^2\big]\big)\right)\right] \\
&\leq \prod_{i=1}^{n} \exp\left(\frac{32\sigma^4\lambda^2}{2}\right) \\
&= \exp\left(\frac{32n\sigma^4\lambda^2}{2}\right)
\end{aligned}
$$

which implies that $\sum_{i=1}^{n} \|x_i\|^2$ is $(4\sqrt{2n}\sigma^2, 4\sigma^2)$-sub-exponential. Thus the tail bound of sub-exponential [Adams, 2022, Proposition 2.38] applied to $\sum_{i=1}^{n} \|x_i\|^2$ yields

$$\mu^n\left(|\sum_{i=1}^{n} \|x_i\|^2 - \mathbb{E}\big[\sum_{i=1}^{n} \|x_i\|^2\big] \geq s\right) \leq \exp\left(-\frac{1}{2}\min\left(\frac{s^2}{32n\sigma^4}, \frac{s}{4\sigma^2}\right)\right)$$

Plugging-in $s = nt$, we have the following:

$$\mu^n\left(\frac{1}{n}\sum_{i=1}^{n} \|x_i\|^2 - \mathbb{E}_{x\sim\mu}\|x\|^2 \geq t\right) \leq \exp\left(-\frac{1}{2}n\min\left(\frac{t^2}{32\sigma^4}, \frac{t}{4\sigma^2}\right)\right)$$

Lastly, take $\delta = \exp\left(-\frac{1}{2}n\min(\frac{t^2}{32\sigma^4}, \frac{t}{4\sigma^2})\right)$. By rearranging the $t$ with respect to $\delta$, we have the following:

$$\mu^n\left(\frac{1}{n}\sum_{i=1}^{n} \|x_i\|^2 - \mathbb{E}_{x\sim\mu}\|x\|^2 \geq 8\sigma^2 \max\left(\sqrt{\frac{\log(1/\delta)}{n}}, \frac{\log(1/\delta)}{n}\right)\right) \leq \delta.$$

This proves the first part of the Lemma. Specifically, we have the following:

$$\mu^n\left(\frac{1}{n}\sum_{i=1}^{n} \|x_i\|^2 - \mathbb{E}_{x\sim\mu}\|x\|^2 \geq 8\sigma^2 \frac{\log(1/\delta)}{n}\right) \leq \delta \quad \text{if } \delta \leq e^{-n}$$

$$\mu^n\left(\frac{1}{n}\sum_{i=1}^{n} \|x_i\|^2 - \mathbb{E}_{x\sim\mu}\|x\|^2 \geq 8\sigma^2 \sqrt{\frac{\log(1/\delta)}{n}}\right) \leq \delta \quad \text{if } \delta \geq e^{-n}$$

The second inequality proves the last part of the Lemma. $\square$

