# OpenReview forum: "Minimum norm interpolation by perceptra: Explicit regularization and implicit bias"
_NeurIPS.cc/2023/Conference — NeurIPS 2023 poster_

### Official Review · Reviewer_hUXj · 2023-06-29

**Soundness:** 3 good
**Presentation:** 3 good
**Contribution:** 2 fair
**Rating:** 4
**Confidence:** 2

**Summary:**

This paper investigates how shallow ReLU networks interpolate between known regions as the number of data points and parameters tends to infinity. The authors prove that minimizers of regularized risk functionals converge to minimum norm interpolants of a given target function in an infinite parameter limit. The authors study the performance of optimization algorithms by comparing numerical solutions to the minimum norm interpolant in cases where the latter is known explicitly.

**Strengths:**

The paper has rigorous theoretical proof.

**Weaknesses:**

The paper does not show the effectiveness for practical neural network optimization, such as CNN and Transformer.

**Questions:**

The paper does not show the effectiveness for practical neural network optimization, such as CNN and Transformer.

---

> ### Author Rebuttal · Authors · 2023-08-08
>
> > The paper does not show the effectiveness for practical neural network optimization, such as CNN and Transformer.
>
> This is true, and it is not the article's goal. While more advanced network architectures are of great practical interest, the theoretical instruments to describe them are not sufficiently developed to achieve comparable results in those cases, as the function space theory is not readily available. As we note in the final paragraph of Section 2 (Line 115 - 117), our methods apply to all function spaces which satisfy a Rademacher bound, a suitable compact embedding and a direct approximation theorem. Spaces with such properties were developed for instance by E and Wojtowytsch in *On the Banach spaces associated with multi-layer ReLU networks* in the context of multi-layer perceptron architectures. Non-linear function classes for infinitely deep ResNets ("neural ODEs") are developed by E et. al.: *The Barron Space and the Flow-Induced Function Spaces for Neural Network Models*. However, even in those established settings, minimum norm interpolants are not known.

---

### Official Review · Reviewer_5DHL · 2023-07-06

**Soundness:** 3 good
**Presentation:** 3 good
**Contribution:** 3 good
**Rating:** 5
**Confidence:** 1

**Summary:**

This paper investigates the behavior of shallow ReLU networks in terms of convergence to minimum norm interpolants of given data. The authors present provable results when explicit regularization is included and empirical evidence when it is not. The paper concludes with a summary of the authors' empirical insights into the implicit bias of neural network optimizers. . The authors suggest that minimum norm interpolation serves as a useful testbed to study the implicit bias of optimizers, initialization, and regularization, and anticipate the extension of this analysis to deeper networks in the future.

**Strengths:**

The authors summarize their empirical insights into the implicit bias of neural network optimizers, providing a concise overview of the behaviors exhibited by various algorithms.

The discussion on the impact of initialization scale, optimization algorithm choice, and explicit regularization is informative and relevant to understanding the behavior of shallow ReLU networks

**Weaknesses:**

Further discussion on the trade-off between fit and regularization, as well as potential methods to balance these factors, would enhance the practical understanding of these observations.
Also, it would be valuable if the authors briefly discussed the potential practical applications or implications of their findings to further motivate researchers in the field.

This work is a bit out of my scope. Please discard my rating for final decision.

**Questions:**

N/A

---

> ### Author Rebuttal · Authors · 2023-08-08
>
> We appreciate the reviewer's feedback and their forthcoming assessment of their level of expertise. In order to make the article accessible and interesting to a broad audience, we will include a more detailed discussion of the fit/regularization trade off and potential implications in the final work, if accepted.
>
> Also, in the context of potential implications, we have outlined some of potential implications we expect in Section 1 of our paper. We anticipate our work to yield insights into understanding how a neural network works outside the support of data distribution (Section 2), which is one of the main interests in the topics of distributional shift and adversarial stability (Line 23 - 27). In addition, numerical experiments we conducted (Section 4, 5) can provide a useful benchmark to study the implicit bias and explicit regularization in optimization, by enabling comparisons between numerical solutions from optimization algorithms and theoretically known minimum norm interpolant (Line 28 - 31).

---

### Official Review · Reviewer_Gs5Q · 2023-07-06

**Soundness:** 3 good
**Presentation:** 3 good
**Contribution:** 3 good
**Rating:** 6
**Confidence:** 4

**Summary:**

Paper summary:
* The paper studies the convergence properties of 1-layer Relu nets trained on a regression task with Tikhonov regularisation. The main contribution of the paper is Theorem 2.1, in which the authors prove that under some assumptions on the data distribution and on the optimisation trajectory (or in the limit at they specify in line 110), the model converges to a function f^* that is equal to the optimal unknown data generating function f^* almost everywhere but  f^* has smaller Barron semi-norm.
 * Assumptions of Theorem 2.1: as stated in the theorem and in the main text, the assumption that at each iteration the function f_n minimises the empirical risk given by the current sample is a strong one, but (line 110) the authors specify that the proof holds as long as in the limit of n -> infinity the empirical risk tends to 0.
 * After theorem 2.1, the authors provide two examples in Section 3, first for one-dimensional examples and then for radially symmetric bump functions. In these cases, they provide characterisations of the minimum norm interpolant.
  * From Section 4 onwards, the authors move on to discuss generalisation, optimisation and implicit bias.
  * In Line 170 the authors mention the connection between Barron semi-norms and generalisation, following with connections of related work in the space of generalisation and optimisation (though plenty is missing here, see below).
  * In Section 5, the authors provide a set of experiments in which they test the behaviour of various optimisers on some of the tasks that they discussed previously (optimisers include SGD, SGD with momentum, Adam, BFGS). The experiments assess the effects of data dimensions (Figure 1), optimisers, decay values and gain (Figure 2, where they show the importance of gain in learning smooth convex functions across optimisers), and the gap between the optimisation solution and the optimal interpolation solution at a data dimension (Figure 3).


**Strengths:**

Strengths:
* The paper tackles an important problem: understanding the convergence properties of neural networks (albeit in a limited setting) and connects it with generalisation. The second half of the paper assesses the impact of optimisation in a series of limited settings.
* The paper is clearly written, though at times relies too much on previous work and the setting is not clear enough, such as lines 189-200.


**Weaknesses:**

Weaknesses:
  * The paper applies to a very limited setting, namely 1 layer Relu nets trained on regression tasks.
  * The second part of the paper (Section 4 onwards, experiments on optimisation and implicit bias) does not connect strongly with the first.
  * There is nothing in Thm 2.1 about the implicit bias of specific optimisers. Moreover, this area has been studied extensively in the literature (beyond 1-layer nets), and that literature is not included or mentioned at all in the entire paper. Relevant works include Barrett and Dherin, Smith et al, the momentum recent paper. These works can help explain some of the empirical results shown in the paper, since they show that GD, SGD and SGD with momentum respectively minimise `||\nabla_{\vtheta} L||_2^2` where L is the loss function and $\theta$ are the learned parameters. This quantity is very connected to parameter norms in general networks, though perhaps not in the case of 1-layer Relu nets trained on regression tasks with mean squared error , since there  ``||\nabla_{W} L||_2^2 = (\sum_i  f(x_i) - y_i)  \nabla_{\vtheta} f(x_i) and \nabla_{\vtheta} f(x_i) = Relu’(W x_i + y_i ) x_i``.
  * The paper title (and parts of the paper) mention “shallow” neural networks. The paper does not work on shallow networks (which are usually referred to networks with a small number of layers) but to 1 layer networks. I suggest the authors replace the wording “shallow” and use the more appropriate language.
  * The plots are  are lacking relevant information such as x and y axis labels  and hard to read physically (legends are very small, and the lines use the same type of line and similar colours).
  * Other related works missing: when describing the study of the bias of SGD towards flatness, many recent works are missing, including  Cohen et al,  Gradient Descent on Neural Networks Typically Occurs at the Edge of Stability, who show how increasing learning rates leads to SGD traversing flatter areas.


**Questions:**

Can the authors comment on the weaknesses above, address the connection between the theoretical aspect of the work and the optimisation aspect discussed in the second part of the paper, as well as the other aspects highlighted in the weakness section above.

**Limitations:**

The authors did not discuss limitations in their work (as recommended by the conference). I have highlighted a few limitations above, including the limited applicability inside the 1 layer network family and missing connections with the implicit regularisation literature coming from optimisation.

---

> ### Author Rebuttal · Authors · 2023-08-08
>
> > The paper applies to a very limited setting, namely 1 layer Relu nets trained on regression tasks.
>
> As we note in the final paragraph of Section 2 (Lines 115 - 117), our methods apply to (regression problems in) all function spaces which satisfy a Rademacher bound, a suitable compact embedding and a direct approximation theorem. Spaces with such properties were developed for instance by E and Wojtowytsch in *On the Banach spaces associated with multi-layer ReLU networks* in the context of multi-layer perceptron architectures and non-linear function classes for infinitely deep ResNets ("neural ODEs") with these properties are developed by E et. al.: *The Barron Space and the Flow-Induced Function Spaces for Neural Network Models*. However, even in those established settings, minimum norm interpolants are not known, complicating empirical validation. We believe that this may become more accessible within the next years.
>
>
> > The second part of the paper (Section 4 onwards, experiments on optimisation and implicit bias) does not connect strongly with the first.
>
> We consider the second part of the paper complementary to the first: In the first, we establish the significance of minimum norm interpolation with regularization, while in the second, we empirically study the impact of regularization and use minimum norm interpolants to study the implicit bias of optimization algorithms.
>
>
> > [...] This area has been studied extensively in the literature (beyond 1-layer nets), and that literature is not included or mentioned at all in the entire paper.
>
> We would be very grateful to the referee if they could provide a more precise reference for Smith et. al. so that we can incorporate missing works in the final version of the document.
>
>
> > I suggest the authors replace the wording "shallow" and use the more appropriate language.
>
> We believe that "perceptra" in place of "shallow networks" would be appropriate.
>
> > The plots are are lacking relevant information.
>
> We will replot the figures to address their deficiencies.
>
>
> > Other related works missing.
>
> We are grateful to the referee for pointing us to missing works and will use the feedback when preparing the final version, if accepted.

---

> > ### Comment · Reviewer_Gs5Q · 2023-08-11
> > **Rebuttal update**
> >
> > I thank the authors for the rebuttal. Having read the rebuttal and other reviews I decided to keep my score.

---

### Official Review · Reviewer_CjUq · 2023-07-07

**Soundness:** 3 good
**Presentation:** 3 good
**Contribution:** 3 good
**Rating:** 5
**Confidence:** 2

**Summary:**

This paper studies the optimization and generalization for two-layer ReLU neural network models.
Specifically,
* for the generalization part, this paper gives a Barron-norm based generalization bound for the minimize of the $\ell_2$-regularized squared loss function. The bound involves with three terms: sample size $n$, model width $m$ and the regularization weight $\lambda$.
* for the optimization part, this paper conducts numerical experiments on two examples that the minimum Barron-norm solutions can be explicitly obtained, namely, the one-dimensional example and the radially symmetric bump function example. Numerical results reveal that optimization algorithms, initialization scale, and the regularization weight affect the minimum norm interpolant together.

**Strengths:**

* The paper is written clearly and the ideas have been presented in the proper order.
* The studied topic on the interplay of regularization weight, initialization scaling and the optimization algorithm is important to the deep learning community, which may enhance comprehensive perception for optimization and generalization.

**Weaknesses:**

In case I missed something, please correct me if I am wrong.

* The related work description is not comprehensive for me.  Since I am not that familiar with the Barron-norm related fields, I don't know if there exists other works that study the generalization of neural networks in the Barron space. If there indeed exists, it would be better if the authors could make some comparisons on the generalization bound in Theorem 2.1. with others that are related.

* The Baron semi-norm of a general function is usually not easy to assess. In this paper, two examples that can explicitly derive the Baron semi-norm are given: the one-dimensional example and the radially symmetric bump function.
However, these examples are still restrictive in deep learning models. When we look into the definition for the Baron semi-norm, I try to understand this as the expectation version for the $\ell_2$ norm of the weight parameters, which is easier to calculate.
In that sense, I am not sure what are the real advantages for us to consider the Baron semi-norm when compared with the $\ell_2$ norm?


**Questions:**

In case I missed something, please correct me if I am wrong.

* I don't understand why Theorem 2.1. shows that *minimizers of regularized risk functionals converge to minimum norm interpolants of a given target function in an infinite parameter limit.* (Line 26-27).
As far as I can understand, the generalization bound is for the gap between the model function and the target function, but why this result can show that the regularized risk minimizer converges to the minimum norm interpolant?

* Also for the generalization bound in Theorem 2.1., it seems that a two-layer ReLU neural network can learn a target function even for $\frac{m}{n} \xrightarrow{} 0$.
To see this, we let $m = \sqrt{n}$, $\lambda =  n^{-1/4}$, which satisfy the condition (1), then the bound (2) goes to zero as $n$ goes to infinity. That means a two-layer ReLU neural network even with the parameter size less than the sample size can learn an arbitrary target function as long as its Baron semi-norm is finite?
This is counterintuitive as we usually need to use a large model (with parameter size far greater than the sample size) for learning.

---

> ### Author Rebuttal · Authors · 2023-08-08
>
>
> > The related work description is not comprehensive for me. Since I am not that familiar with the Barron-norm related fields, I don't know if there exists other works that study the generalization of neural networks in the Barron space. If there indeed exists, it would be better if the authors could make some comparisons on the generalization bound in Theorem 2.1. with others that are related.
>
> There exist previous works which study generalization in Barron space, most prominently E et. al.: *A Priori Estimates of the Generalization Error for Two-layer Neural Networks*. Due to spatial constraints, the discussion was postponed to the beginning of Appendix D, where the generalization bound is proved. If the referee believes this to be helpful, we will move it to the main article in the final version, if accepted.
>
> The novelty in our work as it pertains to generalization is the following:
>
> 1. We deal with general sub-Gaussian data distributions $\mu$ rather than data distributions with compact support.
>
> 2. We do not control the magnitude of the bias variables.
>
> 3. We consider l2-loss, which is neither globally Lipschitz-continuous nor bounded.
>
> These generalizations require greater technical care and lead to much more involved proofs in some places. They are also the reason behind a logarithmic term which is not present in the calssical setting.
>
> Going beyond our contribution on generalization, we are unaware of other works that obtain uniform convergence on compact sets in this general setting.
>
>
>
> > The Baron semi-norm of a general function is usually not easy to assess. [...] what are the real advantages for us to consider the Baron semi-norm when compared with the $\ell^2$-norm?
>
> Norms of general functions are often hard to compute - the $L^\infty$-norm or Lipschitz constant of a function of many variables is not easy to determine. The same is true of the Barron norm.
>
> For functions given by a (finite) neural network with a single hidden layer, the Barron norm is the sum of the squares of all weights, except in some pathological situations due to the non-uniqueness in function representation
> $$
> 0 \equiv a\sigma(w\cdot x+b) + (-a)\sigma(w\cdot x+b).
> $$
> The sum of squared weights is always an upper bound and easily computed. Thus (an upper bound for) the Barron norm of a given network with one hidden layer is easy to come by. For a general target function $f:\mathbb R^d\to \mathbb R$ which one wishes to approximate on the other hand, it can be very difficult to compute such bounds.
>
> For general functions, the expectation enters as *infinitely wide neural networks are represented by continuous rather than finite sums (integrals) or normalized sums (expectations)*, and many functions which can be approximated well by neural networks with bounded weights cannot be represented exactly. The expectation could be replaced by an integral with respect to a measure or signed measure as detailed e.g. by E and Wojtowytsch: *Representation formulas and pointwise properties for Barron functions*.
>
>
> > I don't understand why Theorem 2.1. shows that *minimizers of regularized risk functionals converge to minimum norm interpolants of a given target function in an infinite parameter limit.*
>
> Convergence is stated in the theorem as "If $(a,W,b)_n$
>
> $\in argmin \widehat{ \mathcal R}_ {n,m_n, \lambda_n}$ for all $n \in \mathbb N$, then every subsequence of $f_n:= f_{(a,W,b)_ n}$ has a further subsequence which converges to some limit $\hat f^* \in \mathcal B$ with $\hat f^* =f^*$ $\mu$-almost everywhere and $[\hat f^*]_ {\mathcal B}\leq [f^*]_ {\mathcal B}$. Convergence holds in $L^p(\mu)$ for all $p<\infty$ and uniformly on compact subsets of $\mathbb R^d$." (Line 92 - 94)
>
> The fact that $\hat f^*$ is a minimum norm interpolant follows from this since $[\hat f^*]_ {\mathcal B} \leq [f^*]_ {\mathcal B}$ for all functions $f^*$ which match the desired values $\mu$-almost everywhere. In $L^2(\mu)$, also a convergence rate for $\|f_n - \hat f^*\|$ is given here, while the statement for uniform convergence is purely qualitative. In $L^p(\mu)$ for $p<\infty$, a (potentially sub-optimal) rate is obtained in Corollary E.2.
>
> Theorem 2.1 relies in part on a generalization bound (Corollary D.4 in Appendix D), but it additionally uses a Direct Approximation Theorem to construct a function of low error which serves as an energy competitor for the minimizer. The proof of locally uniform convergence uses the notion of $\Gamma$-convergence from the calculus of variations. It thus goes beyond a mere generalization bound.
>
>
>
> > Also for the generalization bound in Theorem 2.1., it seems that a two-layer ReLU neural network can learn a target function even for $\frac mn\to 0$. [...] This is counterintuitive as we usually need to use a large model (with parameter size far greater than the sample size) for learning.
>
> The observation is correct. We believe that the difference arises since we do not take into account the training process in this part of the article. Commonly, overparametrized models are used as they are less likely to get stuck in local minima, but from the approximation and generalization perspective, underparametization is not a problem. In this case, we are in the first part of the 'double descent curve', and regularization may not be required (although we do not prove this).

---

> > ### Comment · Reviewer_CjUq · 2023-08-12
> > **Rebuttal update**
> >
> > I thank the authors for the rebuttal. I have decided to keep my score.

---

### Official Review · Reviewer_etEC · 2023-07-11

**Soundness:** 3 good
**Presentation:** 3 good
**Contribution:** 3 good
**Rating:** 6
**Confidence:** 4

**Summary:**

The authors study the generalisation behaviour of two-layer neural networks in
the over-parameterised regime, where the network can perfectly interpolate its
data in many different ways. What can we say about the functions that are learnt
using SGD in these cases? The authors address this question using mathematical
and experimental work.

They first consider the setting where for a fixed dimension $d$ (I think), a
two-layer ReLU network of width $m$ trained on a data set with $n$ points with
labels generated by a target function $f^*$. In the limit where $n$ and $m$ go
to infinity, they show that minimisers of the mean-square loss with small weight
decay penalty, resemble a minimum norm interpolant, which is the equivalent of a
maximum margin classifier in classification.

Since there is no guarantee that a neural network trained using gradient descent
will converge to an interpolant, or which minimiser of its loss it converges to
if it converges to zero loss, the authors also conduct a series of numerical
experiments to analyse the functional norms of neural networks trained using SGD
on low-dimensional inputs. The experiments confirm the importance of
initialisation scale, and small initial weights in particular, and finds some
peculiarities in the behaviour of Adam.

PS: While I know some of the papers on minimum norm interpolators, I am not
knowledgeable in the proof techniques used in this paper.

**Strengths:**

- The author provide a rigorous statement about the behaviour of a certain limit
  of two-layer neural networks.
- They provide a numerical exploration of neural networks trained using
  different gradient-based optimisers, including an analysis of the trade-offs
  between explicit regularistaion and implicit bias.


**Weaknesses:**

- A minor, but important point: the axes labels and titles of the plots are way
  too small to be readable.


**Questions:**

n/a

---

> ### Author Rebuttal · Authors · 2023-08-08
>
> We would like to thank the reviewer for their kind feedback and insightful comments. We will replot the Figures in the final version with larger labels, if accepted.

---

### Official Review · Reviewer_jgQF · 2023-07-21

**Soundness:** 3 good
**Presentation:** 1 poor
**Contribution:** 2 fair
**Rating:** 6
**Confidence:** 4

**Summary:**

This work studies the global minimizer of the empirical risk with $\ell_2$ norm regularization of the parameters. It considers a one hidden ReLU layer architecture with square loss.
The main result roughly states that for some asymptotic regime on $\lambda$ (reg parameter), $n$ (number of data points) and $m$ (network width), the minimizers of the considered regularized loss converge towards a minimal norm interpolant. Besides this asymptotic result, it also provides a convergence rate (in $L_2$ functional norm) of the estimated function towards the truth.

-----------

I acknowledge having read the author's rebuttal. The authors answered my main concerns and I thus raised my score. I still think that the writing needs a lot of polishing though and would advice the authors to take into account all the suggested modifications in their revised version.

**Strengths:**

I find the convergence result towards the truth very interesting, ie the second part of Theorem 2.1.
Getting non-asymptotic generalisation bounds for the considered problem is of particular interest and I really enjoyed the proposed result, which gives nice dependencies in the different parameters $\lambda,n,m$ and $[f^{*}]_{\mathcal{B}}$.

I think this result alone deserves a deeper investigation/understanding, as I believe it to be the most significant contribution of this work: I would suggest the authors to put an emphasis on this particular point of their main theorem as being their main contribution.

**Weaknesses:**

Overall, I find the paper misleading. The writing choices make the message unclear: I am unsure of the conclusions/insights brought by this work. Additionally, I think that the paper lacks contextualizing the results with the literature; numerous works are not cited, while they cleary deserve to be discussed in my opinion.
Lastly, I find the overall contribution rather weak: to me, the only real contribution is the Theorem 2.1, which is not discussed enough and might not be a strong enough result to justify a publication alone. The remaining of the paper is then mostly corrolaries of Theorem 2.1 and numerical experiments, which are of minor interest to me.
I detail all these points below.

First, I find the writing misleading/message unclear. Already in the abstract, the following sentence is problematic: "Our analysis yields convergence to a minimum norm interpolant [...]". This paper is actually about the convergence of the global minimum towards the the min norm interpolant when $n,m,\frac{1}{\lambda}$ converge to $0$, but I first understood this sentence as "some optimisation algorithm converges towards the min norm interpolant when $t$ (iterations of opt algorithm) go to $\infty$". The abstract does not make it possible to distinguish between both cases, while the latter is actually much stronger.
This might seem a small detail, but I think already illustrates how the message is not totally clear and it is sometimes hard to understand where the authors want to go.
The other main confusion to me comes from the Section 3. I only understood its purpose once I read Section 5. Actually, I think Section 3 does not deserve so much space, as its goal is only to provide two examples where the min norm interpolant is exactly known (which are not original contributions). I would prefer these results to just be small paragraphs in Section 5.1 and 5.2.
The message is not even made clearer in the conclusion: only one sentence summarises the paper contribution and interest, while the remaining of the conclusion are experimental observations that could easily be postponed to the appendix.
As another example, implicit bias is mentioned a lot in the literature but is never described, making it hard to follow for a non-expert.

A lot of significant related works are not cited, while they clearly deserve to be discussed into details.
For example, there are many works that aim at describing minimal norm interpolants which are not even cited here [1,2,3]. Also, the semi-Baron norm in the unidimensional case has also been characterised by [4].
The seminal work of [5] is missing in the implicit bias paragraph. Actually the implicit bias does not exactly consider the same norm on the parameters, as it also counts the bias terms (see [2] for a further discussion on this aspect). Is there a particular reason you are not counting the norm of the bias terms here?
About the generalization bounds for small Baron norms functions, I believe it to be closely related to [7,8]. How are the result from Appendix C different?
It has recently been proven by [9] that the implicit bias of GD does not always lead to minimal norm interpolants, but sometimes to different kinds of interpolants: this contradicts some of the claims made in some paragraphs of the paper (eg  in Section 4).
About the conservation of symmetries when training neural networks, I guess you could also mention [10].

The main result of the paper is Theorem 2.1. I think it deserves a much deeper discussion. The paper gives no idea of how the authors manage to prove it, where lie the difficulties in proving it, what are the reasons for the different needed assumptions/scaling of the parameters? In particular, I would have believed that the Theorem would hold with only assuming $\lambda_n\to 0$ and $m_n\to\infty$. What is the reason for needing the second limit in equation (1) ? I believe it hides the difficulty of the proof, as the first part of Theorem 2.1 seems easy to me: the result would be a stronger contribution if it indeed is not so easy to prove.
Actually I really like the second part of the Theorem (rate of convergence) and believe it to be a significant result. I would have liked a deeper discussion on it. Why do we need $\mathbb{E}[\|x\|]+\sigma^2\geq 1$? I believe this is not really needed as we could otherwise deal with a rescaling argument.
About the fixed $n$ setting, [6] actually proved that having any $m\geq n+1$ does not change the minimum of the problem (ie having $m=n+1$ or $m=\infty$ is the same), so I actually think having separately $\lambda_m\to 0$ and $m\geq n+1$ is sufficient in that case.

-----------------
Minor remarks:
- in Section 2, why postpone the definition of $\mathcal{B}$? it can be very shortly defined
- In Section 3.1, the minimum might be different as you also need to consider non-smooth functions. However, using the characterisation of  [4], it is actually the correct minimum
- Theorem 3.3 only seems to be a corollary of Theorem 2.1 and Proposition 3.2 (and thus deserves to be called a corrolary in my opinion)
Also, $f^*$ is not precised here.
- Section 5: the implicit bias also depends on the initialisation regime -> it should also be mentioned
- the sentence line 239 is not clear. Actually, what is happening here seems close to what is described in [2]: for small initialisation, the implicit bias towards min norm (counting bias terms) interpolant kicks in, which leads to the unique min $|x|$. For larger initialisation, there is no such implicit bias and only an explicit regularisation (which does not count the bias terms here) and thus allows for less sparse minima as $f(x)=\max(|x|,1)$.
- for $\lambda=0$ and large $\alpha$, it seems we are in the NTK regime. In this case, we have another implicit bias towards min spline interpolant (see eg [11])
- in Section 5.2, what is $n$?

-----------------
References:
[1] Debarre T, Denoyelle Q, Unser M, Fageot J. "Sparsest piecewise-linear regression of one-dimensional data"
[2] Boursier E, Flammarion N. "Penalising the biases in norm regularisation enforces sparsity"
[3] Stewart L, Bach F, Berthet Q, Vert JP. "Regression as Classification: Influence of Task Formulation on Neural Network Features"
[4] Savarese P, Evron I, Soudry D, Srebro N. "How do infinite width bounded norm networks look in function space?"
[5] Lyu K, and Li J. "Gradient descent maximizes the margin of homogeneous neural networks"
[6] Wang Y, Lacotte J, Pilanci M. "The hidden convex optimization landscape of regularized two-layer relu networks: an exact characterization of optimal solutions"
[7] Kurková V, Sanguineti M. "Bounds on rates of variable-basis and neural-network approximation"
[8] Bach F. "Breaking the curse of dimensionality with convex neural networks"
[9] Chistikov D, Englert M, Lazic R. "Learning a Neuron by a Shallow ReLU Network: Dynamics and Implicit Bias for Correlated Inputs"
[10] Hajjar K, Chizat L. "Symmetries in the dynamics of wide two-layer neural networks"
[11] Jin H, Montúfar G. "Implicit Bias of Gradient Descent for Mean Squared Error Regression with Two-Layer Wide Neural Networks"

**Questions:**

What is the main message of this work?
What are the technical difficulties/main novelties of Theorem 2.1?

---

> ### Author Rebuttal · Authors · 2023-08-08
>
> We are grateful to the reviewer for their detailed feedback. We are unable to fully address all points within the character limit, but we will take their full feedback into account for a revision.
>
> > What is the main message of this work?
>
> The key messages of this work are:
>
> 1. Neural network minimizers of *regularized* risk functionals converge to minimum norm interpolants (Section 2).
>
> 2. Empirically, common gradient-based optimization algorithms often find minimum norm interpolants even without regularization, but there is a notable impact of algorithm and initializations scheme.
>
> 3. Especially when minimum norm interpolants are known explicitly, it is possible to use this knowledge productively to study the impact of the choice of optimization algorithm and initialization scheme on what kind of solution is found (Sections 3, 4, 5).
>
> The program is implemented primarily for neural networks with a single hidden layer, but parts of it generalize to function spaces for more general situations (Lines 115-117).
>
> > What are the technical difficulties/main novelties of Theorem 2.1?
>
> Due to spatial constraints, this discussion was postponed to the beginning of Appendix D, where the generalization bound is proved. We will be sure to include it in the main body in a revised version. The main contributions are:
>
> a) We deal with general sub-Gaussian data distributions $\mu$ rather than data distributions with compact support.
>
> b) We do not control the magnitude of the bias variables.
>
> c) We consider l2-loss, which is neither globally Lipschitz-continuous nor bounded.
>
> d) We use the tools of $\Gamma$-convergence from the calculus of variations to prove *uniform* convergence on compact subsets, even away from the support of the data distribution (see also the scaling constraints between $m, n, \lambda$ below).
>
> The combination of a), b), c) requires a more careful analysis than previous works. To the best of our knowledge, d) is novel and the tools of $\Gamma$-convergence have not been used in this context before.
>
> > The abstract does not make it possible to distinguish between [convergence of minimizers and convergence of an optimization algorithm]. [...] A lot of significant related works are not cited, while they clearly deserve to be discussed into details.
>
> We thank the reviewer for bringing this to our attention and will be sure to fix this.
>
> > Is there a particular reason you are not counting the norm of the bias terms here?
>
> We are not considering the magnitude of the biases in order to utilize known minimum norm interpolants. We are aware that this follows the convention of only part of the research community (such as Hanin (2021) and Ongie et. al. (2019) in our references).
>
> > About the generalization bounds for small Baron norms functions, I believe it to be closely related to [7,8].
>
> We would like to ask if [7] is the intended article as we were unable to find any mention of generalization there.
>
> > Why do we need $\mathbb E\|x\| + \sigma^2 \geq 1$? I believe this is not really needed
>
> This is true, but the exact presentation of the generalization bound would change. We assumed this lower bound to simplify the presentation and some constants. It is neither crucial to our argument, nor did we find it overly restrictive.
>
> > About the fixed $n$ setting [...] having separately $\lambda\to 0$ and $m\geq n+1$ is sufficient in that case.
>
> Using $m\geq n+1$ neurons, perfect interpolation is always possible. In general, there are minimizers which can be represented using $m+1$, but not $m$ neurons (for instance, add break points to a convex one-dimensional function in the setting of Section 3.1). Different versions of this result can be stated under different assumptions.
>
> > In Section 3.1 [...] you also need to consider non-smooth functions.
>
> We concur: ''The same estimate holds for non-smooth Barron functions if the second derivative is interpreted as a Radon measure.'' (Line 126 - 127) We do not require smoothness in Proposition 3.1 and state that any convex interpolant with the correct derivatives at $\pm\infty$ is a minimizer. Our goal was to give intuition without technicality.
>
> > What is the reason for needing the second limit in equation (1)?
>
> $\lambda > 0$ is used to bound the Barron norm. If $\lambda$ is too small compared to $\log n/\sqrt n$, then an undesirable trade-off between fitting labels at a small number of data points and keeping the norm small may be possible. The network may fail to generalize. If on the other hand $\lambda$ is too small compared to $1/m$, then it may be possible to perfectly fit all labels (e.g. if $m > n$), but without sufficient regularization, we lose control of the norm and thus the generalization gap. More precisely:
>
> $$
> [f_n ]_ {\mathcal B} \leq \frac1{ \lambda_n }  \widehat{ \mathcal R}_ {n,m_n, \lambda_n} (a_n, W_n, b_n)
> \leq  \frac 1 {\lambda_n} \left( \frac1{m_n}  + \frac{\log n} {\sqrt n} + \lambda _n [f] _ {B} + l.o.t. \right)
> $$
>
> We require that $1/(\lambda_n m_n), \log n /(\lambda_n\sqrt n) \to 0$ to prove the $\Gamma$-convergence of rescaled risk functionals $\frac1{\lambda_n} \widehat{\mathcal R}_ {n,m_n,\lambda_n}$ to $[\cdot]_{B}$ and $+\infty$ respectively (depending on whether the data is fit).
>
> The details can be found in Appendix E.2. By proxy, it is required for the uniform convergence to minimum norm interpolants on compact sets, also away from the support of the data distribution $\mu$. Theorem 2.1 holds more generally without an assumption of exact minimality if
> $$
> \widehat{\mathcal R}_ {n,m_n, \lambda_n} (a_n, W_n, b_n) - \lambda_n [f^*]_{B} \ll \lambda_n.
> $$
> This is ensured for minimizers by the second limit.
>
> > In Section 5.2, what is $n$?
>
> For plots in Figure 3, the training set size varied (10^4, 10^5 and 10^6 data points respectively) as described in the caption and the plot titles. We will fix the fact that the plot labels are too small to read. For Figure 4, $n= 10^5$. We will add this information to the caption.

---

> > ### Comment · Reviewer_jgQF · 2023-08-14
> > **Author rebuttal**
> >
> > I thank the authors for their detailed answer. First, I apologize for the reference [7] that is indeed not about generalization bound.
> >
> > I think the reason for needing the second limit in equation (1) is a very interesting problem and that a further discussion on this aspect should be found in the paper. Actually, I am not sure to totally understand your explanation. In other words, what you describe could be
> >
> > > If $\lambda$ is too small for large $m$, then we could have catastrophic overfitting.
> >
> > Yet, I am not sure to understand why such a catastrophic overfitting might happen when there is no label noise (as in Theorem 2.1). Indeed, in that case overfitting can always be reached by $f^*$, which is small norm. As a consequence, the regularized ERM always has a norm smaller than $f^*$. Unfortunately, this contradicts your argument. As a consequence, I am still not totally convinced for the need of such a limit (except from an analytical point of view).
> >
> > On the other hand, if there was label noise, it is indeed known that catastrophic overfitting could happen. This was for example proven in the very recent paper [1]. In the presence of label noise, I would thus understand why this second limit is needed; but I currently don't without noise.
> >
> > About the use of $\Gamma$ convergence tools, I would also suggest to mention the paper [2], which deals with a similar (yet different) type of problem and also uses such tools.
> >
> > As in my original review, I believe that the technical result (Theorem 2.1) presented in this work is worth publishing. However I still think the current paper requires a substantial rewriting to deliver a much clearer and impactful message. As a consequence, I decide to maintain my score
> >
> > [1] Joshi, Nirmit, Gal Vardi, and Nathan Srebro. "Noisy Interpolation Learning with Shallow Univariate ReLU Networks."
> > [2] Neumayer, Sebastian, Lénaïc Chizat, and Michael Unser. "On the Effect of Initialization: The Scaling Path of 2-Layer Neural Networks."

---

> > > ### Author Response · Authors · 2023-08-15
> > >
> > > We are grateful to the reviewer for their response and we understand the need to more carefully explain the result and its proof in the main text, which would easily be possible within an additional page, if accepted.
> > >
> > > We do, however, maintain that the proposed counterexample misses a core concept of the problem: In general, there exists no representative $\hat f^*$ which
> > >
> > > 1. coincides with the target function $f^*$ $\mu$-almost everywhere and
> > >
> > > 2. can be expressed by a neural network with $m < \infty$ neurons.
> > >
> > > This is for example true if $f^*$ is compactly supported or radially symmetric in dimension $d>1$ (e.g. $f_d^*$ in our primary example). In the situation that the target function *can* be represented by a finite neural network with $M$ neurons $\mu$-almost everywhere (in particular, independently of $n$), the scaling condition could indeed be replaced by the requirement $m>M$ and the Barron norm of the finite representative could be used.
> > >
> > > On the other hand, if the target function cannot be represented by a finite neural network $\mu$-almost everywhere, we cannot compare the function value of the regularized ERM directly to that of $f^*$ to obtain an upper bound. The regularized ERM with a given number of neurons may have a Barron norm *larger* than $f^*$ since an approximation result is required. This is generally true even if $m>n$ since the finite neural network that interpolates given data depends at least on $n$ and may have larger Barron norm than $f^*$. Thus the scaling condition is necessary to treat a larger class of functions than finite neural networks.

---

> > > > ### Comment · Reviewer_jgQF · 2023-08-16
> > > >
> > > > I thank the authors for this clarification. I indeed forgot this technical point in my last message. I now clearly understand how the second limit in Eq. (1) is required to hinder catastrophic overfitting. As a consequence, I decide to raise my score and summarize my opinion (which is more important than my actual score) about this work below:
> > > >  - its technical contribution is really nice and is worth acceptance in my opinion
> > > >  - yet, the current paper seems to require substantial rewriting for clarity of exposition

---

### Decision · Program_Chairs · 2023-09-21

**Decision:**

Accept (poster)

**Comment:**

The authors investigate the generalization performance of two-layer neural networks in the interpolation regime. They provide both an asymptotic convergence result on the regularized empirical risk minimization (ERM) towards a minimal norm interpolant and a convergence rate of the estimated function towards the ground truth. This paper lies on the borderline, with both reasons for acceptance and rejection. Reviewers acknowledge the paper's valuable theoretical contributions. However, they express concern over the clarity of the presentation, suggesting substantial rewriting for improved understanding. I recommend accepting the paper, but I strongly advise the authors to include the reviewers' suggestions in the final camera-ready version.